# Revisiting five decades of $^{234}$Th data: a comprehensive global oceanic compilation.

Elena Ceballos-Romero[1,2], Ken O. Buesseler[2], María Villa-Alfageme[1]

[1]Departamento de Física Aplicada II. ETSIE. Av. Reina Mercedes 4. Universidad de Sevilla. 41012 Sevilla, Spain.

[2]Department of Marine Chemistry and Geochemistry, Woods Hole Oceanographic Institution, Clark Building 447, Woods Hole, MA 02543-1541, USA.

*Correspondence to*: Elena Ceballos-Romero (eceballos1@us.es)

**Abstract.** We present here a global oceanic compilation of $^{234}$Th measurements that collects results from researchers and laboratories over a period exceeding 50 years. The origin of the $^{234}$Th sampling in the ocean goes back to 1967, when Bhat et al. (1969) initially studied $^{234}$Th distribution relative to its parent $^{238}$U in the Indian Ocean. However, it was the seminal work of Buesseler et al. (1992) - which proposed an empirical method to estimate export fluxes from $^{234}$Th distributions - that drove the extensive use of the $^{234}$Th-$^{238}$U radioactive pair to evaluate the organic carbon export out of the surface ocean by means of the biological carbon pump. Since then, a large number of $^{234}$Th depth profiles have been collected using a variety of sampling instruments and strategies that have changed during the past 50 years. The present compilation is made of a total 223 data sets: 214 from studies published either in articles in referred journals, PhD thesis or repositories, and 9 unpublished data sets. The data were compiled from over 5000 locations spanning all the oceans for total $^{234}$Th profiles, dissolved and particulate $^{234}$Th activity concentrations (in dpm L$^{-1}$), and POC:$^{234}$Th ratios (in µmol dpm$^{-1}$) from both sediment traps and filtration methods. A total of 379 oceanographic expeditions and more than 56600 $^{234}$Th data points have been gathered in a single open-access, long-term and dynamic repository. This paper introduces the dataset along with informative and descriptive graphics. Appropriate metadata have been compiled, including geographic location, date, and sample depth, among others. When available, we also include water temperature, salinity, $^{238}$U data (over 18200 data points) and particulate organic nitrogen data. Data sources and methods information (including $^{238}$U and $^{234}$Th) are also detailed along with valuable information for future data analysis such as bloom stage and steady/non-steady state conditions at the sampling moment. The data are archived on PANGAEA repository, with the dataset's DOI doi.pangaea.de/10.1594/PANGAEA.918125 (Ceballos-Romero et al., 2021). This provides a valuable resource to better understand and quantify how the contemporary oceanic carbon uptake functions and how it will change in future.

## 1  Introduction

For several decades, radioactive tracers have been used to gain a better understanding of different oceanographic processes. In the context of the Biological Carbon Pump (BCP) (Eppley and Peterson, 1979; Volk and Hoffert, 1985), radionuclides such as $^{210}$Po and thorium isotopes, are extensively used to study the various physical, chemical or biological processes involved in the particle export and flux attenuation in the oceans (Cochran and Masqué, 2003). Radionuclides are characterized by a unique

property: their half-lives, which account for the time it takes for one-half of the atoms of a radioactive element to undergo radioactive decay and thus transforming into a different isotope. Half-lives are not affected by temperature, physical or chemical state, or any other influence. As a result, the concentration of naturally occurring radioactive elements varies over time at well-characterized decay and production rates. Observations of radionuclides' distributions in the water column, in time and space, provide valuable insights into the presence and rates of ocean processes on spatial scales from local to basin-wide and timescales of days to millenniums depending on the radionuclide employed.

The naturally occurring radioisotope $^{234}$Th has been commonly used to understand natural aquatic processes in four major areas: particle cycling, horizontal transport, sediment dynamics, and vertical transport (Waples et al., 2006). $^{234}$Th is collected by a variety of sampling procedures since its initial use by Bhat et al. (1969). $^{234}$Th chemistry dictates that it is adsorbed onto particles surface and effectively scavenged from the dissolved water phase. Hence, when biological activity is high, $^{234}$Th is removed from the surface ocean and transported downward by the sinking particles, thereby generating a deficit relative to its soluble or "conservative" parent $^{238}$U. This deficit can be used to calculate the downward flux of $^{234}$Th. An excess in $^{234}$Th activity – i.e. a daughter concentration higher than the parent one - is attributed to fragmentation processes that result in the conversion of sinking to non-sinking particles, generically termed "remineralization" (Maiti et al., 2010). Due to its short half-life of $T_{1/2} = 24.1$ days (decay constant: $\lambda = \ln2/T_{1/2} = 0.02876$ d$^{-1}$, mean lifetime: $\tau = 34.8$ d) and its particle reactivity in seawater it is suitable to trace processes occurring in the upper ocean on time scales from days to months (van der Loeff et al., 2006), or even shorter when there is high scavenging by particles (Turnewitsch et al., 2008) (see Section 4.2).

$^{234}$Th has been an indispensable tool in many oceanographic field expeditions. The most widespread application of the $^{234}$Th approach is to estimate the gravitational settling of carbon as Particulate Organic Carbon (POC) out of the surface layer, which results in a downward flux of POC (see e.g., review by Le Moigne et al., 2013b and references therein). To a lesser extent, this radionuclide is also used to estimate the downward flux of other elements to the deep ocean, such Particulate Inorganic Carbon (e.g., Le Moigne et al., 2013a; Wei et al., 2011) Biogenic Silica (e.g., Buesseler et al., 2005; Lemaitre et al., 2016; Le Moigne et al., 2013a), Particulate Organic Nitrogen (PON) (e.g., Buesseler et al., 1992; Charette and Buesseler, 2000; Murray et al., 1996), or trace metals fluxes (e.g., Black et al., 2018; Lemaitre et al., 2016, 2020; Weinstein and Moran, 2005).

It was in the 90's that an increasing number of studies for $^{234}$Th took place. This increase in use is in part due to a variety of new sampling instruments and strategies that have changed over the years. In 2006, a special issue entitled Future Applications of $^{234}$Th in Aquatic Ecosystems (FATE, https://www.sciencedirect.com/journal/marine-chemistry/vol/100/issue/3) was published in Marine Chemistry with the purpose of thoroughly reviewing the use of $^{234}$Th in aquatic systems. Papers included reviews of the applications and future uses of this radiotracer (Benitez-Nelson and Moore, 2006; Waples et al., 2006), discussions on the techniques and methodologies used for $^{234}$Th analyses (van der Loeff et al., 2006), the impact of POC:$^{234}$Th ratios and their sampling methodology on POC flux estimates (Buesseler et al., 2006), and $^{234}$Th sorption and export models in the water column (Savoye et al., 2006), among other topics. As one of the most actively used tracers in oceanography, Waples et al. (2006) already reported 237 papers dealing with $^{234}$Th published in refereed journals. However, after 5 decades

of extensive use, a unique repository of $^{234}$Th measurements has never been compiled and $^{234}$Th data remain scattered when not belonging to major sampling programs (see Section 4 for details). Therefore, it is valuable to bring together all existing $^{234}$Th data, along with appropriate metadata, in one repository to facilitate further use and analysis.

Previous efforts compiling $^{234}$Th-based data have been created to access $^{234}$Th-derived POC fluxes (see Le Moigne et al.,
2013b), total $^{234}$Th activity from the surface to 1000 m deep (Le Gland et al., 2019) and, more recently, POC:$^{234}$Th ratios (see Puigcorbé et al., 2020 and doi.pangaea.de/10.1594/PANGAEA.911424). In contrast, we have compiled the complete results of $^{234}$Th measurements in sea water and particles at every depth, location, and time of sampling. The compilation can be found in doi.pangaea.de/10.1594/PANGAEA.918125. This article is the report of the compilation, a unique dataset to better understand and quantify how the contemporary oceanic carbon uptake functions and how it will change in future.

The goal of this effort is to serve as a basis for an open-access, long-term and dynamic oceanic repository of $^{234}$Th measurements and valuable metadata to be used in an accessible, easily findable, and inter-operable way that grows from now on from the contribution of other authors involved in $^{234}$Th sampling. Moreover, given the great number of metadata and parameters compiled, the compilation offers multiple ways to use $^{234}$Th, even opening the possibility of exploring new applications. For that reason, we have chosen not to compile the derived parameters reported by other authors, such as $^{234}$Th
downward fluxes or $^{234}$Th-derived POC fluxes, but rather provide the data necessary for others to make such analyses, open to the criteria, modeling, and interpretation chosen by each researcher.

## 2   Data

### 2.1   Data organization

We have gathered data sets consisting of $^{234}$Th measurements in oceanic waters sampled between 1967 and 2018. The
compilation includes a total of 56631 data points for $^{234}$Th activity concentrations (in dpm L$^{-1}$, referred to as simply $^{234}$Th concentrations from now on), distributed as follows: i) 21457 for total, ii) 6591 for dissolved, iii) 13977 for particulate $^{234}$Th measurements, and iv) 10856 and v) 3750 for POC:$^{234}$Th and PON:$^{234}$Th ratios, respectively (in μmol dpm$^{-1}$). Note that when total carbon is provided instead of POC:$^{234}$Th ratios (such as in e.g., Owens et al., 2015) this is indicated in the "methods" section of the "metadata" a sheet (see Section 2.1.1.) Additionally, $^{238}$U concentrations (18256 data), and POC (7651 data) and
PON (1740 data) concentrations (in μmol L$^{-1}$) are also reported. POC and PON concentrations are referred to as "CHN" (Carbon-Hydrogen-Nitrogen) in the compilation. We are aware that CHN is not the only analytical method that can be used to determine POC and PON concentrations. An elemental analyzer – isotope ratio mass spectrometer (EA-IRMS) is also widely used for this purpose (see e.g., studies by Lemaitre et al., 2018; Planchon et al., 2015), but we have used this notation for the sake of simplicity. Please refer to the original source for the analytical method used to measure these data. Temperature and
salinity values are also included when possible, making a total of 5652 values compiled for temperature and 12721 values for salinity. When fields are missing for a given record, the data are entered as "-999".

The data have been extracted from a total of 219 different studies published in refereed journals between 1969 and 2020, 5 PhD dissertations, 9 data sets accessible in 4 different public repositories, and 9 unpublished ones directly provided by authors (listed in *Table 1*) spanning all oceanic regions (*Figure 1a*) and compiled in a total of 223 data sets. The data repositories include: 1) BCO-DMO® ("Biological & Chemical Oceanography Data Management Office", https://www.bco-dmo.org/), 2) DARWIN® ("Data 110 and Sample Research System for Whole Cruise Information in JAMSTEC", http://www.jamstec.go.jp/e/about/informations/notification_2021_maintenance.html), 3) EDI Data Portal® ("Environmental Data initiative", https://portal.edirepository.org/nis/home.jsp), and 4) PANGAEA® (https://www.pangaea.de/). Additionally, many of the data corresponding to published GEOTRACES® (https://www.geotraces.org/) cruises can be found in GEOTRACES website (see the most recent version of its intermediate data product released in 2021 (GEOTRACES Intermediate Data Product Group, 2021)).

Each data set is univocally identified with a unique integer record ID number (denoted as "Reference_USE") and consists of two tables: i) the "metadata" and ii) the "data".

### 2.1.1. "Metadata" sheet

The "metadata" table is a list of the data origin in its broadest sense at a glance. It contains information structured in 6 categories: 1) "REFERENCE_USE", 2) "INFO", 3) "DATA", 4) "METHODS", 5) "ADDITIONAL_DATA", 6) "DATA_SOURCE". The full list of metadata included in each dataset and a brief description of the table fields can be found in the Supplementary Material (see *Table S1)*.

There are data from 379 cruises, covering 5134 locations spanning across all oceanic regions (*Figure 1a*). Some stations were part of cruise transects whereas others were part of small-scale surveys or reoccupation of the same location at different seasons and years. In all cases, sampling region and period – including bloom stage at the sampling moment when indicated by the authors – are given as metadata as part of the "INFO" section, where a total of 17 fields are reported (see *Table S1*).

Information such as the project name, when sampling took place within an observational program, cruise name, leg details, research vessel, and chief scientist are indicated. A summary of the sampling period, the maximum and minimum latitude and longitude of the region surveyed, and the maximum depth sampled for $^{234}$Th is also provided. When available, for the sake of a better interpretation of the data for the accurate POC flux assessment, the stage of the bloom have also been included (categorized as "bloom", "pre-bloom", "post-bloom" and "no bloom"), as recommended by e.g., Ceballos-Romero et al. 2016, 2018. Note that we do not assess the bloom stage but instead we include the information as indicated by the original authors when provided. To distinguish between the periods before and after the peak in primary production, we identify "development of the bloom" and similar expressions as pre-bloom stage and "decline of the bloom" and similar as post-bloom phase. Only when bloom or non-bloom conditions are stated by authors, we assigned these phases. When none of these stages are referenced to, not information is included (noted as "-999"). We acknowledge there could be issues on the way the different authors have decided if the conditions where non-bloom, pre-bloom, bloom, or post-bloom over the different years that are out of the scope

of this compilation and were not evaluated. We plan to address this gap in the compilation in future versions of it (see Section
5).

The "DATA" section provides information of the data set contents at glance with YES/NO indicators to the basic data of $^{238}$U, $^{234}$Th (total, dissolved and particulate phases) and POC(PON):$^{234}$Th ratios. The fraction size(s) details and the number of total stations and samples are also reported. A total of 13 parameters are detailed in this section.

The "METHODS" section is intended to provide useful interpretive information regarding how the sampling and/or measurements were accomplished at a glance. It provides basic information about i) $^{238}$U determination: whether it was directly measured, or salinity derived, in which case the salinity relationship employed is specified; ii) total $^{234}$Th sampling and radiochemical purification methods, sampling methods for the iii) dissolved, and iv) particulate $^{234}$Th phases; and v) the modeling approach followed when $^{234}$Th data were used to estimate POC fluxes (i.e., the assumption of steady (SS) or non-steady state (NSS) conditions). A final space for comments of any kind is included in this section.

The potential of $^{234}$Th data increases when combined with methods that account for export episodes over different timeframes of a bloom period, such as $^{210}$Po-$^{210}$Pb, or sediment traps (see e.g., Ceballos-Romero et al., 2016). For that reason, we included information regarding the availability of some other techniques when combined with $^{234}$Th sampling in the "ADDITIONAL_DATA" section. Additional data of interest are specific with YES/NO indicators for the cases of i) $^{234}$Th underway sampling, ii) sediment traps deployments, iii) $^{234}$Th was paired with $^{210}$Pb-$^{210}$Po disequilibrium sampling, and iv) CHN data. Note that this section is merely intended as informative of the sampling methods used complementarily to the $^{234}$Th technique. Both $^{234}$Th underway data and POC:$^{234}$Th and PON:$^{234}$Th ratios from sediment traps are reported when available. However, $^{210}$Pb-$^{210}$Po concentrations are not compiled in this dataset. The availability of these data is indicated as reported by authors. We have only consulted publications, and cruise reports when accessible to gather information for these metadata so we acknowledge that information as to the existence of these data might be missing. We therefore recommend using them cautiously when stated "NO (available)" but fully trust it when stated "YES". To report an update to the metadata or an error in the data compiled, we encourage authors to contact us, and changes will be included in future versions of the compilation (see Section 5).

It is also worth mentioning here that the amount of data that could be additionally reported in the compilation is very extensive due to the wide applicability that has been shown for $^{234}$Th along the years (see e.g., review by Waples et al., 2006). The inclusion of new parameters in a future version of the compilation will be discussed in Section 5 as part of our assessment of steps towards improving the global data set.

Finally, several details regarding the data source are included in the 8 fields specified in the "DATA_SOURCE" section, including a YES/NO indicator for the publication date. The data owners are always clearly indicated by the first author(s) of the publication or data set. In the case of data published in research articles, the journal and the publishing year are indicated, while in the case of unpublished data obtained from personal communication (as is the case for 9 datasets), "np" (stating for

non-published) is given in the journal information and no data are provided as publication year (indicated as "-999")). If data had been assigned a Digital Object Identifier (DOI), this is included in DOI/others. When no DOI is available, the data URL source –either database or publication links, is included instead. Other URL sources or personal communication from data source is keyed by the text variable "data_resource" in the "metadata" table, when available.

Most of the measurements were obtained from publications. In these cases, if the data were transcribed from tables, the table number is also given as "data localization". If data were only available graphically, a computer program to digitize the data from plots was used (WebDigitizer, https://automeris.io/WebPlotDigitizer/) and the figure number is also given. In the rare cases in which data were not accessible through any of these procedures, the authors were directly contacted for data. For those cases, the author(s) contacted is(are) indicated in the "data localization" field.

Finally, a space for further links or information of interest is provided under the text variable "other DOI/resources". In the few cases that the same data were reported in another publication (e.g., Murray et al., 1996, Dunne et al., 1997 and Murray et al., 2005 reported the same data from the U.S. JGOFS (Joint Global Ocean Flux Study) program during 1992 in the Equatorial Pacific), it was indicated under this text variable.

### 2.1.2. "Data" sheet

The "data" table is the data set core and contains all the data detailed above (more details in *Table S2*).

Each data point is accompanied by cruise and station IDs, location – latitude, longitude, depth, and additionally bottom depth when available – and sampling date – including month-date-year and Day of Year (DOY) formats –. All $^{234}$Th (total, dissolved and particulate) concentrations were converted to dpm L$^{-1}$ (density 1027 kg m$^{-3}$) if not already reported in these units. POC:$^{234}$Th and PON:$^{234}$Th ratios are given in µmol dpm$^{1}$ and include samplings with sediment traps and filtration methods, in which we

report i) bottles- Go-Flo and Niskin types- ii) filtration systems, iii) SPLITT (split flow-thin cell fractionation), and iv) (large or small volume) in-situ pumps for either the entire particulate fraction or two sizes classes (preferably 1-53 µm and > 53 µm) when available. These size-classes were chosen largely based on results of Bishop et al., (1977), Clegg and Whitfield (1990), and others, who assumed that the >53 µm size-class was responsible for most of the mass flux into traps. Other cut offs of 51 µm or 70 µm and other size classes are found and are noted when different than 53 µm. The particles' sampling method –

categorized as "method 1" (for filtration methods) and "method 2" (for sediment traps) – and size fractions – categorized as "small" or "large" – are specified as part of the data. Total, particulate, and sediment traps $^{234}$Th sampling depths are separately indicated. When reported, water temperature, salinity, and $^{238}$U measurements are included. Moreover, if accessible, POC and PON concentrations ("CHN" data in µmol L$^{-1}$) are included.

In all cases we assume that the originating authors and editors have undertaken steps necessary to control data quality.

Measurement uncertainties in the data points are compiled as provided by the original authors (e.g., "uncert_total_$^{234}$Th"). Please refer to original source for whether data uncertainty includes only the one sigma counting error, or other factors, such as uncertainty on volumes, detector efficiency, background, etc.

## 2.2 Data formats and availability

The data are archived on PANGAEA repository, with the data set's DOI https://doi.pangaea.de/10.1594/PANGAEA.918125
(Ceballos-Romero et al., 2021). The data table is available for download either as a unique merged file containing all data sets and metadata or as individual excels files.

Moreover, the template followed to compile the data set (including the "metadata" and "data" tables) along with instructions to fill in this template are made available in PANGAEA for any author who either wants to review, complement a data set included in this compilation or contribute to its extension with a new data set. We strongly encourage authors to contact us to submit suggestions or request to amend the data sets compiled.

## 3    Scope and introduction to the data set

In this article, we aim at providing a broad overview of the character of the data sets to be used for different purposes in future studies. We therefore provide several graphics to indicate the scope and nature of the data compiled. These include maps of sampling locations (*Figure 1a*) and time histories of analyses per year for $^{234}$Th measurements (*Figure 2*) among others. Additionally,    a    summary    table    with    the    most    remarkable    metadata    is    also    provided    in doi.pangaea.de/10.1594/PANGAEA.918125 (in the "Download data" section and "View dataset as HTML" option).  Note that the compilation here presented does not overlap the previous compilation by Le Moigne et al. (2013), where $^{234}$Th flux, POC:$^{234}$Th ratios and$^{234}$Th-derived POC fluxes were gathered together along with primary production data mostly for a fixed depth of 100 m or 150 m only.

### 3.1. General overview

In this section we give an overview of the compilation and some important aspects for its use. In section 4 we present a historical review of the 50 years of the $^{234}$Th studies. In Section 5 we discuss steps towards improving the global data set. And in Section 6 we briefly suggest some future perspectives and applications for the data sets.

The sampling locations shown in *Figure 1a* are dominated by cruise tracks mostly in the Northern Hemisphere (NH) (*Figure S1*), although there are also a few locations in the Southern Hemisphere (SH) with significant repeated sampling, especially in the Southern Ocean *(Figure 1b)*. Over a total of 379 cruises compiled, 294 took place exclusively in the NH (78%), only 53 in the SH (14%), and a total of 32 (8%) crossed the Equator and sampled locations in both hemispheres.

Many of these expeditions were carried out in the framework of larger research programs. For those cases, this information has been included as metadata in the compilation. During these 50 years of $^{234}$Th data, we have selected a total of 13 ocean programs where $^{234}$Th has been widely applied and that have provided invaluable knowledge to the $^{234}$Th approach (see *Table 2*). Note that these programs have been chosen with the purpose of providing a wide global distribution and international representation of $^{234}$Th from different institutions and organizations.  A total of 68 (30%) of the data sets were collected during

surveys as part of one of these 13 programs. Additionally, for cruises belonging to specific projects or experiments, this information has also been reported.

Sampling locations for [234]Th also include a number of long-term, high-frequency observations at fixed locations in the open ocean, referred to as (long-term) time-series stations (TSS) from now on. These stations offer a crucial observational strategy for capturing the dynamic temporal changes in ocean conditions and biogeochemical processes at the seasonal and decadal scale. Moreover, it allows investigating short-term episodic events usually unaccounted for by oceanographic cruises. A great number of TSS that are either operational, registered (for future operation), inactive or closed spread across the globe (source:

https://www.ocean-ops.org). We have acknowledged the importance of sampling at these TSS by identifying matches with [234]Th sampling locations (see *Figure 1c)*. Coincidences between locations with [234]Th data available and the position of a TSS station have been reported in the metadata section with the label "time-series" in the "number of stations" field. Note that this should not be mistaken with repeated sampling of a location during a cruise, which has also been acknowledged with the label "reoccupied" in the same field. A total of 98 data sets (44%) reported samplings on at least one TSS location while only 33

(15%) reoccupied the sampling site. The most frequent TSS sampling sites found in the compilation include the *Bermuda Atlantic Time Series Study* (BATS: 31.50°N, 64.10°W, http://bats.bios.edu/), the *Hawaii Ocean Time series* (HOT: 22.45°N, 158°W, https://hahana.soest.hawaii.edu/hot/hot_jgofs.html) at station ALOHA, the *Dynamics of Atmospheric Fluxes in the MEDiterranean* sea (DYFAMED: 43.42°N, 7.87°E, http://www.obs-vlfr.fr/cd_rom_dmtt/sodyf_main.htm) – all of them part of JGOFS. And some other relevant ones such as the *South East Asia Time Series* (SEATS: 18.00°N, 166.00°E,

https://www.odb.ntu.edu.tw/seatsen/#:~:text=The%20SEATS%20time%2Dseries%20station,Arctic%20Sea%20is%20the%2 0largest), also from JGOFS, the *Palmer Antarctica* and *California Current Ecosystem Long-Term Ecological Research Project* (PAL-LTER (https://pal.lternet.edu/) and CCE-LTER (https://cce.lternet.edu/) respectively), or the *Porcupine Abyssal Plain* (PAP, 49.00°N 16.50°W, https://projects.noc.ac.uk/pap/) site sustained observatory among others (*Figure 1c).* More details of the history of these locations repeatedly sampled for [234]Th will be given in forthcoming sections.

Over time, the interdisciplinary scope of the studies and the number of techniques and parameters simultaneously assessed have gradually evolved. Field surveys have changed not only in terms of number but also in terms of strategy (e.g., duration, spatial resolution, and repeated occupation of the same site). This has influenced not only the number of data points measured – with a sustained increasing trend - but also the type of [234]Th data reported (i.e., total, dissolved and particulate phases, and POC(PON):[234]Th ratios). Similarly, it has driven changes in the number of [234]Th studies published. We have identified 3 key

indicators of these inflection points. In the 90's the number of [234]Th studies dramatically increased after the publication of the first empirical routine method to estimate downward POC fluxes in the ocean by Buesseler et al. (1992). Another tipping point took place in the 2000's likely due to significant improvements in the [234]Th methodology derived from the introduction of the small-volume technique (Benitez-Nelson et al., 2001b). Moreover, it is important to note that it was in the late 2000's when Yool et al., (2007) proved that other empirical methods such as the 15N new production techniques (f-ratio) largely

overestimated the efficiency and magnitude of the carbon export. Finally, a later shift took place with the change in the way

ocean is currently explored and the expansion of applications to trace metals and isotopes introduced by GEOTRACES, whose first $^{234}$Th-related publication was released in 2010 with the first version of the "cookbook – sampling and sample-handling protocols" by Cutter et al. (2010). More details of these milestones and the consequences they brought to the $^{234}$Th technique will be discussed in Section 4.

The overall temporal evolution of $^{234}$Th measurements is depicted in *Figure 2*. *Figure 2a* shows the annual number of field expeditions including $^{234}$Th measurements per sampling year between the initial measurements in 1967 to 2018, separated by oceans. *Figure 2b* shows the equivalent figure for the annual oceanic distribution of $^{234}$Th data points and *Figure 2c* shows the histogram of $^{234}$Th data points sampled since 1967 separated by data type. Additionally, the time distribution of annual $^{234}$Th publications (either in referred journals, PhD thesis or repositories) since the first reported publication in 1969 separated by 265    oceans is shown in *Figure S2*.

Despite the fact that the first $^{234}$Th study took place in the Indian Ocean in 1967, this ocean along with the waters from the Arctic Ocean, visited by the first time in 1994, is the least surveyed one during this 50-year compilation (*Figure 2a*). The Atlantic and Pacific oceans dominate the sampling locations, followed by the Southern Ocean, to a much lower extent. Field expeditions to the Pacific Ocean started in 1970, in 1977 to the Atlantic and in 1987 to the Southern Ocean. Other locations 270    include several rivers, lakes, and bays (see *Figure 1a*).

During the 60's–70's, expeditions including $^{234}$Th measurements were scarce, with the total number of annual samplings never exceeding 5. It was in the 90's when the number of $^{234}$Th samplings started to dramatically increase, with an average number of annual expeditions of 15. In the subsequent years, a great number of cruises took place, with an average of 10 per year during the last two decades. However, the annual number of dedicated cruises is not representative of the number of $^{234}$Th 275    points reported by year. In the past, the number of cruises including $^{234}$Th measurements greatly increased over time, and so did the number of data points, while more recently, a larger number of data (*Figure S2*) are reported from a reduced number of cruises (see *Figure 2a)*.

There are several remarkable peaks in the annual record of $^{234}$Th data points which are related to dedicated carbon export programs and experiments that would be detailed in next sections. A delay of about 3-4 years relative to sampling is commonly 280    observed for the publication of these $^{234}$Th data and derived results, which started in 1969 (see *Figure S2*). Namely, as shown in *Figure 2b*, peaks in the number of data points stand out in: i) 1992, when the JGOFS equatorial Pacific process study took place, published by several groups in 1995-1997 (Bacon et al., 1996; Buesseler et al., 1995; Dunne et al., 1997; Murray et al., 1996) and extended in 2001 with the publication of POC:$^{234}$Th ratios from sediment traps results by Hernes et al., (2001); ii) 1997, when the JGOFS Southern Ocean study and the Arctic expedition ARK XIII/2 were carried out, published by Buesseler 285    et al., (2001) and Rutgers van der Loeff et al., (2002b) respectively; iii) 2004, when several intense expeditions took place, with VERTIGO voyages as the most productive ones in terms of data contribution, published by Buesseler et al., (2009) and Lamborg et al., (2008b); the Shelf-Basin Interactions (SBI, https://arctic.cbl.umces.edu/sbi/web-content/) Phase II field

program, published in 2007 by several authors (Lalande et al., 2007; Lepore et al., 2007 and Lepore and Moran 2007); the EDDIES project, synthesized by Buesseler et al., (2008a); and the CROZEX project available in Morris et al., (2007), iv) 2010,

where a total of 17 cruises took place, with the Southern Ocean cruises (Owens, 2013) as the most intense ones in terms of number of [234]Th samples per day and the several GEOTRACES section cruises: 2 cruises to the South Atlantic Ocean (GA10, UK) published by Le Moigne et al., (2014); 3 cruises to the Atlantic Ocean (GA03 (legs 1 and 2), U.S., and GA02 (leg 3) Netherlands) published by Owens et al., (2015); and 2 cruises in the Atlantic Ocean from 64°N to the equator (GA02(legs 1 and 2), Netherlands) published by Puigcorbé et al., (2017a); and finally v) 2018, when 2 GEOTRACES cruises were carried

out in the Pacific Ocean (GP 15 Leg 1, U.S., Pacific Meridional Transect, and GP 15 Leg 2, U.S) unpublished until now that have been reported in this compilation by personal communication from J. Kenyon; and the first field expedition of the EXPORTS program took place (published by Buesseler et al., (2020)).

### 3.2. Total, particulate and dissolved [234]Th

Changes in the type of [234]Th data measured during field expeditions are depicted in *Figure 2c*. While within the first years of

the [234]Th technique, authors focused on the total, dissolved and particulate phases mainly for the study of the scavenging of [234]Th and the partitioning of Th species between phases (see e.g., Bacon and Anderson, 1982; McKee et al., 1986; Murray et al., 1989; Nozaki et al., 1981), in the 90's the broader potential of [234]Th and its application as a tracer for particle dynamics and export fluxes became evident, which drove the inclusion of sampling strategies for the determination of POC(PON):[234]Th ratios. Given the reduced annual number of data points in the 70–80`s, in comparison to subsequent years, it is difficult to

perceive the contribution of each data type to the total number of data points within the first years of study. A detailed review of the evolution of [234]Th data available through years in the framework of the mentioned milestones is given in Section 4.

In summary, total [234]Th is almost always reported by all authors by either direct measurement or determined as the sum of the dissolved and particulate phases. It is worth mentioning that the distinction between dissolved and particulate is operational. Dissolved and particulate specie have traditionally been discriminated by filter sizes, typically of 0.2-l µm pore size (Moran

and Buesseler, 1993), with the submicron colloidal matter – nanometer to submicrometer size range ~0.001-l µm (Stumm, 1977) – included in the dissolved phase. Note that colloidal size ranges have not been included in the compilation but indicated in the metadata section when available. The 0.7 µm cut-off for the dissolved-particulate fractions is very frequent since it is the nominal pore size of the Whatman glass microfiber filters (so-called GF/F) that has been historically used to collect particulate [234]Th particulate (Buesseler et al., 1998). Nonetheless, the use of 1 µm pore size quartz microfiber filters (QMA)

filters has been more common since the early 2000's due to its lower radioactivity blank with direct beta counting (e.g., Buesseler et al., 2001a). A general view of the [234]Th particulate data compiled is provided in *Figure S3*, which shows the locations with [234]Th concentrations sampled on either small (generally from 0.7-1 to 53-70 µm) large (>53 or >70 µm) or both size fractions. An overall 75% of the studies compiled reported particulate [234]Th concentrations, more than half of them (56%) reported data in 2 size fractions.

320 In a reduced number of studies, [234]Th concentrations were determined as a single vertically integrated sample, normally between 0 and 100 m depth (i.e., Buesseler et al., 1998, 1994, 1995; Charette et al., 1999; Charette and Buesseler, 2000; Cochran et al., 2000; Evangeliou et al., 2011; Hall et al., 2000; Ma et al., 2005; Maiti et al., 2008; Schmidt et al., 2002). This sampling approach was deliberately done to reduce sample numbers yet maximize the number of locations sampled, at a time when at sea analyses were more difficult. This has also been indicated both in the metadata and data sections in each individual

325 spreadsheet with the code number "-555" in the depth column (see *Table S1* and *Table S2).*

A total of 66% of the studies reported POC:[234]Th ratios measured with some of the filtration methods (described in detail in Section 2.1.2), while only ~16% reported PON:[234]Th ratios. In ~40% of the studies compiled, POC(PON):[234]Th ratios collected with sediment traps were reported from a great variety of trap designs: Indented Rotating Sphere (IRS), surface-tethered, free-floating, bottom-moored, automated, cylindrical, VERTEX-style, U-type, CLAP-type, RESPIRE type, Free-drifting

330 Lagrangian Sediment Traps (LST), High Frequency Flux (HFF) standard drifting particle interceptor traps (PITS) and, more recently, neutrally buoyant sediment traps (NBSTs) and Particle Export measurement using a LAGRAngian (PELAGRA) trap. An overview of POC:[234]Th data compiled is shown in *Figure S4*.

Note that a specific, more detailed, compilation of global POC:[234]Th ratios in the ocean has been recently published by Puigcorbé et al (2020) (archived in the data repository PANGAEA® under doi.org/10.1594/PANGAEA.911424) with the

335 purpose of elucidating the spatial, temporal and depth variations of this crucial parameter. The authors present a database of 9318 measurements sampled on 3 size fractions ($\sim > 0.7\,\mu m$, $\sim 1$–$50\,\mu m$, $\sim > 50\,\mu m$) collected with in-situ pumps, bottles, and sinking particles collected in sediment traps from the surface down to > 5500 m. Our compilation includes the studies gathered in Puigcorbé et al (2020) and expands the data set with a total of 10851 POC:[234]Th ratios.

### 3.3. [238]U measurements

340 Finally, in order to allow the assessment of POC fluxes, [238]U concentrations have been also included. This is generally reported by authors either by direct measurement or, most commonly, derived from salinity data by applying one of the [238]U-salinity relationships available. From the studies compiled, we have identified a total of 11 salinity-[238]U relationships published in journal articles, chronologically ordered as follows: i) Ku et al., (1977); ii) Broecker and Peng (1982), iii) Coale and Bruland, (1985), iv) Chen et al., (1986), v) Andersson et al., (1995), vi) Delanghe et al., (2002), vii) Rutgers van der Loeff et al., (2006),

345 viii) Pates and Muir (2007), ix) Owens et al., (2011), x) Not et al., (2012), xi) Martin et al., (2013), along with some other relationships also frequently used such as 2.4 dpm/kg at salinity 35, 0.06813 x salinity ‰, or 0.07097 x salinity ‰. The most extended [238]U-salinity relationships are the ones given by Ku et al. (1977) ($A_U = 0.07081 \times salinity$), Chen et al. (1986) ($A_U = 0.0686 \times salinity$) and, more recently, by Owens et al. (2011) ($A_U (\pm 0.047) = 0.0786 \times salinity - 0.315$).

### 4. Discussion: [234]Th timeline

The compilation covers a temporal range of 5 decades. The temporal distribution of oceanic $^{234}$Th measurements begins in 1969 to measure its distribution in the hydrologic cycle (Bhat et al., 1968) and extends up to these days as an indispensable tool in oceanographic expeditions. We consider it was not the passing of years but rather the publication of seminal key studies that delineates the progression of $^{234}$Th studies. For this reason, we have divided the $^{234}$Th technique time history in four well distinguished eras, marked by four publications, summarized in *Figure 3*. Therefore, is the publication date rather than the

sampling date what determines the era to which a dataset belongs. Note that this definition of eras by publications rather than sampling dates implies that cruises that took place within the time frame of a previous era are included in the era of their publication. Delays of up to decades between sampling and publication can ben sometimes found. In addition, in the case of a later publication (from a later era) used data from a cruise that was already published in a previous era, that publication is included in the earlier era, despite of the year of the actual publication.

**4.1. Era 1: "The Old Man and the Sea"**

*1969-1991: The beginning of $^{234}$Th as tracer of particle scavenging.*

*"Now is no time to think of what you do not have. Think of what you can do with what there is."*

*— Ernest Hemingway, The Old Man and the Sea.*

The first era within the $^{234}$Th timeline was initiated by the study of Bhat et al. (1969), in which a total of 6 profiles of total

$^{234}$Th:$^{238}$U ratios in depths up to 250 m were collected on-board the U.S.C. and G.S.S. Oceanographer during 1967 at several sampling sites in the Indian Ocean to study scavenging processes. Polyvinylchloride (PVC) tube samplers of 30 L capacity designed by Shale J. Niskin (Niskin, 1962) were used.

It was in the framework of GEOSECS (the GEochemical Ocean SECtions Study) project (http://iridl.ldeo.columbia.edu/SOURCES/.GEOSECS/) that the concept of scavenging by particles really emerged, especially

for the open ocean. The initiation of global ocean chemistry, hydrographic, and tracer survey efforts took place within this era, especially with GEOSECS, which was fundamentally different in style and scale than anything before. This project provided the first comprehensive data set for the distribution of chemical species in the world ocean between 1972 and 1978 (Moore, 1984). In 1977, the seminal work of Turekian introduced the concept of the "great particle conspiracy" to denote the fact that dissolved elements are, to a certain extent, particle reactive. All this contributed to further increase $^{234}$Th research and pointed

towards the second era of the $^{234}$Th technique.

The pioneering study by Bhat et al., was expanded by many other investigators (e.g., Kaufman et al., 1981; Knauss et al., 1978; Lee et al., 1991; Matsumoto 1975; Tanaka et al., 1983; Tsunogai et al., 1986). All these studies analyzed total $^{234}$Th (both particulate and soluble forms) giving different interpretations to the total $^{234}$Th vertical distribution. Bhat et al. (1969) assumed that essentially all $^{234}$Th in seawater was in particulate form, whereas Matsumoto (1975) assumed particulate $^{234}$Th to be an

insignificant part of the total $^{234}$Th. This led to not very informative scavenging model to be applied to the surface euphotic

layer until the sampling of profiles of [234]Th in both dissolved and particulate was introduced. The analysis of particulate [234]Th was incorporated by Krishnaswami et al., (1976) during a survey in the Pacific Ocean. Several authors sampled both total and particulate [234]Th concentrations (e.g., McKee et al., 1984; Minagawa and Tsunogai, 1980), while the sampling of the [234]Th dissolved phase was initiated by Santschi et al., (1979).Bay From 1979 on, many studies sampled both dissolved and
particulate phases (see e.g., Coale and Bruland, 1985; McKee et al., 1984; Wei and Murray, 1991).

Initially in this this era, little attention was given to [234]Th in comparison to [228]Th and [230]Th for studying scavenging processes in the open ocean (Broecker and Peng, 1983). It was only later, with the influential study of Coale and Bruland (1985) that the role of [234]Th as a tracer of short-term particle dynamics was really highlighted. In that study several profiles of [234]Th in both dissolved and particulate form were presented and used to elucidate the partitioning of [234]Th between these phases. This
allowed the authors to demonstrate that this radioisotope is an ideal particle reactive tracer for studying the scavenging of thorium from surface waters.

In fact, a series of papers by Coale and Bruland in the mid-80's (Bruland and Coale, 1986; Coale and Bruland, 1985, 1987) were key for establishing the baseline for future [234]Th studies that would use [234]Th as a proxy for POC fluxes in the next era. The authors discovered that the [238]U-[234]Th disequilibrium is a direct tracer of the rates of sinking particles from the upper
ocean, which led to the acknowledgment of the relevance of the downward [234]Th flux. Previously, on a one-year time-series study, Tanaka et al. (1983) found large variations in the total [234]Th concentrations, with the minimum in [234]Th inventory coinciding with the early spring bloom, which they proposed was related to biological activity in the surface waters. It was within this context that Eppley et al. (1989) proposed that, if [234]Th is scavenged by biogenic particles, [234]Th could be used as a tracer of export production.

The "Coale and Bruland papers" were followed by many other investigators (e.g., Bacon and Rutgers van der Loeff 1989; Huh and Beasley 1987; Kershaw and Young 1988; Murray et al. 1989; Rutgers van der Loeff and Berger 1991; Schmidt et al. 1990; Wei and Murray 1991).

A series of milestones have been identified as the most remarkable of this era, which summaries as follows:
- 1969: initial measurement of total [234]Th and introduction of the co-precipitation of [234]Th with $Fe(OH)_3$ (Bhat et al.,
1969).
- 1972: beginning of GEOSECS program, which would prolong until 1978 and transformed the field of chemical oceanography as it existed at that time by exploiting new technologies available then (see synthesis of results by Broecker and Peng (1982).
- 1977: introduction of the concept of the "great particle conspiracy" by Turekian, (1977).
- 1979: initial analysis of particulate and dissolved [234]Th in the Narragansett Bay (Santschi et al., 1979).
- 1984:

- beginning of the Joint Global Ocean Flux Study (JGOFS) program, which grew out of the recommendations of a National Academy of Sciences workshop and would last until 2003;

- introduction of the $MnO_2$-impregnated filter cartridges technique (Mann et al., 1984), used with in-situ pumps.

- 1985: initial analysis of particulate and dissolved $^{234}$Th in open ocean and link between biological processes and $^{234}$Th deficits clearly demonstrated (Bruland and Coale, 1986; Coale and Bruland, 1985, 1987).

- 1987: 1 TSS project established in the framework of the French MOOSE project (Mediterranean Ocean Observing System for the Environment) and the JGOFS France program: *DYFAMED: Dynamics of Atmospheric Fluxes in the MEDiterranean sea* (43.42°N, 7.87°E).

- 1988: beginning of JGOFS field workwith the establishment of 2 TSS projects:
    - *BATS: The Bermuda Atlantic Time-series Station* in the Atlantic Ocean (31.50°N, 64.10°W), and
    - *HOT: Hawaii Ocean Time-series* in the Pacific Ocean at Station ALOHA: A Long-term Oligotrophic Habitat Assessment (22.45°N, 158°W).

- 1989: $^{234}$Th proposed to trace export production (Kolarova, 1975).

- 1990: 1 TSS project established in the frame of JGOFS program: *OSP- Station PAPA Ocean Weather Station "P"* in the Pacific Ocean (50.1°N, 144.9°W). Note that observations at Station Papa (former Ocean Station Peter and referred to as OSP or Ocean Station "P") started in 1949, although the larger surveys became a focus of activities in support of JGOFS during the 1990s (Freeland, 2007).

There exist two distinct radioanalytical methods for the $^{234}$Th extraction and purification from water samples that were initiated during this era: i) the co-precipitation of $^{234}$Th with $Fe(OH)_3$, proposed by Bhat et al., (1969), and ii) the scavenging of this nuclide onto $MnO_2$ cartridges, introduced by Mann et al., (1984). A thorough review of these techniques can be found in Rutgers van der Loeff et al., (2006). Briefly, for the $Fe(OH)_3$ technique, 20-30L seawater samples are treated and beta-counted. The addition of Fe carrier forms a precipitate that removes Th (and other elements) from solutions, so ion exchange purification procedures are required (at sea or quickly after return to shore) to separate $^{234}$Th from its parent and other potential beta emitters. For the $MnO_2$ cartridges technique, seawater is sequentially pumped through filters and two $MnO_2$ impregnated cartridges connected in series to scavenge dissolved Th isotopes. This technique was often used with large volume samples ($10^2$-$10^4$L), needed primarily for $^{228}$Th, $^{230}$Th and $^{232}$Th analyses, which required large amounts of ship time given the use of in-situ pumps for filtration, limiting the spatial coverage of the $^{234}$Th profiles. $MnO_2$ cartridges do not adsorb appreciable $^{238}$U, which is another advantage as ingrowth after sampling from $^{238}$U can be neglected. The large samples represented by a $MnO_2$ cartridge also allowed for direct gamma counting, thus eliminating the need for laborious radiochemical purification.

A wide variety of methods were used to sample the different $^{234}$Th phases within era 1. For the total phase, PVC tube and Van Dorn samplers (i.e., horizontal water bottle) were the prevalent equipment (Bhat et al., 1969; Matsumoto, 1975; McKee et al., 1984 or Minagawa and Tsunogai 1980 among others), although a few studies used pumping systems (Lee et al., 1991; Tsunogai et al., 1986), and bottles (Bacon and Rutgers van der Loeff, 1989). In the majority of studies measuring dissolved and

particulate [234]Th phases, the total [234]Th concetration was estimated as the sum of them (see e.g., Coale and Bruland 1985; Murray et al., 1989). Regarding particulate [234]Th analyses, most of the studies used filtration systems on volumes between 30 and 700L, a few studies used bottles (with Go-Flo model more typical than Niskin one), and one study introduced the use of pumps (Bacon and Rutgers van der Loeff, 1989) Note that the distinction between the dissolved and particulate phases is generally operational, with the term "dissolved" usually comprising all the phases passing through a pore size cut-off of 0.45 µm (see i.e., Bacon and Rutgers van der Loeff, 1989; Dominik et al., 1989; Rutgers van der Loeff and Berger, 1991; Santschi et al., 1979; Schmidt et al., 1990; Wei and Murray, 1991).

A total of 25 published works in refereed journals comprise the [234]Th studies of this era, compiled in a total of 24 datasets. Sampling was characterized by cruises mostly in the Pacific Ocean (see *Figure 2a* and *Figure 4a*), with a reduced number of locations sampled (maximum of 29 locations and averaged value of 6 locations per cruise) and a reduced number of data points per study (rarely over 100, averaged value of 60 samples per cruise). A total of 36 cruises were compiled from this era, with a total of 1493 [234]Th data points. Most surveys took place solely in the NH (29) between April and September (*Figure 5a*). Only 1 expedition surveyed the SH, but a total of 6 cruises crossed the Equator and sampled in both hemispheres.

Only 2 studies reported samplings that were part of a selected ocean program, which included the GEOSECS program (Krishnaswami et al., 1976) and the DYFAMED program (Schmidt et al., 1990) (see *Table 2* for a summary). Another study took place within the framework of the Joint Chinese-American Field Program (JCAFP) (McKee et al., 1984). A total of 9 studies reported sampling TSS (see e.g., Tanaka et al., 1983).

During this era, [234]Th studies were commonly focused on analyzing the parameters influencing the partitioning of a species between dissolved and particulate phases as a way to understand the mechanisms and rates for the scavenging of particle-reactive species. One way to quantify the scavenging of particle-reactive species is the use of distribution coefficients ($K_d$), which measure the partitioning of a species between dissolved and particulate phases (McKee et al., 1986). As the knowledge of scavenging increased, novel applications of [234]Th were developed, reducing the interest on the dissolved phase of [234]Th over time and driving changes in the [234]Th data type collected during field expeditions (see next sections).

The majority of the studies reported [238]U concentrations along with at least one [234]Th phase concentration (684 [238]U data points reported in this era), but none of them measured POC:[234]Th (*Figure 6a*), as the importance of this parameter was not evinced until era 2 (see Section 4.2). [238]U was either measured (a total of 6) or derived from salinity (a total of 10) mostly using the relationship from Ku et al., (1977). It is also worth mentioning that very few studies reported simultaneous sampling of complementary non-thorium measurements, such as sediment traps (a total of 6 studies, see e.g., Coale and Bruland 1987 and Tsunogai et al., 1986), or [210]Pb-[210]Po disequilibrium (5 studies, see Krishnaswami et al., 1976; Moran and Moore, 1989; Santschi et al., 1979, 1980; Tanaka et al., 1983). And none of the studies reported CHN (i.e., POC and/or PON) data.

Finally, in terms of modeling [234]Th data, more than half of the studies (56%, a total of 14) did it, and most of the cases used a two-box model, following Coale and Bruland (1985). Except from Tanaka et al., (1983) that applied both a SS and NSS model

to estimate the residence time of $^{234}$Th by accounting for the change in the inventory of $^{234}$Th with time, all the studies that provided information in this regard assumed SS conditions during sampling, which is not surprising since stations were not usually reoccupied during cruises for the collection of time-series data. The physical advection and diffusion (often referred to as *V* term) was neglected in most of the cases.

## 4.2. Era 2: "The Sea around us"

*1992-2000: Introduction of the empirical determination of POC export from $^{234}$Th profiles.*
*"When I think of the floor of the deep sea…I see always the steady, unremitting, downward drift of materials from above, flake upon flake, layer upon layer…the most stupendous "snowfall" the earth has ever seen."*
— *Rachel Carson, The Sea around us.*

The beginning of era 2 is marked by a new approach that was introduced to estimate POC fluxes from $^{234}$Th distributions from samples collected during the U.S. JGOFS North Atlantic Bloom Experiment (NABE) carried out in 1989. During this field experiment, the seminal work of Buesseler et al. (1992) found a clear relationship between the onset of the spring bloom, the subsequent drawdown of nutrients and $CO_2$, the net removal of $^{234}$Th and the flux of POC. Because of the conservative nature of $^{238}$U in the ocean, any measurable deficit of $^{234}$Th relative to its parent can be assumed to imply a significant removal by scavenging and particle sinking flux over a period of days to weeks before sampling, as the mean residence time of $^{234}$Th is dictated by its decay constant and removal rate by particles (Coale and Bruland, 1985). Buesseler et al. (1992) postulated the empirical determination of POC export fluxes (in µmol m$^{-2}$ d$^{-1}$) from $^{234}$Th-$^{238}$U oceanic disequilibrium following

$$POC_{flux}(z) = POC{:}^{234}Th(z) \cdot {}^{234}Th_{flux}(z) \ . \tag{1}$$

Where POC:$^{234}$Th is the ratio of POC to $^{234}$Th measured on sinking particles at the desired depth $z$ (in µmol dpm$^{-1}$), and $^{234}$Th$_{flux}$ is the $^{234}$Th downward flux measured at the same depth (in dpm m$^{-2}$ d$^{-1}$).

Eq. (1) is the base of the so-called $^{234}$Th method and the POC flux obtained is referred to as $^{234}$Th-derived POC flux. Note that the concept for this empirical method was introduced in a conference abstract in one much earlier study in the North Pacific (Tsunogai et al., 1976). Buesseler et al. (1992) proposed that the sinking flux of any element – such as carbon, phosphorus or nitrogen - could be derived from $^{234}$Th flux if the ratio of this element to $^{234}$Th on sinking particles is known. Element:$^{234}$Th ratios are directly determined from their in-situ measurement and vary with both depth and particle size (Buesseler et al., 2006).

$^{234}$Th flux can be calculated by evaluating the change in the corresponding total $^{234}$Th concentration with time and the contributions due to horizontal and vertical advection and diffusion processes. The simplest solution to estimate $^{234}$Th flux is SS conditions and ignoring advective and diffusive transport. These assumptions are the most commonly used (e.g., Le Moigne et al., 2013b are references therein) as generally only a single $^{234}$Th profile can be measured. The neglection of the physical term became inadequate with the expansion of $^{234}$Th research to coastal and more dynamic regimes (Savoye et al., 2006). In

the open ocean, the most relevant physical process is vertical upwelling, although there are also areas with downwelling, and it will typically result in underestimations of $^{234}$Th export if it is not included (Buesseler et al., 1995). Furthermore, Dunne and Murray (1999) developed a model to estimate advection and diffusion, concluding that incorporating advection to estimate carbon export might overestimate the sinking flux if diffusion is not considered. For the temporal variation of $^{234}$Th concentration, alternatively to the SS, a NSS model can be applied when temporal fluctuations in $^{234}$Th concentration can be assessed owing to repeated sampling, ideally over the course of 2 to 4 weeks (Resplandy et al., 2012) and only if the same water mass is sampled (Savoye et al., 2006) (i.e., the NSS approach is difficult to assess in dynamical settings).

The study by Buesseler and co-authors motivated an increase in oceanic $^{234}$Th measurements in the 1990's and we consider it one of the main milestones of this era along with the following ones:

- 1992: introduction of the empirical method for the POC flux estimate from $^{234}$Th concentrations proposed by Buesseler et al., 1992.

- 1992-1993: studies of the role of colloidal material (i.e., ~0.001 < colloids < 1 µm (Stumm, 1977)) in $^{234}$Th scavenging (Baskaran et al., 1992; Moran and Buesseler, 1993).

- 1995:
  - introduction of the use of the $^{210}$Pb-$^{210}$Po pair in a similar manner to the $^{238}$U-$^{234}$Th pair by Shimmield et al., (1995);
  - development of a regional 3-D $^{234}$Th flux model to estimate the physical components to the flux (i.e., the $V$ term) by Buesseler et al., (1995).

- 1996: beginning of the expansion of the $^{234}$Th-approach to particulate inorganic carbon fluxes (Bacon et al., 1996).

- 1998: 1 TSS project established in the frame of JGOFS program, operated by Taiwan: *SEATS: South East Asia Time-Series Station* (18°N, 116°E).

1999: introduction of the 20-L MnO$_2$ co-precipitation technique for $^{234}$Th by Rugters van der Loeff and Moore (1999). to allow particulate and dissolved concentrations to be analyzed from a single aliquot and to be measured by beta counting on board. In the earlier experiments of this era, bottle POC data were generally compared to $^{234}$Th-derived from individual cartridge filters or other particle collectors, such as in Buesseler et al., (1992). Several options of large volume in-situ pumps that allowed for measurement of POC and $^{234}$Th on the same filter emerged during this era: i) large volume filtration using Challenger Oceanic's submersible pumps was introduced by Shimmield et al., (1995), ii) MULVFS (multiple-unit large-volume filtration system) in-situ pumping system by Charette et al., (1999), with large volumes (2500-3500L) passing sequentially through a 1 µm cartridge pre-filter, and iii) large volume (400L) samples collected using in-situ battery-operated pump deployed on the CTD/Rosette frame (Charette and Buesseler, 2000). Ratios were normally determined using small pored sized filters, either >0.4-0.5 µm (-An Huh and Prahl, 1995; Cochran et al., 1995; Laodong et al., 1997; Santschi et al., 1999), or >1 µm (e.g., Benitez-Nelson et al., 2000; Gustafsson et al., 1997; Rutgers van der Loeff et al., 1997) typically made of glass or quartz fiber. A total of 10 studies sampled 2 size fractions determined by filtering through 53 µm (Bacon et al., 1996; Buesseler et al., 1998,

1995; Charette and Moran, 1999; Moran et al., 1997; Moran and Buesseler, 1992; Moran and Smith, 2000; Murray et al., 1996) or 70 µm Nitex screen (Charette and Buesseler, 2000; Cochran et al., 2000), but only 1 of them reported POC:$^{234}$Th on both small and large particles. In total, only 3 studies reported POC:$^{234}$Th measured with filtration methods (Buesseler et al., 1992; Charette et al., 1999; Murray et al., 1996) Additionally,POC:$^{234}$Th ratios in sinking particles were collected in sediment traps

by a total of 6 studies, including modern designs such as neutrally buoyant sediment trap (NBST) (see e.g., Buesseler et al., 2000) and surface-tethered particle interceptor trap (PIT) (see e.g., Buesseler et al., 1994).

A total 43 data sets comprise this era, extracted from a total of 48 publications. Sampling was focused on the Atlantic Ocean (see *Figure 2a and Figure 4b*). Surveys in the Pacific Ocean were limited while increased sampling in the Southern Ocean, and field expeditions to the Arctic were conducted. Both the number of expeditions and the locations sampled increased.

Despite its short duration (less than a decade), a total of 70 cruises were reported within this era, indicative of the dedicated programs and experiments that marked this era, resulting in 8739 and 1133 data points for $^{234}$Th and $^{238}$U respectively. Similar to the first era, cruises mainly took place exclusively in the NH (52 cruises), although both expeditions solely to the SH and to both hemispheres increased, with a total of 7 and 11 cruises respectively. Samplings mainly took place within the first semester of the year (*Figure 5b*).

As previously mentioned, this period of the $^{234}$Th history partially overlapped with the golden years of the JGOFS program, and therefore, many of the studies published during era 2 reported $^{234}$Th measurements collected during field expeditions that took place in the frame of this international program. A summary of these activities is provided in *Table 2*. In the case of U.S. JGOFS, this included $^{234}$Th measurements at the BATS TSS (see e.g., Buesseler et al., 1994, 2000) and during several process studies in well-defined areas at strategic oceanic locations: 1) NABE (North Atlantic Bloom Experiment,

http://usjgofs.whoi.edu/research/nabe.html) that was one of the first major activities of JGOFS with 3 cruises along longitude 20°W in 1989. It was published 3 years later by Buesseler et al., (1992); 2) EqPac (Equatorial Pacific, http://usjgofs.whoi.edu/research/eqpac.html) process study that was conducted along 140°W during year 1992 and included a total of 4 cruises. It was published between 1994 and 1997 by Bacon et al., (1996); Buesseler et al., (1995); Dunne et al., (1997); Murray et al., (1996) and extended in the next era by Hernes et al., (2001) and Murray et al., (2005);3) the Southern

Ocean expedition during October/November 1992 published by Rutgers van der Loeff et al., (1997) and expanded in era 3 by Friedrich and Rutgers van der Loeff (2002); 4) Arabian Sea (http://usjgofs.whoi.edu/research/arabian.html), beginning in October 1994 and ending in January 1996 and was reported a few years later by Buesseler et al., (1998); 5) AESOPS (Antarctic Environment and Southern Ocean Process Study, http://usjgofs.whoi.edu/research/aesops.html), which carried out field work between August 1996 and April 1998 and was published within this era by Cochran et al., (2000) and extended in the next era

by Buesseler et al., (2001b). Additionally, studies also reported data from other JGOFS expeditions from the: 6) Indian program, which completed three major sampling expeditions for $^{234}$Th to the eastern and central Arabian Sea in April-May 1994, February-March 1995 and July-August 1995. It was reported by Sarin et al., (1996) and extended in PANGAEA in 2013 (https://doi.org/10.1594/PANGAEA.807500); 7) Canadian program in the northeast Pacific Ocean which had two phases,

from 1992 -1994 and from 1995-1997, although [234]Th was only sampled during the second phase. Data were published by
(Charette et al., 1999); and 8) France DYFAMED program, sampled in 1987 but published in this era by Schmidt et al., (1992)
and extended in the next era (see Schmidt et al., 2002b, 2002a). Finally, field work also included 8) the Southern Ocean Iron
RElease Experiment (SOIREE, https://www.bco-dmo.org/project/2051), which was the first in-situ iron fertilization
experiment performed in the polar waters of the Southern Ocean. It took place in February 1999 south of the Polar Front in the
Australian-Pacific sector of the Southern Ocean and was reported in Charette and Buesseler (2000). Data from this iron
enrichment experiment was compiled along with others in a common open-access database during Iron Synthesis program
(FeSynth, https://www.bco-dmo.org/program/2017) started in 2007.

In addition to JGOFS, cruises from another major initiative that included [234]Th sampling was the Ocean Margins Program
(OMEX, http://po.msrc.sunysb.edu/omp/), a large field-based study, with extensive physical, chemical, biological and
geological measurements, carried out on northwest European shelf break that ran in two phases, from 1993-1996 and from
1997-2000. [234]Th data were made available by Hall et al., (2000).

The number of data points measured per cruise significantly increased within this era, with an average of 125 [234]Th data points
reported per cruise in comparison to the 41 data points averaged of era 1.  Such an increase was significantly marked for the
dissolved and particulate phases, whose measurements increased more than a 5-fold relative to those from era 1 (see *Figure
6b*). Additionally, measurements to determine POC and PON to [234]Th ratios became routine, therefore allowing the estimate
of POC fluxes (see *Figure 4b*). However, only half of the studies reported [238]U concentrations along with [234]Th data, in their
majority using a variety of [238]U-salinity relationships, with Chen et al., (1986) as the prevalent one. An overall 69% of the
studies (33 out of 48 that comprises this era) modeled [234]Th data. Time-series data were collected more often than during the
first era, likely with the purpose of following the NSS approach (Buesseler et al., 1992, 1995). Nonetheless, the majority of
the studies (70%) that modeled [234]Th data assumed SS conditions (see e.g., Bacon and Rutgers van der Loeff 1989; Baskaran
et al., 1992; Gustafsson et al., 1997b). A total of 10 studies applied the NSS model, with 7 of them combining it with the SS
model (see e.g. Buesseler et al., 1994; Cochran et al., 2000; Kersten et al., 1998 for the combined use of both approaches). In
terms of additional data, 1 study introduced underway [234]Th sampling (Hall et al., 2000), a total of 11 studies reported sampling
with sediment traps (see e.g., Cochran et al., 2000; Murray et al., 1996; Smoak et al., 1999), 10 studies sampled for [210]Pb-[210]Po
disequilibrium (e.g., Moran and Moore 1989; Santschi et al., 1999; Wei and Murray 1992), and 13 studies reported CHN data
(see e.g., Niven et al., 1995; Rutgers van der Loeff et al., 1997; Santschi et al., 1999 among others).

### 4.3. Era 3: "In the heart of the sea"

*2001-2009: Improvements of the [234]Th technique.*
*"At sea, things appear different."*
*— Nathaniel Philbrick, In the heart of the Sea.*

The final boost to the widespread and increasing use of [234]Th as a particle flux tracer was motivated by the works of Buesseler et al., (2001) and Benitez-Nelson et al., (2001), which dramatically increased the number of [234]Th measurements and marked the third era with the introduction of the small-volume (2-4L) technique. This procedure modified the 20 L-method developed by Rugters van der Loeff and Moore (1999) in era 2 and uses the lowest sample volumes of all known [234]Th methods to date. It not only allowed immediate on-board beta-counting of [234]Th concentration and avoided some tedious folding sessions, but

also enhanced both spatial and temporal resolution of particle export.

The revolution that this novel technique brought with it was substantial. The small-volume technique is essential to capture the particle dynamics and export flux variations on scales that could be better related to local biogeochemical conditions. The main advantage is the convenience of handling small volumes and more rapid processing times. Multiple sampling casts are often required for a 20-L sample profiles whereas a 4-L technique usually allows simultaneous sampling of [234]Th with other

parameters on a single cast (e.g., nutrients, phytoplankton biomass, etc.). As a result, this method can be easily applied at sea using samples obtained through CTD rosette water samplers that are available on most research vessels.  For that reason, we have chosen the introduction of the small-volume technique as a shifting milestone in the [234]Th timeline.

 Further improvements of this technique within this era were carried out some years later by Pike et al. (2005) and Cai et al., (2006) works, which are major milestones of this era, whose timeline summaries as follows:

• 2001: introduction of the small-volume technique by Buesseler et al., (2001) and Benitez-Nelson et al., (2001).
   • 2002:
      - 1 long-term multidisciplinary and moored observatory established in the northwest Atlantic coordinated by the National Oceanographic Center of Southampton (UK): *PAP site: Porcupine Abyssal Plain* (49° 13.5°E).
      - expansion of the [234]Th-approach to Biogenic Silica fluxes (Friedrich and Rutgers van der Loeff, 2002).
• 2005: introduction of the use of yield tracers - double spike technique - for the sake of maintaining the reproducibility (Pike et al., 2005), The advantage of this modified approach is a precise knowledge of [234]Th recovery, leading to an enhancement in data quality of [234]Th, for what it is standardly used nowadays and recommended by GEOTRACES protocol (Maiti et al., 2012).
   • 2006: reduction of the filtration time and introduction of the alpha spectrometric measurement of [230]Th recovery by
means of using a combination of water bath heating and a reduction in reagent quantities (Cai et al., 2006). This also modified the typical ion-exchange chemistry to allow for the alpha spectrometric measurement of [230]Th recovery.

Moreover, a technological innovation was introduced during era 3 with the development of 2 in-situ pumping systems for high volume filtration for [234]Th concentrations and POC:[234]Th ratios in sinking particles widely used from this era on: 1) McLane large-volume water transfer system (WTS-LV) pumps by McLane Laboratories (Falmouth, MA), and 2) SAPS (Stand-Alone

Pump System) by Challenger Oceanic (Surrey, U.K.). Moreover, one study reported the use a novel split flow-thin cell fractionation (SPLITT) (see Gustafsson et al., 2006).

This has been the golden age of the $^{234}$Th so far in terms of publications released by year (almost 10, versus 1 and 6 in eras 1 and 2), with a total of 159 cruises reported in a period of 9 years. A total of 19676 and 4567 data points for $^{234}$Th and $^{238}$U respectively (see *Figure 4c* and *Figure 6c*) were compiled in 76 datasets, extracted from 87 studies (81 publications in referred journals, 1 PhD thesis, and 5 repositories). Data reported in the context of selected ocean programs continued (see *Table 2* for more details). This included JGOFS – through the TSS of HOT (Benitez-Nelson et al., 2001a), BATS (Sweeney et al., 2003), and DYFAMED (e.g., Szlosek et al., 2009); and OMEX (Schmidt et al., 2002a). Other new studies, such as the Carbon Retention In A Colored Ocean (CARIACO, http://www.imars.usf.edu/cariaco) time-series program operative between 1995 and 2017, published by Smoak et al., (2004), were also initiated. Additionally, novel JGOFS initiatives took place: 1) JGOFS-France MEDATLANTE (http://www.obs-vlfr.fr/cd_rom_dmtt/other_main.htm) in the Gibraltar Strait and the N.E. Atlantic Ocean. Data were made available by (Schmidt et al., 2009b); 2) JGOFS-France ANTArctic RESearch program (ANTARES, http://www.obs-vlfr.fr/cd_rom_dmtt/an_main.htm). Data published by Coppola et al., (2005); 3) JGOFS-France DYNAPROC (http://www.obs-vlfr.fr/cd_rom_dmtt/other_main.htm) cruise, reported by (Schmidt et al., 2002c, 2009b) and the 4) JGOFS-Japan North Pacific Process Study carried out in at KNOT station (44°N 155°E, https://www.nodc.noaa.gov/archive/arc0013/0001873/1.1/data/1-data/general/res_outline/KNOT.html) by Kawakami and Honda, (2007). Data from great number of projects and experiments were published during this era (see *Table S3* for a chronological summary).

The publications included in this era report data covering the entire ocean and with increased number of data points in all regions, except the Indian Ocean (see *Figure 2a and Figure 4c*). Expeditions to the SH increased, with 18 reported, yet were far below those in the NH, where a total of 134 cruises were undertaken (see *Figure 5c)*. Additionally, a total of 7 surveys sampled in both hemispheres. Stations included the long-term observatories previously sampled for $^{234}$Th along with novel ones, such as the PAP site (Turnewitsch and Springer, 2001). Fieldwork took place throughout the entire year, with a remarkable number of cruises in March. Both the overall number of data points and the data points measured per year increased due to the large number of field expeditions reported during this era and the spread use of the small volume technique. An average of 2183 $^{234}$Th data points was reported per year in comparison to an average of 60 and 971 data points during eras 1 and 2 respectively. As for the $^{234}$Th sample types in this era, total number of measurements significantly increased for all phases (2-fold for the dissolved and particulate, and 4-fold for the total), and for POC:$^{234}$Th ratios, and remain constant for PON:$^{234}$Th (see *Figure 6c*).

A great increase in the number of data points also was detected for $^{238}$U activities, with an increase of 4-fold in the data points. Once again, salinity-derived was the most extended approach, with almost all the studies using Chen et al., (1986).

A total of 66 studies out of the 87 that comprise this era (i.e., 76%) modeled $^{234}$Th data: 42 of them applied the SS model (see e.g., Amiel et al., 2002; Guo et al., 2002; Speicher et al., 2006), 2 applied the NSS one (Kawakami et al., 2004 and Smoak et al., 2004), and 22 studies applied both approaches (e.g., Amiel and Cochran, 2008; Gustafsson et al., 2004; Kim and Church, 2001).

In terms of sampling strategy, 2 studies reported $^{234}$Th underway sampling (Rutgers van der Loeff, 2007a and Schmidt et al., 2002a). 30 studies reported the use of sediment traps, with 24 of them measuring POC:$^{234}$Th ratios (Charette et al., 2001; Schmidt et al., 2002b; Sweeney et al., 2003 among others). Around a third of the studies reported measurements CHN data (see e.g., Lepore et al., 2009; Santschi et al., 2003) , while a very reduced number of the studies, around 10%, analyzed $^{210}$Po (see Porcelli et al., 2001 and Somayajulu et al., 2002 amonth others).

### 675 **4.4. Era 4: "Twenty thousand leagues under the sea"**

*2010-present: GEOTRACES program and a new way to study the ocean.*
*"The sea is everything. It covers seven tenths of the terrestrial globe. Its breath is pure and healthy. It is an immense desert, where man is never lonely, for he feels life stirring on all sides. The sea is only the embodiment of a supernatural and wonderful existence."*
― *Jules Verne, Twenty thousand leagues under the sea.*

The beginning of this era is identified with the first publication of $^{234}$Th data from a GEOTRACES cruise, reported by Cai et al., 2010. GEOTRACES is an international study of the marine biogeochemical cycles of trace elements and their isotopes which changed the way to explore the oceans by combining ocean sections, process studies, data syntheses, and modeling. Its launching marked the beginning of internationally dedicated large-scale collaborative projects characterized by long field 685 surveys with very high spatial resolution and many different parameters measured simultaneously. For this reason, the initial $^{234}$Th-related GEOTRACES publication was chosen as the milestone that starts the - so far - final era in the $^{234}$Th timeline.

The history of the GEOTRACES dates back to 2000, when the initial idea started with group discussions at international meetings inspired by both the successes and the limitations of GEOSECS and JGOFS. After some years in the planning and enabling phase, the GEOTRACES Science Plan was published in 2006. During 2007-2009 the first GEOTRACES cruises 690 took place, including those that were part of the International Polar Year (IPY, https://www.geotraces.org/geotraces-in-the-international-polar-year/) devoted to detecting and understanding a suite of trace elements and isotopes in the Arctic and Antarctic marine environments, and the intercalibration cruises (to the Atlantic and Pacific oceans in 2008 and 2009 respectively). A GEOTRACES transect in the Drake Passage was also carried out in 2008 (GIPY05, Germany, ZERO and DRAKE) in the framework of IPY. Finally, the GEOTRACES program formally launched its seagoing effort in January 2010 695 and was fully announced to the scientific community at Ocean Sciences Meeting that year in Portland (Oregon, USA). Numerous cruises have taken place since then, with many of them including $^{234}$Th sampling. Many of these results have been reported in the Intermediate Data Products, the first one released in 2014 (IDP2014 by Mawji et al., (2015)), the second one in 2017 (IDP2017, by Schlitzer et al., (2018)), and the third and most recent one in 2021 (IDP2021, by GEOTRACES Intermediate Data Product Group, (2021)).

The GEOTRACES program brought a new philosophy that, in the context of $^{234}$Th, drove a shift from mostly deriving POC fluxes only to include trace metal fluxes (see e.g., Black et al., 2018) and the implementation of standards and intercalibration

initiatives to establish procedures and protocols for sampling at sea to ensure that samples are collected, handled, and stored without contamination or other sources of bias (see e.g., the versions of the "cookbook – sampling and sample-handling protocols or GEOTRACES Cruises" by GEOTRACES Standards and Intercalibration Committee (Cutter et al., 2014, 2017)).

This era includes the development of new technologies to accelerate the collection and analysis of samples, the intercalibration of those technologies to ensure internal consistency among the participating labs, the development of a data management system to facilitate access to the results to the entire oceanographic community, and a broad collaborative effort to model, synthesize, and interpret the results. The most remarkable milestones for this era are:

- 2010:
    - initial publication of $^{234}$Th from a GEOTRACES cruise by Cai et al., 2010, which reported data from ARK-XXII/2 expedition to the Arctic Ocean in 2007 carried out in the context of the IPY-GEOTRACES program (GIPY11, Germany);
    - intercalibration initiative by Cutter et al., (2010) to ensure that $^{234}$Th results produced by different groups were comparable and internally consistent, using deep waters or stored samples as standards where $^{234}$Th and $^{238}$U are known to be in secular equilibrium.
- 2012: intercalibration initiative by Maiti et al., (2012), which carried out an intercomparison of $^{234}$Th measurements in both water and particulate samples between 15 laboratories worldwide.
- 2014: release of the 1$^{st}$ GEOTRACES IDP (Mawji et al., 2014)
- 2017: release of the 2$^{nd}$ GEOTRACES IDP (Schlitzer et al., 2018a).
- 2021:
    - improvement of the small-volume technique has been recently carried out by Clevenger et al., (2021), which introduces a revised protocol that decreases sample volumes to 2 L, shortens wait times between steps, and simplifies the chemical recovery process, expanding the ability to more rapidly and safely apply the $^{234}$Th method;
    - release of the 3$^{rd}$ GEOTRACES IDP (GEOTRACES Intermediate Data Product Group, 2021)

A total of 80 data sets were compiled from this era, including 9 previously unpublished ones, extracted from a total of 82 studies (65 publications in referred journals, 4 PhD thesis, and 4 data repositories). A total of 114 cruises and 26755 and 11872 data points for $^{234}$Th and $^{238}$U respectively were compiled, mostly distributed between the Atlantic and Pacific oceans and, to a lesser extent, the Southern Ocean (see *Figure 2a*). This era is the second more productive, after era 3, in terms of publications released per year (7 per year). It is also the most intense in terms of sampling, with an average of 235 $^{234}$Th measurements reported by cruise and over 2400 $^{234}$Th data points reported by year. Once again, the NH dominated the surveys (79 cruises sampled exclusively above the Equator), particularly numerous in early spring and fall (see *Figure S1*). Nonetheless, this era reports the highest number of cruises to the SH, with a total of 27 sampling exclusively below the Equator, and an additional 8 cruises sampling in both hemispheres.

More than half of the expeditions of this era reported data from cruises that took place in the framework of major ocean programs (see *Table 2*) and some minor projects, such as the i) Arctic Ocean 2001 (AO-01) expedition, reported by Gustafsson and Andersson, (2012); ii) 2[nd] and 3[rd] Chinese National Arctic Research Expedition (CHINARE-2 and 3), carried out in 2003 and 2008 respectively and published by Yu et al., (2010), (2012); iii) Indo–German iron fertilization experiment LOHAFEX (https://epic.awi.de/id/eprint/21440/) in 2009, published 4 years later by Martin et al., (2013); iv) second KErguelen Ocean and Plateau compared Study expedition (KEOPS2) of 2011, published by Planchon et al., (2015); or v) Fate of the northwestern Mediterranean open sea spring bloom (FAMOSO, http://digital.csic.es/handle/10261/94572) project, which took place between March and September 2009 and was personally communicated by V. Puigcorbe to be included in this compilation. Furthermore, a 46% of the data sets included $^{234}$Th data from an established time-series location (see *Figure 1c*).

An interesting characteristic of this period is that a relatively smaller number of cruises produced a greater amount of data in comparison to previous eras. The most relevant peaks are found in years 2013 and 2015, which coincides with the publication of several GEOTRACES transects (see Section 3). Dissolved $^{234}$Th measurements reduced drastically (4-fold), while the rest of measurement types increased to some extent, most notably for total $^{234}$Th and $^{238}$U (see *Figure 6d,* note the change in the scale of this plot along the eras). Once again, $^{238}$U derived from salinity was the most commonly used approach with almost all the studies using the U-salinity relationship of Chen et al., (1986) or Owens et al., (2011).

In terms of modeling $^{234}$Th data, a total of 64 out of the 82 studies (78%) modeled $^{234}$Th concentrations and specified whether a SS, a NSS, or both models were used. From them, 50 (78%) studies applied the SS approach (see e.g., Alkalay et al., 2020; Cai et al., 2010; Ceballos-Romero et al., 2016), just 1 (2%) study applied the NSS one (see Kawakami et al., 2015), and a total 13 (20%) studies compared both approaches (see Buesseler et al., 2020a; Kim et al., 2011; Lemaitre et al., 2018 among others). The number of studies carrying out $^{234}$Th underway sampling increased to 7 (see e.g., Black et al., 2018; Estapa et al., 2015; Martin et al., 2013). More than 32% of the studies reported POC:$^{234}$Th ratios collected with sediment traps (Haskell et al., 2013; Kawakami et al., 2010; Maiti et al., 2016 among others). Around a third of the studies (34%) reported CHN data (see e.g., Rosengard et al., 2015), while a very reduced number (~9%) analyzed $^{210}$Po (such as Alkalay et al., 2020; Ceballos-Romero et al., 2016; or Wei et al., 2011).

### 4.5. Launching of dedicated large-scale collaborative projects: the beginning of era 5?

*"Dawn begins pushing back the covers of night and the sun rolls out of bed."*

— *Carl Safina, Song for the Blue Ocean.*

Significant milestones within recent years are marked by the launching of an unprecedented number of large-scale collaborative projects including $^{234}$Th measurements: e.g., 1) PICCOLO: Processes Influencing Carbon Cycling: Observations of the Lower limb of the Antarctic Overturning (UK, 2016-2020, https://roses.ac.uk/piccolo/), a project devoted to quantify the crucial processes that determine carbon cycling in the lower limb of the Southern Ocean overturning circulation through observations and models. 2) COMICS: Controls over Ocean Mesopelagic Interior Carbon (UK, 2017-2021,

https://www.comics.ac.uk/), a research project that aims to quantify the flow of carbon in the ocean's twilight zone (part of the ocean between 100 and 1000 m below the sea surface) in order to more accurately model global climate change; 3) EXPORTS: EXport Processes in the Ocean from Remote Sensing project (US, 2017-2022, https://oceanexports.org/), a large-scale NASA-led field campaign that will provide critical information for quantifying the export and fate of upper ocean net primary production using satellite observations and state of the art ocean technologies; 4) CUSTARD: Carbon Uptake and Seasonal Traits in Antarctic Remineralisation Depth project (UK, 2018-2022, https://roses.ac.uk/custard/), an initiative which will examine how seasonal changes in food availability for phytoplankton influences how long carbon is trapped in the ocean in the Southern Ocean; 5) MOBYDICK: Marine Ecosystem Biodiversity and Dynamics of Carbon around Kerguelen: an integrated view (France, 2017-2022, https://mobydick.mio.osupytheas.fr/), a project that combines investigation of the BCP and the end-to-end structure of the pelagic food web, 6) OTZ: Ocean Twilight Zone (US, 2018-2024, https://twilightzone.whoi.edu/), a project devoted to the study of this region of the ocean, with particular emphasis on development of new technologies for scientific exploration of the biomass and biodiversity, animal lives and behavior, food webs interactions, and flow of carbon through the twilight zone, and 7) SOLACE: Southern Ocean Large Area Carbon Export (Australia, December 2020-January 2021, https://solace2020.net/), a 6-weeks voyage aimed at developing an approach to quantify the changing effectiveness of $CO_2$ sequestration by the BCP using remote-sensing of the ocean surface by satellites and it's interior by autonomous vehicles (specially BIO-ARGO profiling floats) and MOBYDICK project. Additionally, there is another funded future project that include $^{234}$Th in its methodology: the APERO program: Assessing marine biogenic matter Production, Export and Remineralisation: from the surface to the dark Ocean (France, 2022-2026, with a cruise planned for 2023 in the western North Atlantic, https://www.polemermediterranee.com/Activites-Projets/Ressources-biologiques-marines/APERO). The Joint Exploration of the Twilight Zone Ocean Network (JETZON, https://www.jetzon.org/) is a UN Ocean Decade Program that was created in 2020 to serve as a focal point for Twilight Zone studies. It includes $^{234}$Th measurements as one of the key methodologies and aims acting as an international coordinating umbrella of projects such as COMICS, EXPORTS, CUSTARD, OTZ, SOLACE, APERO or PICCOLO, among many others, providing a link between researchers working on these projects through JETZON, so that cruises, data, results, and conclusions are shared.

Note that so far, only data from the first expeditions of the EXPORTS project have been published (Buesseler et al., 2020a) and are available in the compilation (see peak in *Figure 2b* and *Figure 2d*). The benefits to be derived from the success of these endeavors and the inclusion of the remaining data in the compilation could trigger the beginning of a new era for the $^{234}$Th technique (i.e., era 5). Whether or not the publication of the $^{234}$Th measurements collected during these dedicated large-scale collaborative projects would mark the beginning of era 5 is yet to be known.

**5.    Steps towards improving the global data set**

We present here two perspectives that will improve the $^{234}$Th global data set and broaden its applicability: detect ocean regions with low $^{234}$Th measurements and identify which parameters could be useful to include in future versions of the compilation for a wider application of $^{234}$Th data.

## 5.1. Recommendations on $^{234}$Th sampling

During the last few decades, considerable progress has been made towards unraveling the behavior of the BCP and understanding the factors influencing the carbon dynamics and the ocean carbon cycle, e.g., primary production, aggregation, ballasting, and the activities of zooplankton and bacteria (see reviews by e.g., Buesseler and Boyd 2009; De La Rocha and Passow 2007; Sanders et al. 2014; Turner 2002). Capturing the spatiotemporal variability of $^{234}$Th concentrations could play a key role on our precise quantification of carbon uptake, storage rates, and subsequent ecosystem impacts. Strategies up to now have included time-series of process studies in key ocean regions, global surveys of carbon parameters, TSS and models and databases built from field observations. However, there are gaps in the ship-based $^{234}$Th sampling.

In terms of spatial distribution, the SH remains clearly undersampled. Of the total of 5134 locations compiled, 3351 belong to the NH (which accounts for a total of 65% of the sampling locations) and only 1749 to the SH (35%), with the 34 remaining ones corresponding to the Equator (<1%). This is the result of the majority of expeditions taking place in the NH, 78% solely in the NH and 8% as part of surveys in both hemispheres.

*Figure 1a* highlights an especially important gap in the South Pacific region and some additional notable gaps in the data set in the Benguela system (although this was included in COMICS project and data will be available in the near future), the Mauritanian upwelling or the southern Indian Ocean. More attention should be paid to these regions when planning future expeditions for $^{234}$Th sampling. Moreover, *Figure 1c* shows numerous established time-series stations that have never been sampled for $^{234}$Th and we recommend visiting in next expeditions.

In terms of temporal distribution, research cruises are frequently conducted between January and October, although data are skewed to early spring and summer in both hemispheres (see peak in March and October in *Figure S1b*). This is especially important for the SH, in the Southern Ocean, where tough wintertime conditions complicate shipboard operations and research cruises are therefore frequently skewed to calmer spring and summer months. This means that sampling in Southern Ocean mostly occurs during bloom conditions, which affects the reliability of the SS approach to quantify carbon export from $^{234}$Th data (e.g., Buesseler et al., 1995; Ceballos-Romero et al., 2018; Maiti et al., 2008; Resplandy et al., 2012; Savoye et al., 2006). Therefore, results in the Southern Ocean are biased, when compared to the results in other areas, as they correspond mainly to POC fluxes in bloom conditions, whereas in the rest of the oceans, pre- and post-bloom conditions are also included in the global analyses.

Given that the sampling period is not always up to choose, especially when surveying rough seas, in next section we recommend a series of ancillary parameters to $^{234}$Th data, often missed in the interpretation of $^{234}$Th results that will be useful to accurately interpret the results in those specific situations. Furthermore, these complementary parameters will ensure accuracy when i) interpreting $^{234}$Th deficits, ii) temporally contextualizing $^{234}$Th-derived export results, and iii) extrapolating export results to longer time scales for annual patterns and global estimates of the BCP.

## 5.2. Recommendations on $^{234}$Th ancillary parameters

It is worth mentioning that there is a vast amount of data that can be useful to report in a [234]Th compilation due to the wide applicability that has been shown for this technique along the years. However, for the sake of feasibility, we had to set boundaries and we left some parameters out of this first version of the compilation. Nevertheless, we would like to emphasize the dynamic character of this compilation, which has been conceived as an enduring compilation. Thus, the authors will keep the data set "alive" by means of i) identifying new [234]Th data sets to be included, ii) adding new data provided by direct contact to us by [234]Th data measurers, iii) amending existing data sets if errors are detected or reported, and iv) by extending the amount of parameters included in the data sets.

Following the dynamic spirit behind this effort, we will periodically update the compilation in PANGAEA. As so, we acknowledge ways to improve the compilation in future versions of it. Both new data and ancillary parameters to the [234]Th measurements will be included in future versions of the compilation, including new useful parameters arisen from novel research and also following suggestions by researchers. We recommend [234]Th data users to consider the following parameters as essential complement to [234]Th data and therefore report them when publishing [234]Th studies.

A necessary complementary data to be included in future version of the compilation for the application of the [234]Th method, are the many auxiliary sensors included on CTDs to measure parameters such as fluorescence-chlorophyll-a and PAR (photosynthetically active radiation) These parameters allow to track the variable depth of sinking POC production and choose the integration depth to estimate [234]Th fluxes by either using the euphotic zone (see e.g., Giuliani et al., 2007) , the PAR depth (e.g., Rosengard et al., 2015) or the primary production zone (e.g., Owens et al., 2015). The integration depth used to estimate POC fluxes has been shown to impact the quantification of the 'strength' (i.e., magnitude) of the BCP (Buesseler et al., 2009, 2020b). Therefore, fluorescence and/or PAR data combined to [234]Th data are expected to resolve the mechanisms that control the transfer of carbon to depth by means of the BCP by normalizing POC flux to the depth where it is produced and below which only particle flux attenuation occurs so progress can be made on untangling the relationships between POC flux and primary production, food web controls, and variable sinking and degradation rates among others (see Section 6 for new studies proposed in this direction). However, most of these auxiliary sensors have only been recently incorporated to the CTD (Jijesh et al., 2017) . On the contrary, temperature and salinity data have been a focal point of oceanography since the late 19[th] century and fully overlap with the 50 years of lifespan of the [234]Th technique. Is for that reason that the temperature and salinity were the only CTD parameters included in this initial version of the compilation. We acknowledge fluorescence and PAR data are crucial parameters for an accurate use of [234]Th disequilibrium to evaluate POC export and will therefore be added in future versions of the data set. We recommend [234]Th data users to consider these data as essential ancillary parameters to [234]Th data and therefore report them when publishing [234]Th studies.

On the other hand, regarding gaps in parameters overall, only a 36% of the studies compiled provided information regarding bloom conditions and bloom stage during sampling, which prevents us from assessing the reliability of the assumption of SS and NSS conditions to calculate [234]Th fluxes. Most of these studies (72%) followed the SS approach, while 24% of them combined it with the NSS one, and only 4% applied only the NSS model. The reliability of the SS/NSS approach applied to

[234]Th deficits to estimate the POC flux depends on the sampling time in relation to the bloom dynamics. The available information on the bloom timing, and/or peak flux date, and/or duration detailed by station is a key requirement for a quantitate evaluation of the overall accuracy of current [234]Th -SS-derived POC flux estimates (see e.g., Ceballos-Romero et al., 2016, 2018). Therefore, we recommend providing as much of this information as possible when reporting future [234]Th data. Additionally, for future versions of the compilation, in order to include in the metadata, the bloom stage at the sampling moment following a standardized criterion we carry out a revision of up-to-date literature (Cole et al., 2012; Rumyantseva et al., 2019; Thomalla et al., 2015) to select a quantitative and robust method, easy to apply, of determination. This way, for further datasets, the bloom stage will be reported when available, previous discussion with authors. A standardized criterion to assign a bloom stage to new datasets in the compilation, based on sampling date and ocean region will be established.

## 6.    Towards a better understanding of oceanic carbon uptake: data use perspectives

The efficiency of the BCP at transporting carbon to the deep ocean and regulating atmospheric $CO_2$ levels is affected even by small perturbations to oceanic ecosystems such as seawater chemistry or nutrients distribution (Kwon et al., 2009). Ongoing changes in temperature, oxygen concentration and stratification of the water column derived from anthropogenic warming are predicted to alter export and transfer efficiency by different means. Specifically, a reduction in carbon sequestration, and therefore an increase in oceanic emissions of $CO_2$ is expected (Kwon et al., 2009; Le Quéré et al., 2009). More observational data and mechanistic models to describe the particle flux and its attenuation are required in order to unravel and properly quantify the current role of the BCP and provide a baseline for future estimates of oceanic $CO_2$ uptake.

There are many analyses of BCP processes that [234]Th data could be used for. *Figure 7a* shows a global summary of [234]Th/[238]U ratios at $100 \pm 10$ m of the water column water in the data set (a total of 17370 data points). Ratios have been categorized as i) "deficit" for those values were [234]Th concentrations are lower than [238]U ones within a 10% of uncertainty (ratio < 0.9), ii) "equilibrium", for those values that have similar concentration of [234]Th and [238]U (0.9 < ratio < 1.1), and iii) "excess" for those the values where [234]Th concentrations excess [238]U ones (i.e., ratios > 1.1). This figure can be interpreted as the distribution of probability of [234]Th reaching equilibrium (or not) with its parent at 100 m. Most of the data points fall within the equilibrium (53% of the data points) and deficit (39%) categories, while "excess" data points accounted only for an 8%. A closer look at the deficit data indicates that most of the samples with ratios <0.9 were very close to the equilibrium value. Therefore, most of the data point compiled reach equilibrium around 100 m depth (see peak with 250 samples with [234]Th/[238]U ratios = 1). This is important because 100 m is the reference depth that many past studies used to calculate POC fluxes (see Figure 1 in Le Moigne et al., (2013b)). It is important to note that the depth where [234]Th equilibrium is reached do not necessarily agree with the euphotic zone. Equilibrium is usually re-established below the euphotic zone, at depths ranging from 50 m (e.g., Thomalla et al., 2006) to 500 m depth (e.g., Owens et al., 2015), for reason not yet clear (see discussion by Rosengard et al., (2015)). Therefore, most recent literature recommends using depths relative to the euphotic zone as reference depth (Buesseler et al., 2020b). Nonetheless, our data show that, in a first approximation, the choice of using a 100 m fixed depth to integrate [234]Th fluxes was accurate in many cases.

Overall, *Figure 7a* shows at glance the number of data points in the compilation that could be used to evaluate process in the upper ocean such as export flux and export efficiency (e.g., Buesseler et al., 2020b), scavenging rates of trace metals (e.g., Black et al., 2018; Lemaitre et al., 2020), or particles sinking velocities (e.g., Villa-Alfageme et al., 2016, 2021) by using "deficit" ratios; and those that could be used to study processes such as particle remineralizations (e.g., Usbeck et al., 2002) by using the "excess" ratios. Recent studies have highlighted the role that disaggregation and fragmentation could play in setting the magnitude of flux attenuation (Baker et al., 2017; Briggs et al., 2020; Cavan et al., 2017) and pointed at it as the most important currently unaccounted for process for improving modern-day export flux simulations (Henson et al., 2021). $^{234}$Th excess could be useful to study such mechanisms by detecting low-sinking (i.e., < 20 md$^{-1}$) POC flux below the mixed layer depth (Baker et al., 2017) by high vertical resolution sampling of $^{234}$Th concentrations and POC:$^{234}$Th ratios. Additionally, these data could be used to investigate alternative export pathways, such as active flux by zooplankton and fish diel vertical migration, which is referred to as "active transport" as some of the POC ingested at the surface is transferred to depth as part of their daily migrations and residence at depth (Saba et al., 2021; Steinberg and Landry, 2017). Quantification of the impact of this so-called migration pump (Boyd et al., 2019) has been elusive as it is only indirectly estimated or modeled, with active flux ranging widely as a percent of the passive flux, from <4% (Le Borgne and Rodier, 1997) to equal or larger than the total passive flux measured by classical methods (Yebra et al., 2018). $^{234}$Th excess could shed light into this, since a local maximum in export should be detected around the deep scattering layer where migrators reside at depth during the day if migrators accounted for >40% of the gravitational POC flux (Buesseler and Boyd, 2009).

On another note, data from the compilation could be used to provide revised and more robust global estimates of the ocean's BCP by different means such as, i) evaluating $^{234}$Th-derived POC fluxes regionally, ii) studying the temporal evolution of the BCP strength through time-series data, iii) studying carbon export at shallow depths, and iv) globally analyzing carbon export patterns. As a future study, we propose using the compilation to integrate all data sets to a chosen depth – e.g., to the depth of the 0.1% light level, following recommendation by (Buesseler et al., 2020b)–, so different regions of the upper ocean could be compared across seasons and regions to each other in terms of the BCP strength.

In a second example, $^{234}$Th-$^{238}$U disequilibrium could be combined with $^{210}$Po-$^{210}$Pb and/or sediment traps data to analyze simultaneously key parameters for the functioning of the BCP in different temporal and spatial scales (e.g., Ceballos-Romero et al., 2016; Roca-Martí et al., 2016).

Finally, as an example of potential and powerful uses of this compilation, we show the results of from Davidson et al., (2021), where global $^{234}$Th concentrations were estimated out of the regionally varying euphotic zone by suing data from these data set. $^{234}$Th were regressed to a 2.8-degree grid using a suite of machine learning and minimum variance interpolation algorithms. The gridded data was used to drive a 3D global model using a set of sparse, implicit, and explicit transport matrices derived from a global, coupled General Circulation Model (Khatiwala et al., 2005). The model predicts an average global concentration of $^{234}$Thof 2.03 ± 0.18 dpm L$^{-1}$ and greatest concentrations in ocean gyres (*Figure 7b)*.

Overall, the compilation presented here provides a valuable resource to better understand and quantify how the contemporary oceanic carbon uptake functions and how it will change in future from the multidisciplinary approach that draws knowledge and methodologies from the fields of physics, mathematics, chemistry, biology, oceanography, and marine sciences that is necessary to generate better mechanistic models of the BCP.

**Data availability**

Our data set is archived in the data repository PANGAEA® (http://www.pangaea.de), under the following DOI: https://doi.pangaea.de/10.1594/PANGAEA.918125 (Ceballos-Romero et al., 2021)

**Acknowledgements**

We thank all the authors that have kindly shared data with us via personal communication, with a special mention to Dr. F. Le Moigne, Dr. J. Dunne, Dr. M. Stukel, Dr. P. Martin and Dr. P. Santschi. Particular recognition to Dr. M. Rugters van der Loeff not only for the great amount of data shared - many of them from many years ago - but for also providing detailed explanations to extract the data of interest to the compilation, which has made this enormous undertaking much easier. Special appreciation to Dr. K. Bruland and Dr. K. Coale for, in addition to data, sharing beautiful memories of the $^{234}$Th world.

We also thank P. Davidson, Dr. J. Kenyon, C. García Prieto and C. Núñez-Nevado for their valuable help with the quality control and the review of the compilation for its final release. Special mention in this regard to P. López Puentes for his help in the many hours filling in ad reviewing excel spreadsheets.

This work was partially funded by V Research Programme from Universidad de Sevilla (E.C.R.) and EU FEDER-Junta de Andalucía funded project US-1263369 (M.V.A.). K.O.B. was supported in part by NSF under GEOTRACES, NASA as part of the EXPORTS program and WHOI as part of the Ocean Twilight Zone project. M.V.A. and K.O.B. are also part of an IAEA Coordinated Research Project "Behaviour and effects of natural and anthropogenic radionuclides in the marine environment and their use as tracers for oceanographic studies". The data used are listed in the references, figures, and supporting information.

**Tables**

**Table 1. Summary of the studies included in the compilation in chronological order of publication. Three types are distinguished as data source type: publications in refereed journals (J), publication in repository (R), PhD thesis (T), and personal communication (P) for unpublished data sets released in this compilation. Note that in some cases, published data have been reported in different publications, for which all the publications are indicated. Basic information of sampling details (i.e., region and period) is also included. Additionally, availability of seven different categories of $^{234}$Th data are indicated with an "X" as follows: 1) total $^{234}$Th, 2) dissolved $^{234}$Th, 3) particulate $^{234}$Th, 4) POC:$^{234}$Th ratios from filtration methods, 5) PON:$^{234}$Th ratios from filtration methods, 6) POC:$^{234}$Th ratios from sediment traps, and 7) $^{234}$Th underway sampling.**

| Data source | | Sampling details | | Data available | | | | | | |
|---|---|---|---|---|---|---|---|---|---|---|
| Type | Reference | Sampling region | Sampling period | 1 | 2 | 3 | 4 | 5 | 6 | 7 |
| Era 1 (1969-1991: The beginning of $^{234}$Th as tracer of particle scavenging) | | | | | | | | | | |
| J | Bhat et al., 1969 | Indian Ocean | 01/01/1967 - 31/12/1967 | X | | | | | | |
| J | Matsumoto, 1975 | Pacific Ocean | 31/12/1969 - 01/01/1970 | X | | | | | | |
| J | Krishnaswami et al., 1976 | Pacific Ocean | 03/08/1973 - 09/11/1973 | | | X | | | | |
| J | Knauss et al., 1978 | Pacific Ocean | 01/01/1973 - 31/12/1973 | X | | | | | | |
| J | Santschi et al., 1979 & Santschi et al., 1980 | Narragnasett Bay | 16/01/1978 - 21/02/1979 | X | | X | | | | |
| J | Minagawa and Tsunogai, 1980 | Pacific Ocean | 09/09/1974 - 13/04/1975 | X | | X | | | | |
| J | Kaufman et al., 1981 | Atlantic Ocean | 29/04/1977 - 08/05/1977 | X | | | | | | |
| J | Tanaka et al., 1983 | Pacific Ocean | 25/04/1979 - 26/02/1980 | X | | | | | | |
| J | McKee et al., 1984 | Yangtze River China | 01/11/1981 - 30/12/1981 | X | X | X | | | | |

| J | Coale and Bruland, 1985 | Pacific Ocean | 14/11/1978 - 01/09/1980 | X | X | X | | | | |
| J | Bruland and Coale, 1986 | Pacific Ocean | 01/09/1980 - 09/09/1980 | | X | X | | | | |
| J | McKee et al., 1986 | Amazon River mouth | 08/06/1983 - 10/06/1983 | | X | | | | | |
| J | Tsunogai et al., 1986 | Pacific Ocean | 19/05/1981 - 15/03/1982 | X | | | | | | |
| J | Coale and Bruland, 1987 | Pacific Ocean | 01/10/1981 - 31/08/1983 | | X | X | | | | |
| J | Huh and Beasley, 1987 | Pacific Ocean | 01/10/1985 - 30/10/1985 | | X | X | | | | |
| J | Kershaw and Young, 1988 | Atlantic Ocean | 01/11/1985 - 30/11/1985 | | X | X | | | | |
| J | Bacon and Rutgers van der Loeff, 1989 | Pacific Ocean | 03/07/1979 - 17/08/1980 | X | X | X | | | | |
| J | Dominik et al., 1989 | Lake Geneva | 21/04/1986 - 29/10/1986 | X | | X | | | | |
| J | Moran and Moore, 1989 | Atlantic Ocean | 06/09/1988 - 11/09/1988 | X | | X | | | | |
| J | Murray et al., 1989 | Pacific Ocean | 06/06/1986 - 23/06/1986 | | X | X | | | | |
| J | Schmidt et al., 1990 | Mediterranean Sea | 08/04/1986 - 06/04/1988 | | X | X | | | | |
| J | Lee et al., 1991 | Pacific Ocean | 24/06/1989 - 08/10/1989 | X | | | | | | |
| J | Rutgers van der Loeff and Berger, 1991 | Southern Ocean | 07/11/1987 - 09/12/1987 | X | X | X | | | | |
| J | Wei and Murray, 1991 | Atlantic Ocean | 30/06/1980 - 01/06/1988 | | X | X | | | | |
| **Era 2 (1992-2000: Introduction of the empirical determination of POC export from [234]Th profiles)** | | | | | | | | | | |

| J | | | | | | | | | | |
|---|---|---|---|---|---|---|---|---|---|---|
| J | Baskaran et al., 1992 | Atlantic Ocean | 01/10/1989 - 30/06/1991 | X | X | X | | | | |
| J | Buesseler et al., 1992 & Cochran et al., 1993 | Atlantic Ocean | 19/04/1989 - 05/06/1989 | X | X | X | X | X | X | |
| J | Moran and Buesseler, 1992 | Atlantic Ocean | 14/05/1991 | X | X | X | | | | |
| J | Sarin et al., 1992 | Indian Ocean | 01/03/1991 - 30/04/1991 | X | X | | | | | |
| J | Schmidt et al., 1992 | Mediterranean Sea | 17/03/1987 - 14/05/1987 | | X | X | | | | |
| J | Wei and Murray, 1992 | Dabob Bay | 01/01/1987 - 31/12/1987 | X | X | X | | | | |
| J | Baskaran and Santschi, 1993 | Galveston coast | 01/05/1990 - 31/12/1990 | X | | X | | | | |
| J | Moran and Buesseler, 1993 | Atlantic Ocean | 25/06/1991 - 28/07/1992 | X | X | X | | | | |
| J | Buesseler et al., 1994 | Atlantic Ocean | 24/04/1992 - 08/05/1992 | X | | | X | X | X | |
| J | Kuptsov, 1994 | Arctic Ocean | 08/01/1993 - 31/10/1993 | | | | X | | X | |
| J | Sarin et al., 1994 | Indian Ocean | 01/12/1988 - 31/12/1988 | | X | | | | | |
| J | Sarin et al., 1994a | Indian Ocean | 15/03/1991 - 24/02/1992 | X | | | | | | |
| J | Thunell et al., 1994 | Pacific Ocean | 07/01/1988 - 02/07/1988 | | | X | | | | |
| J | Buesseler et al., 1995 | Pacific Ocean | 01/03/1992 - 02/12/1992 | | X | X | X | X | | |
| J | Cochran et al., 1995 | Arctic Ocean | 19/07/1992 - 11/08/1993 | | X | X | | | | |
| J | Huh and Prahl, 1995 | Pacific Ocean | 07/08/1993 - 09/08/1993 | X | X | X | | | | |
| J | Niven et al., 1995 | Atlantic Ocean | 17/03/1993 - 17/05/1993 | X | X | X | | | | |

| | | | | | | | | | |
|---|---|---|---|---|---|---|---|---|---|
| J | Shimmield et al., 1995 | Southern Ocean | 07/12/1992 | | X | X | | | |
| J | Bacon et al., 1996 | Pacific Ocean | 22/03/1992 - 21/10/1992 | X | X | X | | | |
| J | Baskaran et al., 1996 | Atlantic Ocean | 16/03/1992 - 25/06/1992 | X | X | X | | | |
| J | Murray et al., 1996 & Dunne et al., 1997 & Murray et al., 2005 | Pacific Ocean | 07/02/1992 - 13/09/1992 | X | X | X | X | X | X |
| J | Sarin et al., 1996 | Indian Ocean | 12/04/1994 - 12/08/1995 | X | X | | | | |
| J | Guo et al., 1997 & Santschi et al., 1995 | Atlantic Ocean | 01/06/1992 - 31/07/1994 | X | X | X | | | |
| J | Gustafsson et al., 1997a | Atlantic Ocean | 01/09/1993 – 31/05/1994 | X | | X | | | |
| J | Gustafsson et al., 1997b | Atlantic Ocean | 01/09/1993 - 31/05/1994 | X | | X | X | | |
| J | Langone et al., 1997 | Southern Ocean | 07/12/1994 - 13/12/1994 | X | X | X | | | |
| J | Moran et al., 1997 | Arctic Ocean | 01/08/1994 - 30/09/1994 | X | X | X | X | X | |
| J | Rutgers van der Loeff et al., 1997 & Friedrich and Rutgers van der Loeff 2002 | Southern Ocean | 02/10/1992 - 22/11/1992 | X | X | X | X | X | |
| J | Buesseler et al., 1998 | Indian Ocean | 09/01/1995 - 12/09/1995 | X | X | X | X | X | |
| J | Gustafsson et al., 1998 | Atlantic Ocean | 26/09/1993 - 22/05/1994 | X | | X | | | |

| | | | | | | | | | | |
|---|---|---|---|---|---|---|---|---|---|---|
| J | Wei and Hung, 1998 | Pacific Ocean | 01/11/1991 - 30/11/1991 | | X | X | | | | |
| J | Kersten et al., 1998 | Atlantic Ocean | 02/02/1994 - 09/11/1994 | | X | X | | | | |
| J | Charette et al., 1999 | Pacific Ocean | 01/02/1996 - 28/02/1997 | X | X | X | X | | X | |
| J | Charette and Moran, 1999 | Atlantic Ocean | 15/05/1996 - 11/06/1996 | X | X | X | X | | | |
| J | Santschi et al., 1999 | Atlantic Ocean | 17/05/1993 - 12/07/1994 | X | X | X | X | | | |
| J | Smoak et al., 1999 | Pacific Ocean | 16/10/1993 - 31/03/1996 | | | | X | | | |
| J | Benitez-Nelson et al., 2000 | Atlantic Ocean | 01/03/1997 - 31/08/1997 | | X | X | X | | | |
| J | Buesseler et al., 2000 | Atlantic Ocean | 06/10/1997 | | | | X | X | X | |
| J | Charette and Buesseler, 2000 | Southern Ocean | 09/02/1999 - 23/02/1999 | X | X | X | X | X | | |
| J | Cochran et al., 2000 | Southern Ocean | 08/10/1996 - 04/05/1997 | X | | X | X | X | | |
| J | Gulin, 2000 | Atlantic Ocean | 02/11/1998 - 11/06/1999 | X | X | X | | | | |
| J | Hall et al., 2000 | Atlantic Ocean | 11/06/1997 - 14/06/1997 | X | X | X | X | | | X |
| J | Moran and Smith, 2000 | Arctic Ocean | 01/08/1995 - 30/09/1995 | X | X | X | X | X | | |
| **Era 3 (2001-2009: Improvements of the $^{234}$Th technique)** | | | | | | | | | | |
| J | Benitez-Nelson et al., 2001a | Pacific Ocean | 01/04/1999 - 31/03/2000 | X | | | X | | X | |
| J | Buesseler et al., 2001 | Southern Ocean | 24/10/1997 - 15/03/1998 | X | X | X | X | X | | |
| J | Cai et al., 2001 | Pacific Ocean | 01/04/1999 - 01/05/1999 | X | X | X | X | | | |
| J | Charette et al., 2001 | Atlantic Ocean | 19/03/1995 - 31/08/1997 | X | X | X | X | | X | |

| | | | | | | | | | |
|---|---|---|---|---|---|---|---|---|---|
| J | Dai and Benitez-Nelson, 2001 | Atlantic Ocean | 06/07/1996 - 12/07/1997 | X | X | X | | | |
| J | Hernes et al., 2001 | Pacific Ocean | 07/02/1992 - 13/09/1992 | | | X | | X | |
| J | Kim and Church, 2001 | Atlantic Ocean | 06/10/1996 - 20/08/1997 | X | X | X | | | |
| J | Porcelli et al., 2001 | Atlantic Ocean | 01/06/1995 – 31/06/1995 | X | X | X | | | |
| J | Turnewitsch and Springer, 2001 | Atlantic Ocean | 01/07/1997 - 28/02/1998 | X | X | X | | | |
| R | Turnewitsch, 2001 | Indian Ocean | 01/02/1998 - 31/03/1998 | | X | X | | | |
| J | Amiel et al., 2002 | Atlantic Ocean | 01/05/1998 - 30/09/1999 | X | X | X | X | X | |
| J | Cai et al., 2002 | Pacific Ocean | 15/11/1997 | X | X | X | X | | |
| J | Coppola et al., 2002 | Arctic Ocean | 07/07/1999 | X | X | X | X | X | |
| J | Foster and Shimmield, 2002 | Atlantic Ocean | 19/06/1999 - 27/06/1999 | X | X | X | X | | |
| J | Frignani et al., 2002 | Atlantic Ocean | 17/09/1994 – 28/06/1995 | X | X | X | | | |
| J | Guo et al., 2002 | Atlantic Ocean | 05/07/2000 | X | X | X | X | | |
| J | Rutgers van der Loeff et al., 2002a | Southern Ocean | 30/12/1995 - 18/01/1996 | X | X | X | X | | |
| J | Rutgers van der Loeff et al., 2002b | Arctic Ocean | 26/06/1997 - 11/08/1997 | X | X | X | | | |
| J | Schmidt et al., 2002b & Schmidt et al., 2009 | Mediterranean Sea | 09/02/1994 - 29/05/1995 | X | X | X | X | X | |

| | | | | | | | | | | |
|---|---|---|---|---|---|---|---|---|---|---|
| J | Schmidt et al., 2002a | Atlantic Ocean | 01/06/1997 - 31/01/1998 | X | X | X | X | | | X |
| J | Somayajulu et al., 2002 | Bay of Bengal | 16/12/1991 - 23/12/1991 | X | | | | | | |
| J | Usbeck et al., 2002 | Southern Ocean | 20/03/1999 - 06/05/1999 | X | X | X | | | | |
| J | Baskaran et al., 2003 | Arctic Ocean | 24/07/1998 - 10/10/1998 | X | X | X | X | | | |
| J | Chen et al., 2003 | Pacific & Arctic Oceans | 01/08/1999 - 31/08/1999 | X | X | X | X | | | |
| J | Moran et al., 2003 | Atlantic Ocean | 01/07/1999 - 31/07/1999 | X | X | X | X | | | |
| J | Radakovitch et al., 2003 | Mediterranean Sea | 06/03/1997 - 12/09/1997 | X | X | X | X | | | |
| R | Rutgers van der Loeff and Vöge, 2003 | Southern Ocean | 24/10/2000 - 29/11/2000 | X | X | X | | | | |
| J | Santschi et al., 2003 | Atlantic Ocean | 01/05/2001 - 31/05/2001 | X | X | X | X | | X | |
| J | Sweeney et al., 2003 | Atlantic Ocean | 01/03/1993 - 30/09/1995 | | | | X | | X | |
| J | Gustafsson et al., 2004 | Atlantic Ocean | 17/01/1998 - 15/08/2000 | X | | X | | | | |
| J | Hung et al., 2004 | Atlantic Ocean | 07/07/2000 - 21/05/2001 | X | X | X | X | | | |
| J | Kawakami et al., 2004 & Kawakami, 2009 & Yang et al., 2004 | Pacific Ocean | 12/11/1997 - 04/06/2000 | | X | X | X | | | |
| J | Savoye et al., 2004 & Buesseler et al., 2006 | Southern Ocean | 03/11/2001 - 05/12/2001 | X | | X | X | X | | |

| | | | | | | | | | |
|---|---|---|---|---|---|---|---|---|---|
| J | Smith et al., 2004 | Atlantic Ocean | 15/06/1998 - 15/07/1998 | | X | X | | | |
| J | Smoak et al., 2004 | Atlantic Ocean | 15/11/1995 - 30/04/1996 | | | | | X | |
| J | Trimble et al., 2004 | Arctic Ocean | 01/08/2000 - 31/08/2000 | | X | X | | | |
| J | Waples, 2004 | Lake Michigan | 15/02/1999 - 18/10/1999 | | X | X | | | |
| J | Aono et al., 2005 | Pacific Ocean | 07/07/2001 - 31/07/2001 | X | X | X | X | X | X |
| J | Buesseler et al., 2005 | Southern Ocean | 30/01/2002 - 20/02/2002 | X | | X | X | | |
| J | Coppola et al., 2005 & Coppola et al., 2006 | Southern Ocean | 18/01/1999 - 15/02/1999 | X | X | X | X | | |
| J | Ma et al., 2005 | Arctic Ocean | 07/01/2003 - 30/09/2003 | X | | X | X | | |
| J | Moran et al., 2005 & Lepore and Moran, 2007 | Arctic Ocean | 06/05/2002 - 26/08/2002 | X | X | X | X | X | |
| J | Trimble and Baskaran, 2005 | Arctic Ocean | 01/08/2000 - 31/08/2000 | X | X | X | X | X | |
| J | Cai et al., 2006b | Pacific Ocean | 21/02/2004 - 28/02/2004 | X | X | X | | | |
| J | Cai et al., 2006c | Pacific Ocean | 04/05/2005 - 09/05/2005 | X | X | X | X | | |
| J | Gustafsson et al., 2006 | Atlantic Ocean | 15/08/2000 - 31/03/2001 | | | | X | | X |
| J | Rodriguez y Baena et al., 2006, 2008 | Southern Ocean | 17/11/2003 - 18/01/2004 | X | | | X | | |
| J | Schmidt, 2006 | Atlantic Ocean | 26/12/1988 - 08/09/1989 | X | X | X | X | | |
| J | Speicher et al., 2006 | Mediterranean Sea | 16/03/2004 - 02/05/2004 | X | X | X | X | | |
| J | Thomalla et al., 2006, 2008 | Atlantic Ocean | 02/05/2004 - 26/05/2004 | X | | | X | | |
| J | Giuliani et al., 2007 | Mediterranean Sea | 16/03/1997 - 07/04/1997 | X | X | X | X | | |

| | | | | | | | | | | |
|---|---|---|---|---|---|---|---|---|---|---|
| J | Hung and Gong, 2007 | Pacific Ocean | 18/08/2006 | X | X | X | X | | X | |
| J | Kawakami and Honda, 2007 | Pacific Ocean | 16/10/2002 - 21/08/2004 | | X | X | X | | | |
| J | Lalande et al., 2007 & Lepore et al., 2007 & Lepore and Moran, 2007 | Arctic Ocean | 24/05/2004 - 25/08/2004 | X | X | X | X | X | X | |
| J | Morris et al., 2007 | Southern Ocean | 10/11/2004 - 14/01/2005 | X | | | X | | | |
| R | Rutgers van der Loeff, 2007b | Atlantic Ocean | 21/10/2005 - 13/11/2005 | X | X | X | | | | |
| R | Rutgers van der Loeff, 2007a | Southern Ocean | 16/01/2006 - 30/01/2006 | X | X | X | | | | X |
| J | Stewart et al., 2007 | Mediterranean Sea | 04/03/2003 - 30/06/2003 | X | | | X | | X | |
| T | Thomalla, 2007 | Atlantic Ocean | 17/05/2003 - 04/10/2003 | | X | X | X | | | |
| J | Amiel and Cochran, 2008 | Arctic Ocean | 01/09/2002 – 31/07/2004 | X | X | X | X | | | |
| J | Buesseler et al., 2008 | Atlantic Ocean | 24/06/2004 - 25/08/2005 | X | | | X | X | X | |
| J | Cai et al., 2008 | Pacific Ocean | 04/04/2004 - 04/05/2004 | X | | | X | X | | |
| J | Chen et al., 2008 | Pacific Ocean | 01/07/2000 - 30/11/2002 | X | X | X | X | | | |
| J | Lalande et al., 2008 | Arctic Ocean | 10/07/2003 - 31/05/2005 | X | X | X | X | | X | |
| J | Lamborg et al., 2008b | Pacific Ocean | 23/06/2004 - 09/08/2005 | | | | | | X | |
| J | Lampitt et al., 2008 | Atlantic Ocean | 12/07/2003 - 06/07/2007 | X | | | X | | X | |
| J | Maiti et al., 2008 | Pacific Ocean | 10/03/2005 - 28/03/2005 | X | | | X | X | X | |
| J | Savoye et al., 2008 | Southern Ocean | 23/01/2005 - 12/02/2005 | X | | | X | X | X | |

| | | | | | | | | | | |
|---|---|---|---|---|---|---|---|---|---|---|
| J | Brew et al., 2009 & Stewart et al., 2011 | Atlantic Ocean | 09/05/2006 - 24/03/2007 | X | | X | X | | X | |
| J | Buesseler et al., 2009 | Pacific Ocean | 23/06/2004 - 16/08/2005 | X | | X | X | X | X | |
| R | Charette, 2009 | Southern Ocean | 16/01/2006 - 06/08/2006 | X | | | | | | |
| J | Cochran et al., 2009 | Mediterranean Sea | 10/03/1999 - 30/04/2005 | X | X | X | | | | |
| J | Lepore et al., 2009 | Atlantic Ocean | 03/12/2004 - 06/05/2005 | X | X | X | X | | X | |
| J | Schmidt et al., 2009 | Mediterranean Sea | 03/09/2004 - 24/09/2004 | X | X | X | X | | X | |
| J | Szlosek et al., 2009 | Mediterranean Sea | 06/03/2003 - 01/05/2005 | | | | X | | X | |
| J | Wei et al., 2009 | Pacific Ocean | 06/03/2006 - 08/03/2006 | | X | X | X | | | |
| **Era 4 (2010-present: GEOTRACES program and a new way to study the ocean)** | | | | | | | | | | |
| J | Buesseler et al., 2010 | Southern Ocean | 09/01/2009 | X | | X | X | | X | |
| J | Cai et al., 2010 | Arctic Ocean | 30/07/2007 - 23/09/2007 | X | | X | X | | | |
| J | Hung and Gong, 2010 | Pacific Ocean | 03/12/2008 - 04/12/2008 | | | | X | | X | |
| J | Hung et al., 2010 | Atlantic & Pacific Oceans | 01/08/2005 - 16/06/2009 | X | X | X | X | | X | |
| J | Kawakami et al., 2010 | Pacific Ocean | 26/09/2005 - 17/10/2005 | | X | X | X | | X | |
| J | Sanders et al., 2010 | Atlantic Ocean | 24/07/2007 - 23/08/2007 | X | | X | X | X | | |
| J | Yu et al., 2010 | Arctic Ocean | 30/07/2003 - 12/09/2003 | X | X | X | X | | | |
| J | Evangeliou et al., 2011 | Mediterranean Sea | 01/07/2008 - 17/01/2009 | X | X | X | X | | | |
| J | Jacquet et al., 2011 | Southern Ocean | 17/01/2007 - 20/02/2007 | X | | X | X | | X | |
| J | Kim et al., 2011 | Pacific Ocean | 04/04/2007 - 23/02/2008 | X | | | X | | | |

| | | | | | | | | | | |
|---|---|---|---|---|---|---|---|---|---|---|
| J | Martin et al., 2011 | Atlantic Ocean | 01/05/2008 - 21/05/2008 | X | | | X | X | X | |
| J | Rutgers van der Loeff et al., 2011 | Southern Ocean | 13/02/2008 - 11/04/2008 | X | | X | X | | | X |
| J | Shaw et al., 2011 | Southern Ocean | 15/03/2009 - 15/04/2009 | X | | | X | | X | |
| J | Stukel et al., 2011 | Pacific Ocean | 11/05/2006 - 05/06/2006 | X | | | X | | | |
| J | Wei et al., 2011 | Pacific Ocean | 01/10/2006 - 31/12/2008 | X | X | X | X | | X | |
| J | Xu et al., 2011 | Atlantic Ocean | 30/04/2006 - 05/05/2006 | | | X | X | | X | |
| J | Gustafsson and Andersson, 2012 | Arctic Ocean | 13/07/2001 - 31/07/2001 | X | X | X | X | | | |
| J | Hung et al., 2012 | Pacific Ocean | 02/08/2008 - 12/12/2008 | | | | X | | X | |
| R | Kawakami, 2012 | Pacific Ocean | 09/09/2007 - 28/10/2008 | | X | X | X | X | X | |
| J | Moran et al., 2012 | Pacific Ocean | 29/03/2008 - 31/07/2008 | X | | | X | | X | |
| J | Yu et al., 2012 | Arctic Ocean | 04/08/2008 - 08/09/2008 | X | X | X | X | | | |
| J | Zhou et al., 2012 | Southern Ocean | 24/05/2008 - 08/06/2008 | X | | X | X | | | |
| J | Baumann et al., 2013 | Pacific Ocean | 29/03/2008 - 15/07/2010 | X | | | X | | X | |
| J | Evangeliou et al., 2013 | Mediterranean Sea | 01/01/2010 - 10/01/2010 | X | X | X | X | | | |
| J | Haskell et al., 2013 | Pacific Ocean | 01/02/2010 - 03/04/2011 | X | | | X | | X | |
| R | JGOFS India2013 | Indian Ocean | 09/02/1997 - 11/02/1997 | | X | | | | | |
| J | Le Moigne et al., 2013a | Atlantic Ocean | 13/07/2009 - 08/08/2009 | X | | X | X | | | |
| T | Luo, 2013 | Pacific Ocean | 01/06/2010 - 30/06/2010 | X | X | X | X | | X | |
| J | Martin et al., 2013 | Southern Ocean | 12/01/2009 - 06/03/2009 | X | X | X | X | | X | X |
| T | Owens 2013 | Atlantic Ocean | 10/09/2010 - 05/10/2010 | X | | X | X | X | | |
| T | Owens 2013 | Southern Ocean | 30/12/2009 - 07/02/2010 | X | | X | X | X | X | X |

| | | | | | | | | | | |
|---|---|---|---|---|---|---|---|---|---|---|
| T | Owens 2013 | Southern Ocean | 16/03/2010 - 01/05/2010 | X | | X | X | X | X | X |
| J | Planchon et al., 2013 | Southern Ocean | 21/02/2008 - 14/03/2008 | X | | X | X | | | |
| J | Schmidt et al., 2013 | Atlantic Ocean | 01/05/2002 - 31/05/2002 | X | X | X | X | | | |
| J | Stukel et al., 2013 | Pacific Ocean | 04/04/2007 - 28/10/2008 | X | | | X | | X | |
| J | Zhou et al., 2013 | Pacific Ocean | 01/08/2007 - 31/08/2007 | X | | X | X | | | |
| J | Le Moigne et al., 2014 | Atlantic Ocean | 19/10/2010 - 24/01/2012 | X | | X | X | X | | |
| J | Luo et al., 2014 & Luo, 2013 | Pacific Ocean | 10/03/2009 - 09/02/2011 | X | X | X | X | X | | |
| T | Pabortsava, 2014 | Atlantic Ocean | 07/02/2011 | X | | X | X | X | | |
| J | Cai et al., 2015 | Pacific Ocean | 18/07/2009 - 24/05/2011 | X | | X | X | | | |
| J | Estapa et al., 2015 | Atlantic Ocean | 30/09/2011 - 12/02/2012 | X | | | | | | X |
| J | Kawakami et al., 2015 | Pacific Ocean | 03/06/2006 - 17/07/2006 | | X | X | X | | X | |
| J | Le Moigne et al., 2015 | Arctic Ocean | 05/06/2012 - 01/07/2012 | X | | X | X | | | |
| J | Owens et al., 2015 | Atlantic Ocean | 15/10/2010 - 07/04/2011 | X | | X | X | | | X |
| J & T | Planchon et al., 2015 & Lemaitre et al., 2016 & Lemaitre, 2017 | Southern Ocean | 20/10/2011 - 18/11/2011 | X | | X | X | X | X | |
| J | Puigcorbé et al., 2015 | Pacific Ocean | 10/07/2008 - 07/08/2008 | X | | X | X | X | X | |
| R | Roca-Martí, et al.., 2015 | Southern Ocean | 26/01/2012 - 02/03/2012 | X | | X | X | X | | |
| J | Rosengard et al., 2015 | Atlantic & Indian Oceans | 14/01/2012 - 20/03/2012 | X | | X | X | | | |

| | | | | | | | | | |
|---|---|---|---|---|---|---|---|---|---|
| J | Stukel et al., 2015, 2016, 2017 | Pacific Ocean | 22/04/2009 - 23/07/2010 | X | | | X | | X | |
| J | Ceballos-Romero et al., 2016 | Atlantic Ocean | 01/05/2010 - 07/08/2010 | X | | X | X | | X | |
| J | Haskell et al., 2016 | Pacific Ocean | 16/01/2013 - 19/06/2014 | X | | | X | | X | |
| J | Le Moigne et al., 2016 | Southern Ocean | 13/01/2013 - 05/02/2013 | X | | X | X | | | |
| J | Maiti et al., 2016 | Atlantic Ocean | 01/03/2012 - 31/03/2013 | | | X | X | | | |
| J | Roca-Martí et al., 2016 | Arctic Ocean | 03/08/2012 - 08/10/2012 | X | | X | X | X | | |
| J | Turnewitsch et al., 2016 | Atlantic Ocean | 28/09/2009 - 19/10/2009 | X | | X | X | | | |
| J | Zhou et al., 2016 | Atlantic Ocean | 05/06/2008 - 30/09/2009 | | X | X | X | X | | |
| J | Anand et al., 2017 | Indian Ocean | 16/03/2014 - 19/04/2014 | X | | X | X | | | |
| J | Puigcorbé et al., 2017a | Atlantic Ocean | 02/05/2010 - 04/07/2010 | X | | | X | | | |
| J | Puigcorbé et al., 2017b | Southern Ocean | 07/01/2012 - 11/03/2012 | X | | X | X | X | X | |
| J | Roca-Martí et al., 2017 | Southern Ocean | 07/01/2012 - 11/03/2012 | X | | X | X | X | X | |
| J | Anand et al., 2018b | Indian Ocean | 24/10/2013 - 29/01/2014 | X | | | X | | | |
| J | Anand et al., 2018a | Indian Ocean | 09/05/2014 - 28/05/2014 | X | | X | X | | | |
| J | Black et al., 2018 | Pacific Ocean | 29/10/2013 - 19/12/2013 | X | | X | X | | | X |
| J | Lemaitre et al., 2018 | Atlantic Ocean | 20/05/2014 - 26/06/2014 | X | | X | X | | | |

| Type | Reference | Location | Date | | | | | | |
|---|---|---|---|---|---|---|---|---|---|
| J | Schlitzer et al., 2018 | Mediterranean Sea | 04/05/2013 – 30/05/2013 | X | | | | | |
| J | Stukel et al., 2019 | Pacific Ocean | 09/08/2014 - 11/05/2016 | X | | | X | | X |
| R | Stukel, 2019 | Pacific Ocean | 01/06/2017 - 29/06/2017 | X | | | | | |
| J | Umhau et al., 2019 | Pacific Ocean | 01/02/2014 - 30/09/2015 | X | | X | X | X | X |
| J | Alkalay et al., 2020 | Mediterranean Sea | 25/05/2017 - 03/06/2018 | X | | | X | | X |
| J | Buesseler et al., 2020a | Pacific Ocean | 14/08/2018 - 10/09/2018 | X | | X | X | | X |
| J | Xie et al., 2020 | Pacific Ocean | 13/04/2017 - 23/07/2017 | X | | | | | |
| P | Pers. comm. from K. Buesseler | Southern Ocean | 06/01/2009 - 30/01/2009 | X | | X | X | X | |
| P | Pers. comm. from K. Buesseler | Southern Ocean | 20/02/2009 - 14/03/2009 | X | | | | | |
| P | Pers. comm. from E. Ceballos-Romero 2021 | Southern Ocean | 01/12/2013 - 12/12/2013 | X | | X | X | | X |
| P | Pers. comm. from Gasser | Pacific Ocean | 27/10/2004 - 11/12/2004 | X | X | X | | | |
| P | Pers. comm. from J. Kenyon | Pacific Ocean | 25/09/2018 - 23/10/2018 | X | | X | X | X | |
| P | Pers. comm. from K. Kenyon | Pacific Ocean | 26/10/2018 - 24/11/2018 | X | | X | X | X | |
| P | Pers. comm. from S. Owens | Atlantic Ocean | 08/11/2010 - 26/11/2010 | X | | X | | | X |
| P | Pers. comm. from K. Pabortsava | Atlantic Ocean | 11/08/2011 - 07/09/2011 | X | | | X | X | X |

| P | Pers. comm. from V. Puigcorbe | Mediterranean Sea | 08/03/2009 - 20/09/2009 | X | | | X | | X | |







**Table 2** Selected Ocean programs from the 50 years of [234]Th data compiled and significant activities within these programs with [234]Th samplings chronologically ordered. Activities are categorized as follows: i) cruises (C), ii) experiments (E), iii) Long-Term Time-Series station (TSS), iv) Process Study (PS), and v) subprogram (SP). Programs and activities are detailed by era, as indicated with a cross.

| # | Acronym | Activity | Name | Type | Dates | 1 | 2 | 3 | 4 |
|---|---------|----------|------|------|-------|---|---|---|---|
| | | | **Ocean programs** | | | | | **Eras** | |
| 1 | GEOSECS | | Geochemical Ocean Sections Study | | 1972-1978 | x | | | |
| 2 | LTER | | Long Term Ecological Research (LTER) | | 1980-present | | | | |
| | | PAL-LTER | Palmer Antartica Long Term Ecologycal Research | SP | 1990-present | | | | x |
| | | CCE-LTER | California Current Ecosystem Long Term Ecologycal Research | SP | 2004-present | | | | x |
| 3 | JGOFS | | Joint Global Ocean Flux Study | | 1984-2003 | | | | |
| | | DYFAMED | DYnamique des Flux de mAtière en MEDiterranée | TSS | 1987-present | | x | x | |
| | | BATS | Bermuda Atlantic Time-series Study | TSS | 1988-present | | x | x | x |
| | | HOT | Hawaii Ocean Time-Series | TSS | 1988-present | | | x | x |
| | | MEDATLANTE | MEDATLANTE | SP | 1989-1990 | | x | | |
| | | NABE | North Atlantic Bloom Experiment | PS | 1989 | | x | | |
| | | OSP | Station PAPA Ocean Weather Station "P" | TSS | 1990*-present | | x | | x |
| | | DYFAMED | DYnamique des Flux de mAtière en MEDiterranée | TSS | 1987-present | | x | x | |
| | | EqPac | Equatorial Pacific Project | PS | 1992 | | x | | |
| | | ANTARES | ANTArctic RESearch | SP | 1992-1998 | | | x | |
| | | Arabian Sea | Arabian Sea Process Study | PS | 1994-1996 | | x | | |
| | | DYNAPROC | DYNAPROC | C | 1995 | | | x | |
| | | AESOPS | Antarctic Environment and Southern Ocean Process Study | PS | 1996-1998 | | x | | |
| | | KNOT | Kyodo North Pacific Ocean Time Series | TSS | 1998-2000 | | | x | |
| | | SEATS | South-East Asia Time Series | TSS | 1999-present | | x | x | x |
| | | SOIREE | Southern Ocean Iron Release Experiment | E | 1999 | | x | | |
| 4 | OMEX | | Ocean Margins Exchange Program (I & II) | | 1993-2000 | | x | x | |
| 5 | CARIACO | | CArbon Retention In A Colored Ocean time series program | | 1995-2017 | | | x | |
| 6 | SBI | | Shelf-Basin Interactions (SBI) Phase II field program | | 1998-2009 | | | x | |
| 7 | MedFlux | | MedFlux | | 2003-2006 | | | x | |
| 8 | AMT | | Atlantic Meridional Transect (AMT) programme | | 2007-2012 | | | x | |
| 9 | BEST | | BEST-BSIERP field program | | 2007-2013 | | | | x |
| 10 | CHOICE-C | | Carbon Cycling in the China Seas: Budget, Controls and Ocean Acidification | | 2009-2011 | | | x | x |

| 11 | UKOA | | UK's Ocean Acidification | | 2010-2016 | | | | x |
| 12 | GEOTRACES | | GEOTRACES | | 2010-present | | | x | x |
| 13 | EXPORTS | | NASA EXport Processes in the Ocean from RemoTe Sensing | | 2017-present | | | | x |

*Observations at Station Papa (former Ocean Station Peter and referred to as OSP or Ocean Station "P") started in 1949, although the larger surveys became a focus of activities in support of JGOFS during the 1990s (Freeland, 2007).


**Figures**

**Figure 1. (a) Map showing the distribution of sampling stations catalogued as i) unpublished (yellow diamonds), ii) published exclusively in repositories (dark blue squares), and iii) published in referred journals (magenta dots). (b) Map showing data density by sampling location. (c) Map showing long-term, high-frequency time-series stations (TSS) either i) "operational" (blue diamonds), ii) "registered" for future operation (yellow diamonds), iii) "inactive" (red asterisks), or iv) "close" (black crosses) (source: https://www.ocean-ops.org) and those locations including $^{234}$Th sampling that match a TSS (light blue dots).**


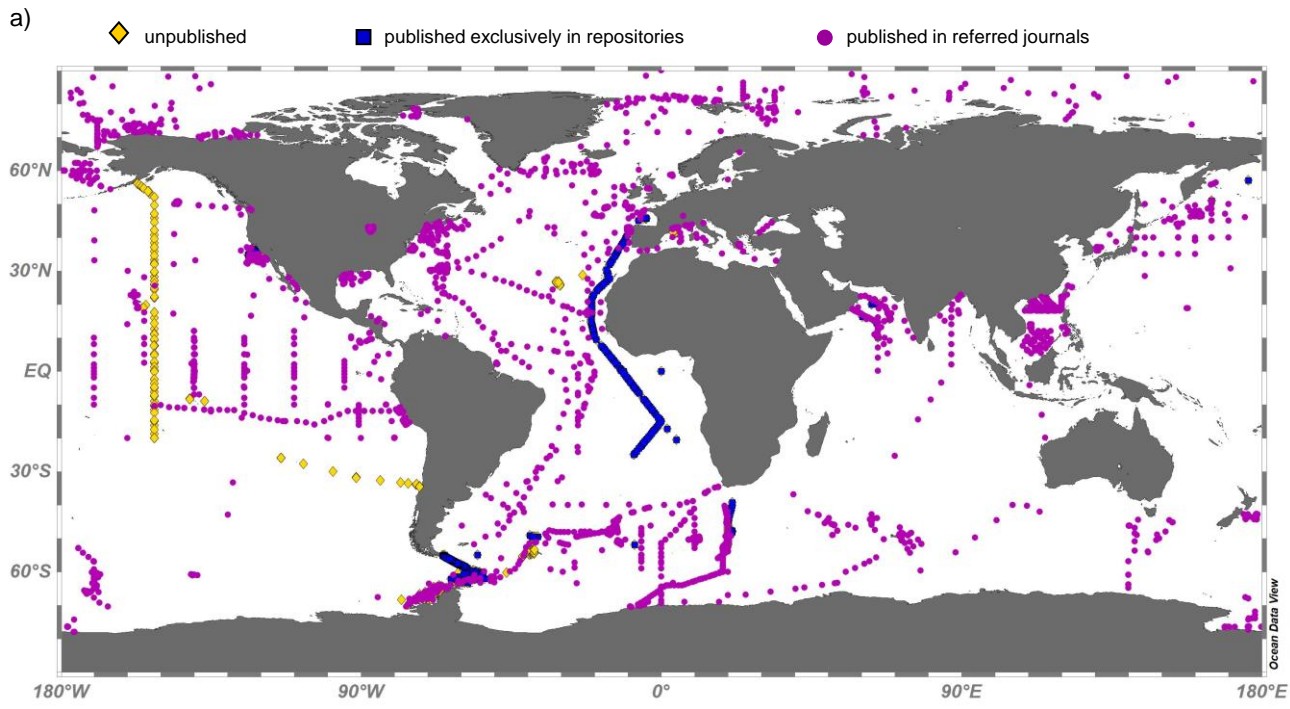


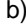

c)

◆ operational    ◆ registered    * inactive    ✕ closed    ● $^{234}$Th sampling

**Figure 2. (a)** Time distribution of annual field surveys with $^{234}$Th measurements since the first reported sampling in 1967 separated by Ocean. **(b)** Equivalent figure for $^{234}$Th data points. Note that Oceans are distinguished as i) Atlantic, ii) Arctic, iii) Indian, iv) Pacific and v) Southern, with and additional category for vi) Others, which includes rivers, lakes, and bays. **(c)** Histogram of $^{234}$Th data points sampled since 1967 separated by data type as follows: i) $^{238}$U, ii) total $^{234}$Th, iii) dissolved $^{234}$Th, iv) particulate $^{234}$Th, v) POC:$^{234}$Th ratios and vi) PON:$^{234}$Th ratios. Dotted lines indicate tipping points in the $^{234}$Th timeline.

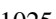


**Figure 3. Summary of [234]Th timeline. The 50 years of [234]Th data have been divided in 4 eras whose beginning has been defined based on seminal [234]Th-related publications that changed the course of the technique. Important years within eras are marked with circles and numerated by era. For each era, key milestones are indicated, with cumulative**


**numeration along the timeline. A brief description of each milestone is also provided. Numbers in brackets denote the following references: [1] Bhat et al., (1969), [2] Turekian, (1977), [3] Santschi et al., (1979), [4] Mann et al., (1984), [5] Coale and Bruland, (1985), [6] Bruland and Coale, (1986), [7] Coale and Bruland, (1987),[8] Eppley et al., (1989), [9] Buesseler et al., (1992), [10] Baskaran et al., (1992), [11] Moran and Buesseler, (1993), [12] Shimmield et al., (1995), [13] Buesseler et al., (1995), [14] Bacon et al., (1996), [15] Rugters van der Loeff and Moore, (1999), [16] Buesseler et al., (2001), [17] Benitez-Nelson et al., (2001), [18] Friedrich and Rutgers van der Loeff, (2002), [19] Pike et al., (2005),**


**[20] Cai et al., (2006),[21] Cai et al., 2010, [22] Cutter et al., (2010), [23] Maiti et al., (2012), [24]** (Mawji et al., 2014), **[25]** (Schlitzer et al., 2018a)**, [26] Clevenger et al., (2021), [27] GEOTRACES Intermediate Data Product Group, (2021).**

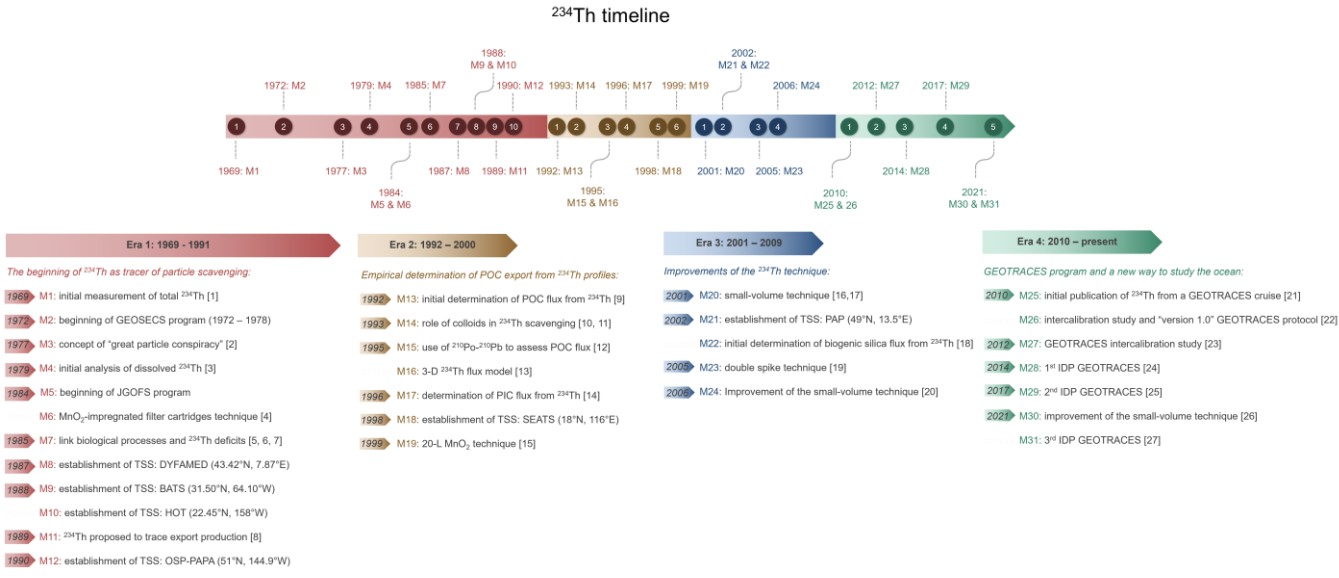





**Figure 4. Map showing the distribution of sampling stations reported (a) within era 1 distinguished by the type of data available as follows: i) dissolved plus particulate [234]Th (yellow diamonds), ii) [238]U plus total [234]Th (dark blue squares),**

iii) no $^{238}$U (magenta dots); and within eras 2 (b), 3 (c) and 4 (d) distinguished by the type of data available as follows: i) total $^{238}$U plus total $^{234}$Th plus POC:$^{234}$Th ratios (yellow diamonds), ii) $^{238}$U not sampled (dark blue squares), iii) total $^{234}$Th not sampled (magenta dots), and iv) POC:$^{234}$Th ratios not sampled (light blue triangles). The "no parameter" categories mean that from the data shown in the maps: i.e., $^{238}$U, dissolved, particulate and total $^{234}$Th in panel (a) and $^{238}$U, total $^{234}$Th, and POC:$^{234}$Th ratios from both filtration methods and sediment traps for panels (b), (c) and (d), the "no" category has not been reported by the original authors and therefore is not included in the compilation, but at least one of the other parameters has been reported. Therefore, in panels (b), (c) and (d), only locations that fall into category i) can be used to estimate POC fluxes.

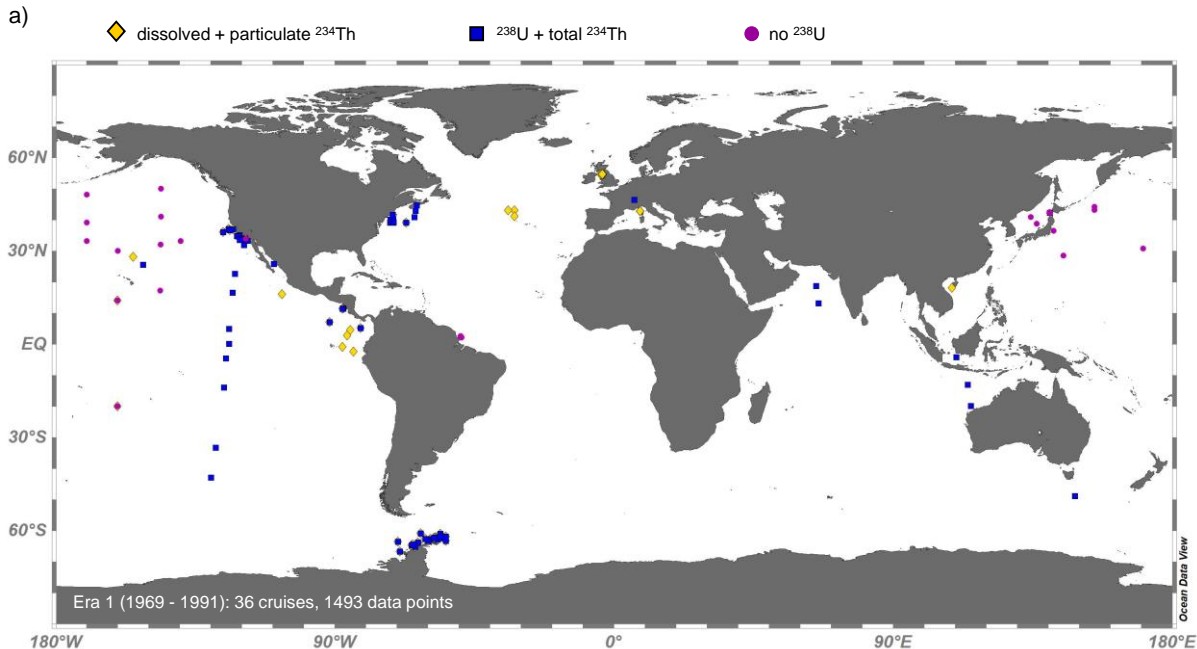

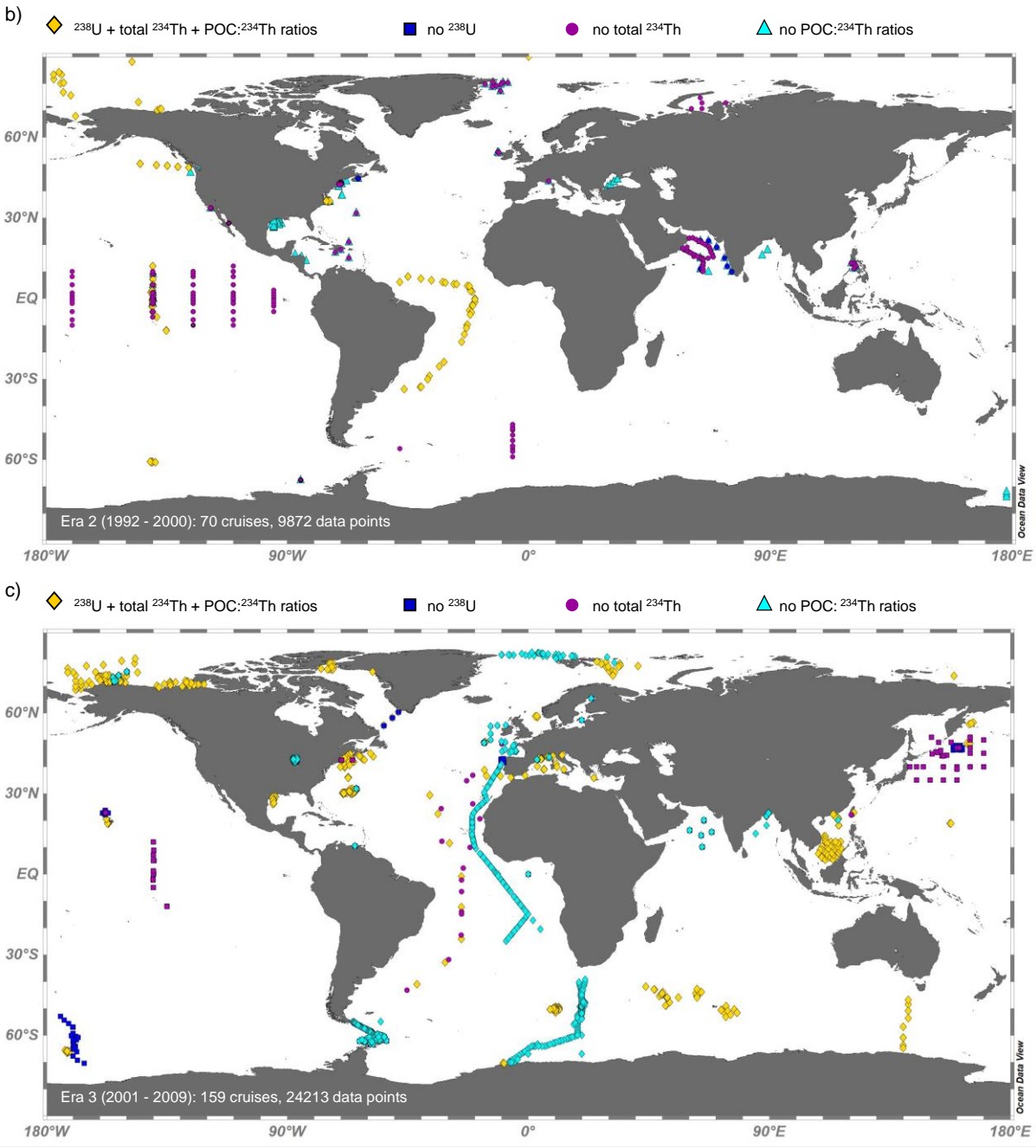

b)

◇ ²³⁸U + total ²³⁴Th + POC:²³⁴Th ratios ■ no ²³⁸U ● no total ²³⁴Th △ no POC:²³⁴Th ratios

Era 2 (1992 - 2000): 70 cruises, 9872 data points

180°W  90°W  0°  90°E  180°E

c)

◇ ²³⁸U + total ²³⁴Th + POC:²³⁴Th ratios ■ no ²³⁸U ● no total ²³⁴Th △ no POC:²³⁴Th ratios

Era 3 (2001 - 2009): 159 cruises, 24213 data points

180°W  90°W  0°  90°E  180°E

d) 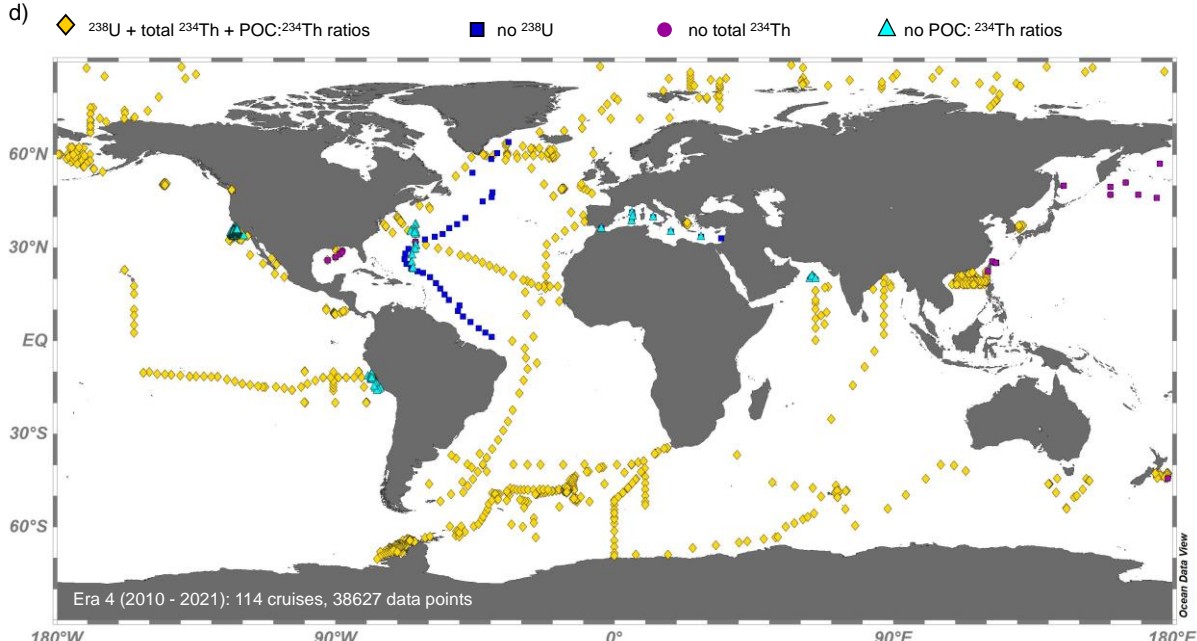

**Figure 5. Seasonal distribution of annual field expeditions with $^{234}$Th sampling reported within eras 1 (a), 2 (b), 3 (c) and 4 (d) separated by hemisphere as follows: i) Northern Hemisphere (HN), ii) Southern Hemisphere (SH), and iii) crossing the Equator and sampling in both hemispheres (both).**

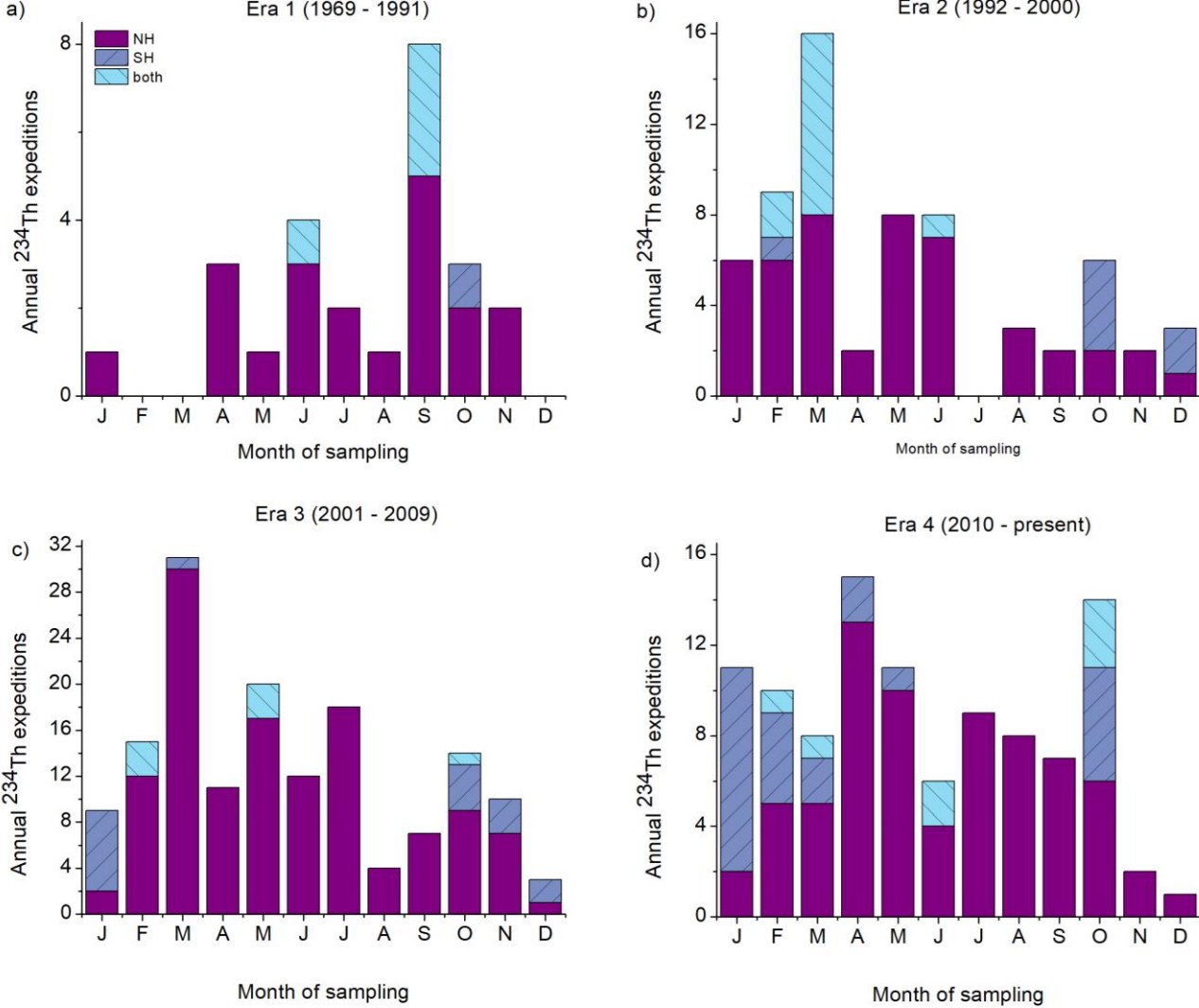

**Figure 6. Histogram of data points reported within eras 1 (a), 2 (b), 3 (c) and 4 (d) separated by data type as follows: i) $^{238}$U, ii) total $^{234}$Th, iii) dissolved $^{234}$Th, iv) particulate $^{234}$Th, v) POC:$^{234}$Th ratios and vi) PON:$^{234}$Th ratios.**

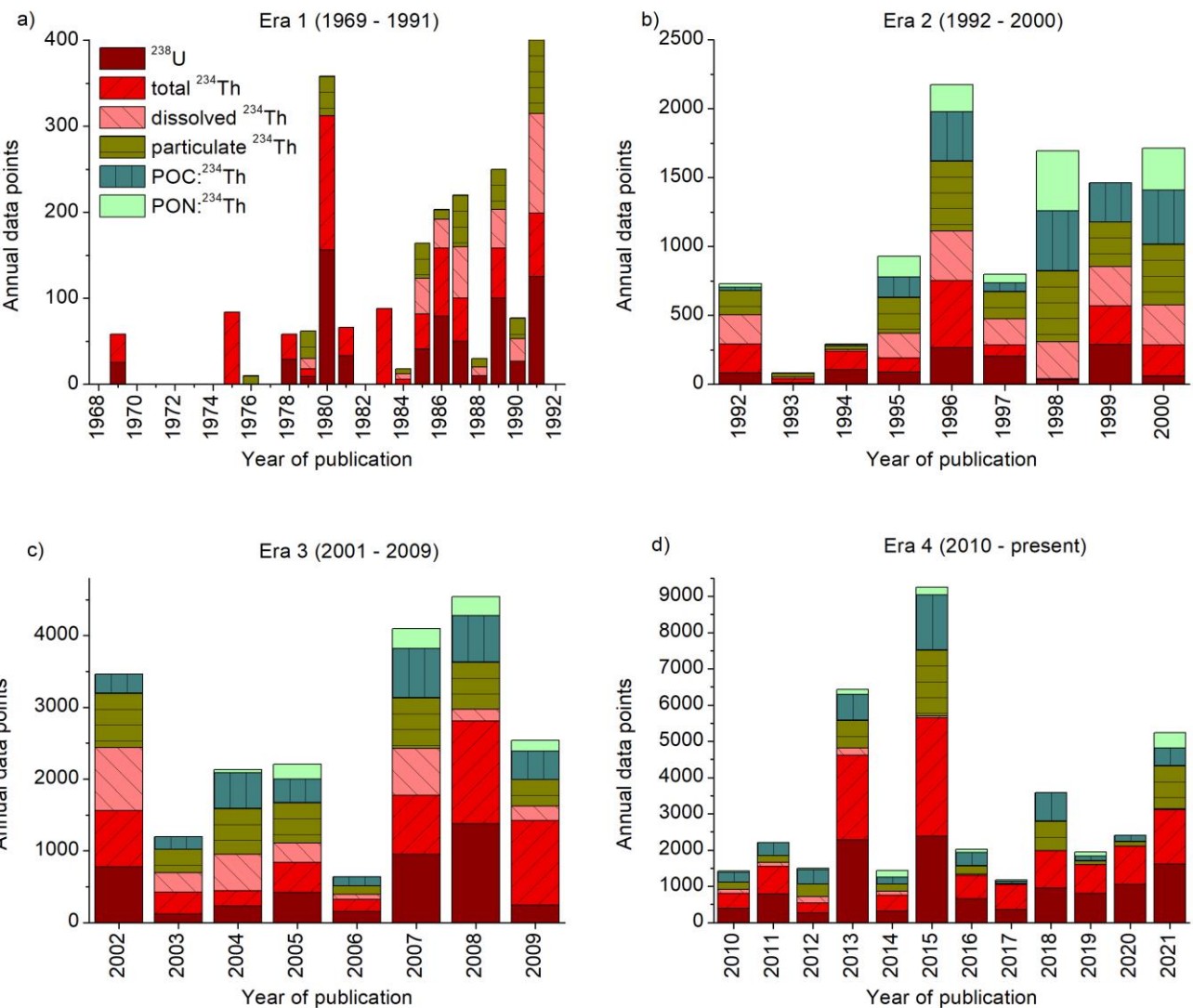

**Figure 7. (a) Histogram of data points of <sup>234</sup>Th/<sup>238</sup>U ratios at 100 ± 10 m of the water column. Ratios have been categorized as i) "deficit, ii) "equilibrium", and iii) "excess" for those the values where <sup>234</sup>Th concentrations lower, equal, and larger respectively than <sup>238</sup>U ones within a 10% of uncertainty. (b) <sup>234</sup>Th activities at the base of the regionally varying euphotic zone (defined here as the depth of the 0.1% light level) (Davidson et al., 2021). Black dotes indicate the observations near the euphotic zone that were used when producing the <sup>234</sup>Th concentration plot.**

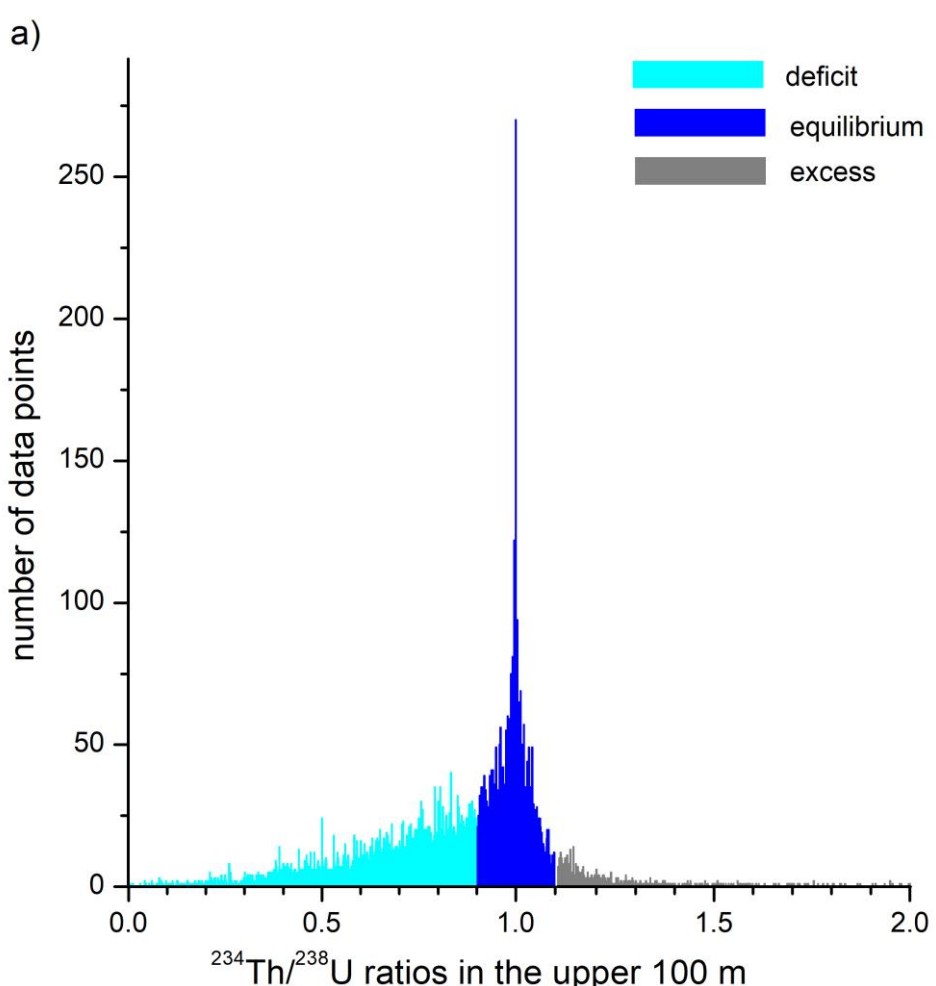

a)

b)

## Modeled $^{234}$Th concentration at the base of the euphotic zone

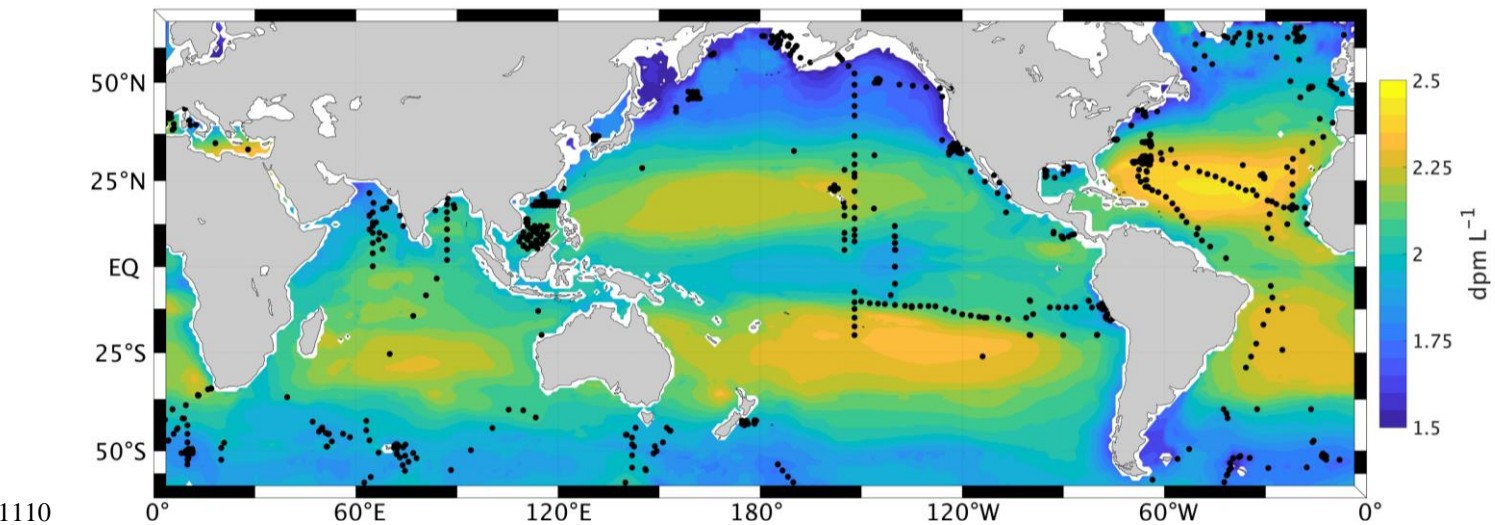

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
