# Peer review of "Revisiting five decades of 234Th data: a comprehensive global oceanic compilation."

_Earth System Science Data, 2021_

## Author Comment (AC1)

We are very thankful to all the reviewers for their thorough and constructive reviews. We have addressed all the issues raised by them and used all their suggestions and comments on the manuscript. We believe our revision improved the manuscript and we hope it pleases the three reviewers.

Specific replies to both general and specific comments for the three reviewers are given in 3 separated files file in our reply to each reviewer's comment, and in a single file here.

We would like to generally address several remarks common among the three reviewers in relation to different aspects of new parameters to be included in individual data set.

It is worth mentioning that the amount of data that could be additionally reported in the compilation is very extensive due to the wide applicability that has been shown for $^{234}$Th along the years. However, for the sake of feasibility, we needed to set boundaries to the scope of the compilation and left some parameters out of this first version of the compilation. However, we would like to emphasize the dynamic spirit behind this effort, which will be periodically reviewed and extended. Both new data and information on the availability of sampling methods used complementarily to the $^{234}$Th technique will be included in future versions of the compilation, both evaluating new useful parameters arisen from novel research and following suggestions by researchers.

All the reviewers have provided very good suggestions about parameters that must definitely be implemented in the compilation. Especially considering that these parameters are now included in the data sets of nowadays cruises and programs.

We will definitely include these parameters as improvements in future datasets, as the idea of this compilation is to make annual reeditions. PANGAEA has designed a specific structure for the compilation that allows a customized updating and we have created the webpage ("Sea of Thorium", coming soon) to stimulate the reception of new contributions.

Furthermore, in the reviewed version of the manuscript we have modified Section 5 and it includes now a subsection that it is related with the several aspects of the compilation that will be upgraded in the next edition, following the reviewers' suggestion. We acknowledge the absence of some useful parameters, such as the CTD auxiliary sensors for fluorescence and PAR pointed out by the reviewer as gaps and discuss which parameters will be included in next versions of the compilation.

**Reviewer #1**

Review of « Revisiting five decades of 234Th data: a comprehensive global oceanic compilation." By Ceballos Romero et al.

This paper presents of global database of discrete 234Th data measurements in the ocean. Following previous databases compiling Th derived POC flux and C:Th ratios, this represents the next step. No need to say that this is a very useful compilation and that such effort fits perfectly well within the scope of ESSD. Having such compilation will provide information not only on downward export rates of particles but also on processes such as remineralisation, fragmentation or even scavenging rates of Fe for instance. The data and the associated metadata are adequately compiled and presented. Pathways to data access and availability is also well described. I particularly enjoyed the historical perspectives provided in section 4. This provides a nice overview of the 234Th story since the start to nowadays.

There are other biological carbon pump processes apart the downward flux that 234Th could be used for. For instance, processes such as remineralisation, fragmentation, scavenging rates of TMs or particles sinking velocities could be estimated. I believe that such section could easily be added to the final paragraph without going into much details.

We agree with the reviewer that $^{234}$Th data are useful to study the additional processes he mentions and it's worth it adding them without increasing the manuscript length. In the reviewed version we have acknowledged these processes as potential data use perspectives (see section 6).

Apart from that, only minor issues should be addressed before publication. These are listed below:

Specific comments:

L129: How was bloom stage assessed?

We did not assess the bloom stage but instead included the information as indicated by the authors when provided. We recognise the way we assigned the four bloom stages we distinguish (i.e., "non-bloom", "pre-bloom", "bloom" and "post-bloom" conditions) was not clear in the previous version of the manuscript and has been clarified in the reviewed version (see section 2.1.1).

L141: Could CTD data including fluorescence and PAR be added?

Yes, and this is planned for future version of the compilation (see section 5 in the reviewed version of the manuscript).

It is worth mentioning that the amount of data that could be additionally reported in the compilation is very extensive, in part due to the wide applicability that has been shown for $^{234}$Th along the years. However, for the sake of feasibility, we needed to set boundaries to the scope of the compilation and left some parameters out of this first version of the compilation. However, we would like to emphasize the dynamic spirit behind this effort, which will be periodically reviewed and extended. Both new data and information on the availability of sampling methods used complementarily to the $^{234}$Th technique will be included in future versions of the compilation, both evaluating new useful parameters arisen from novel research and following suggestions by researcher.

In the reviewed version of the manuscript, we acknowledge the absence of some useful parameters such as the CTD auxiliary sensors for fluorescence and PAR pointed out by the reviewer as gaps in our compilation and discuss important parameters to be included in next versions of the compilation (see section 5).

L145: Formatting

This formatting issue (upper index for 210Po and 210Pb) has been corrected.

L169: µm dpm L-1 ????

The units have been corrected to µm dpm$^{-1}$.

L173: Why not fluorescence?

See section 5.

L208: Larger

This formatting issue ("lager") has been corrected.

L241-244: I would also stress the fact that other empirical methods such as the 15N new prod (f-ratio) techniques has been proven as unappropriated to estimate C export (Yool et al. 2007) in the late 2000's.

We thank the reviewer for pointing out this reference. The conclusion from Yool et al., (2007) has been included in the reviewed version of the manuscript (see lines 253).

L245: Section 4

The text has been corrected.

L276: # ?

The # has been erased.

L295: Most glass fiber filters (so called GF) used for particulate material analysis have a pore size of 0.7 um.

This comment has been added in the reviewed version of the manuscript.

L366: formatting

This formatting issue has been corrected.

L435: formatting

This formatting issue (upper index for 210Po and 210Pb) has been corrected.

L474: 2002 for the BSi (Friedrich and van der Loeff 2002)

This information has been included as a milestone in the corresponding era for year 2002 (i.e., era 3, section 4.3).

L544: Consider adding "….and avoid some tedious filter folding sessions."

Added (see line 605).

L610: formatting

This formatting issue (upper index for 210Po) has been corrected.

L619: GEOSECS

Corrected.

L626: cruises

Corrected.

L628: A third one is coming up very soon. I suggest adding it here if the reference is made available on time.

Added (see line 693).

L700: How about alternative export pathways (not necessarily gravitational)? In addition, fragmentation (Briggs et al. 2020) could also play a role in setting the magnitude of flux attenuation. Would Th excess be useful to study such mechanism?

The reviewer is right, $^{234}$Th can be used study additional processes to those mentioned in the previous version of the manuscript, such as fragmentation, and it is important to stress them all out. For this reason, as previously mentioned, in the reviewed version we have acknowledged these processes as potential data use perspectives in Section 6 (line 868).

L707: format

This formatting issue has been corrected.

L725: not always, see work by (Xie et al. 2020).

In the reviewed version of the manuscript section 5 has been rewritten and this part of the text has disappeared making this reference no longer suitable. But we thank the reviewer for highlighting such an interesting study (see new section 5, from line 791 to 867).

L779: formatting

This formatting issue has been corrected.

References

Briggs, N., G. Dall'Olmo, and H. Claustre. 2020. Major role of particle fragmentation in regulating biological sequestration of CO2 by the oceans. Science (80-. ). doi:10.1126/science.aay1790

Friedrich, J., and M. M. R. van der Loeff. 2002. A two-tracer (Po-210-Th-234) approach to distinguish organic carbon and biogenic silica export flux in the Antarctic Circumpolar Current. Deep. Res. I 49: 101–120.

Xie, R. C., F. A. C. Le Moigne, I. Rapp, J. Lüdke, B. Gasser, M. Dengler, V. Liebetrau, and E. P. Achterberg. 2020. Effects of 238U variability and physical transport on water column 234Th downward fluxes in the coastal upwelling system off Peru. Biogeosciences 17. doi:10.5194/bg-17-4919-2020

Yool, A., A. P. Martin, C. Fernandez, and D. R. Clark. 2007. The significance of nitrification for oceanic new production. Nature 447: 999–1002. doi:10.1038/nature05885

**Reviewer #2**

**Review of "Revisiting five decades of [234]Th data: a comprehensive global oceanic compilation" by Ceballos-Romero et al.**

**General comments**

This paper provides a global dataset on [234]Th data in the ocean, accompanied by additional parameters used in the application of the [234]Th approach to derive export fluxes, mainly particulate carbon export fluxes. This effort comes after the compilation of [234]Th fluxes by Le Moigne et al. (2013) and the recent compilation of POC:[234]Th ratios by Puigcorbé et al. (2020), both published in ESSD, therefore I believe it is well suited within the scope of ESSD. This compilation will be very useful, as the authors mention, to better constrain the contemporary ocean carbon uptake by providing easy access to all the [234]Th data collected over the last 50 years.

This is a major effort that has brought together a large number of datasets. The information provided regarding the global dataset, what parameters are included, how they are organized and how to access them is clear for the reader. Additionally, the authors have added a detailed history of [234]Th and its use as particle tracer from its beginnings until today, and even looking into future programs.

I did enjoy reading the entire manuscript, but I also believe it should be streamlined because there is a bit of repetition between sections and whitin the same section. As a general comment, there were several typos and formatting errors along the manuscript, repeated words and sometimes sentences appeared out of the blue or were unfinished/misplaced. I had the feeling that the manuscript needed an additional proof reading by the authors prior to submission, but I tried to provide detailed comments when I spotted some of these issues.

We are very grateful to the reviewer for its thorough review, attending both formatting issues and the manuscript content itself. We apologize for all the formatting issues and typos that were overlooked in our previous version. We understand that such compilation should be perfectly edited. We employed time and effort to do that, including more than four people in the task. However, we see now, that our revision was far from perfect. That is why we specially appreciate the effort made by the reviewers to polish text, Figures and compilation itself. We had made an additional effort this time to improve the manuscript and the content.

We strongly believe the manuscript is greatly improved after implementing all the reviewer's suggestions.

Particular to Section 5, this section does not always discuss the gaps in the dataset, but it expands on issues that, although very important, most reader will be aware of them and they are not real gaps of the dataset, at least not in the way they are presented. To be honest, this section feels like it is a section to "advertise" the work of the first author (7 citations out of 8 in page 24 are Ceballos-Romero et al.), which it is actually really good, but I don't think it belongs to the "gaps of the dataset" section. Maybe move it to the overview of the dataset or to the discussion. If the authors decide to keep it in Section 5, as mentioned in my detailed comments, those particular paragraphs should be more to the point.

We agree with the reviewer and, as so, the entire section 5 has been rewritten given it a completely different approach more aligned with the take home message of the manuscript (see new section 5, from line 791 to 867).

The [234]Th timeline part (4. Discussion) would largely benefit from a figure showing a timeline and the key dates/milestones listed in the text. I think the readers and future researchers interested in [234]Th would appreciate having Section 4 summarized in a nice timeline figure.

We consider including this figure a very good idea and, as so, it has been implemented in the reviewed version of the manuscript (see new figure 3).

Also related to Section 4, I have provided specific comments below, but I believe that, even though this manuscript has a whole section of "storytelling", without providing additional data (which I enjoyed reading), the division by era's should provide some useful information for scientific purposes (e.g., period of years that the era lasted, using the name of the main program of the era, or stating what was done during such era). Although it is quite "artistic" to have named them by well-known novels linked to the ocean, those titles don't serve a clear purpose (they don't clearly describe the actual era) and changing their names might be more adequate when publishing a research paper.

We appreciate the reviewers' comment on this regard. Changes have been implemented, including the duration of each specific era and a description for the main theme that characterized it. We have kept name of the eras after titles from seminal ocean-related books, but included a quote from each one of them to evince each one of the choices made for the era, which follow an intention are not random.

Moreover, in that line, I think the manuscript, since it is a research publication, should provide some actual data, not just referring to the number of publications and number of datapoint, but actual data on $^{234}$Th, similar to what was done in the two previous compilations related to $^{234}$Th (see Fig. 1, 3, 7 and Table 2 from Le Moigne et al. (2013) or Fig. 2, 3 and 5 from Puigcorbé et al. (2002)). This manuscript does not provide data on $^{234}$Th, just on the number of studies/cruises/stations/parameters. The authors could include a table with $^{234}$Th concentrations (tot/part/diss) similar to Table 2 from Le Moigne et al. (2013) or in a figure similar to Fig. 4 from Puigcorbé et al (2020) or something completly different, but they should report actual $^{234}$Th data/values, not just data points. A description of the number of datasets, studies and locations is already provided in the abstract of the dataset in the PANGAEA repository. In this manuscript, the authors should provide something additional to it, that goes beyond explaining the structure of the dataset, the percentage of studies that do one thing or another, or the history of $^{234}$Th.

We agree with the reviewer that the previous version of the manicurist lack data. Accordingly, we have included 2 new figures with data in section 6 (see line868).

Despite agreeing on the fact that this dataset is very necessary, and it will be very useful, I have two (relatively major) issues with it:

- Not all the studies reported analyzed POC, some analyzed PIC+POC. There is no reference made regarding this "problem". Either in the manuscript or in the dataset (or both), a comment should be made to indicate if the data reported is truly POC, or if it is not, why it is still considered as POC (e.g., negligible amount of PIC).

Only data reported as POC in the sources consulted have been reported as POC data. And equivalent for PIC data.

- The authors have compiled additional data, when available, on water temperature, salinity, $^{238}$U, and particulate nitrogen, and if they can, they also report the bloom stage. However, the authors have omitted a critical parameter for the application of the $^{234}$Th method. This is particularly surprising since one of the authors has recently (2020) published a PNAS paper entitled "*Metrics that matter for assessing the ocean biological carbon pump*". It has been shown that the integration depth has an impact on the magnitude of the fluxes (the strength of the biological carbon pump) and therefore, we should be using reference depths related to the euphotic zone (or primary production zone (PPZ); Owens et al. (2015)). I understand that not all the studies, particularly in earlier days, provided this information, but recently, the integration depth is no longer fixed and authors explain what kind of reference depth they use to integrate the fluxes. I believe fluorescence and/or PAR data to define the euphotic zone or the PPZ (or providing those depths directly) should be added in the dataset, the same way that temperature and salinity are. With that addition, in the manuscript it should also discuss the importance of the integration depth and cite the Buesseler et al. (2020b) paper in a more appropiate line than where it is currently cited (L778). If this dataset is expected to

be "alive" and authors want to be contacted to update it by adding new data, having this kind of input is crucial for future applications of the [234]Th method.

The reviewer is right about this gap in the compilation, integration depth is proved to be a crucial parameter for an accurate use of [234]Th disequilibrium to evaluate POC export. However, only very recent papers acknowledge this issue and include details of the integration depth. On the other hand, as it is stated in the PNAS paper "Metrics that matter for assessing the ocean biological carbon pump", the choice of the integration depth is not straightforward. This way we do not think we should select one specific integration depth for each data set, as that will imply that we are interfering in the interpretation of the data. To this regard, the inclusion of the fluorescence and/or PAR data, as suggested by the reviewers will allow future users of the compilation to estimate the integration depth. We mention in Section 5 that CTD auxiliary sensors for fluorescence and PAR will be some of the parameters to be included in next versions of the compilation. This way, we expect that this will also serve the users of the compilation as a reminder of one important parameter to include together with the [234]Th results.

Finally, regarding the datasets available in PANGAEA, I found that https://doi.pangaea.de/10.1594/PANGAEA.918125 provides the metadata information and there is a table with parameters shown in that file. However, https://doi.pangaea.de/10.1594/PANGAEA.937414 contains the actual data but there is no table of parameters there. Please consider including such table for clarity (even if it is a long one), since not all the "parameters" might be clear to the reader/user, e.g., "integrated_depth_real_particle (m)". Notice my comment below for Table 1, where the starting and end date for the sampling period have been reversed in some cases. The dates are also incorrect in the dataset. I noticed some from Table 1 but please double check them all again in the dataset as well.

We thank the reviewer for his comments. We have already made this request to PANAGEA for the Table referred to by the reviewer also be included in https://doi.pangaea.de/10.1594/PANGAEA.937414.

**Detailed comments:**

L11-16: "However, it was…. the biological pump." This is a very long sentence. I don't think that level of detail is necessary for the abstract. The information detailed here is already discussed in detail in the manuscript. Please, simplify it.

In the reviewed version of the manuscript this sentence has been simplified (see abstract).

L 20-21: "…that include two size classes;…". By reading the rest of the paper more por sizes (e.g., <70 μm) are mentioned and reported. Maybe just finish that sentence after "filtration method" in the abstract.

Done.

L23: "…included, including…". Try to find another word to avoid repetition

Done.

L38-39: I understand where that is coming from but production rates of anthropogenic radionuclides might not always be that well characterized or can vary over time.

We agree with the reviewer that this clarification is important and in the reviewed version of the manuscript we have solved this issue by specifying that it is the case for naturally occurring radioactive elements (see line 35).

L41: I would even say more than just years (paleo tracers).

In the reviewed version of the manuscript, we have replaced years by millenniums (see line 38).

L44: "use" instead of "used"

Corrected.

L45: Delete "a" from "a $^{234}$Th"

Done.

L53: "when high scavenging by particles". Correct, there is something missing.

Corrected.

L55-56: Instead of citing that many papers without a clear reason behind their selection, maybe cite Le Moigne et al. 2013, which provides a compilation of studies using the $^{234}$Th approach to derive estimates of POC export.

Done.

L57-58: PIC and BSi abbreviations are not needed. PIC is only used one more time, in L 474, where it is also defined, and BSi is not used anywhere else in the manuscript.

PIC and BSi abbreviations have been removed.

L61: "…is in part…" seems to imply that there is another part, but it is never mentioned.

Corrected.

L63-64: Not sure how necessary is to provide the editors' names for the special issue.

Editors named have been erased (see line 59).

L86: Maybe delete the "only"

Deleted.

L88: Missing Th after $^{234}$

Corrected.

L93: "..and useful metadata to be used". Change words to avoid repetition of "use…"

Done.

L114: Add the GEOTRACES website

Done.

L124: "…spanning all oceanic regions" maybe add "across": "spanning across all oceanic regions"

Added.

L129: Do you mean when reported by the original authors? There could be issues on the way the different authors, over the different years, have decided if the conditions where non-bloom, pre-bloom, bloom or post-bloom. Please discuss it a bit more.

We have compiled all data as reported by the original authors, except for conversion of units when necessary for the sake of consistency between all datasets. We did not assess the bloom stage but instead included the information as indicated by the original authors when provided. This information has been stressed out in the reviewed version of the manuscript. We recognise the way we assigned the four bloom stages we distinguish (i.e., "non-bloom", "pre-bloom", "bloom" and "post-bloom" conditions) was not clear in the previous version of the manuscript and has been clarified in the reviewed version (see lines 119-129).

Additionally, we have acknowledged the lack of evaluation of the bloom stage at the sampling moment for each of the studies compiled following a standard criterion as a gap in the compilation and proposed such an improvement for future version of the compilation (see new section 5).

L145: 210 should be a superscript. Same for L435, L537, L610, L665 and L666

Corrected throughout the manuscript.

L158-159: "…the authors were directly contacted for data is indicated in the localization." Please rephrase for clarity.

Done.

L169: It should be µmol instead of µm. Same for L178

Corrected.

L169-170: "…and include… in which we included". Try to find synonyms to avoid repetition of words

Modified.

L174: Correct "The particles' sampling…"

Corrected.

L180: "Measurement uncertainties in the data points as provided by the data measurer…". Please rephrase for clarity.

Corrected.

L200-203: Delete or move to the section "General overview", rephrase or delete "here" since you refer to "here" for the entire manuscript, for a particular section that you are discussing and for a particular section that you are about to discuss, which makes it confusing.

Done.

L203: "on in" correct it

Corrected.

L208: "larger" instead of "lager"

Corrected.

L209: "During these 50 years of study, we have….". Maybe rephrase so it doesn't sound like you took 50 year to write this manuscript. Something like "During these 50 years of $^{234}$Th data" similar to the title.

Corrected.

L209 and L211: "14 major ocean programs" vs "as part of one of these 13 programs"

14 has been corrected to 13 (see line ).

L211: "as reported by the authors". I'm not sure which authors are you referring to, the ones of that major program? Yourselves? If it is yourselves there is no need to have that sentence. If it is referring to the authors of those datasets, I don't think it is correct, because they could not tell that their dataset represented 30% of the total datasets.

The reviewer is right, that sentence wasn't correct and has been erased (see line 217).

L212: "reported" instead of "compiled"

Corrected.

L214: delete the - after TSS

Done.

L218: "time-series stations" should be TSS as defined in L214.

Corrected.

L222: "least" instead of "last"

Corrected.

L222-223: "re-occupied" should be "reoccupied", as in L223

Corrected.

L225-231: Missing the degree (°) symbol for the coordinates of all the time series stations

Corrected throughout the manuscript.

L237: POC(PON) to $^{234}$Th ratios have been defined previously (e.g., 144) as POC(PON):$^{234}$Th.

Corrected.

L247: Correct "datasets"

Corrected.

L253: Correct "expeditions to the"

Corrected.

L258: "is not significant of the number". Rephrase for clarity.

Done.

L263: Correct "experiments"

Corrected.

L264: Delete "started in 1969" or add "which"

Added.

L271: it should be "of" instead of "on"

Corrected.

L276: Delete #

Done.

L288: Same as previous comment from L237

Corrected.

L301: it should be "sample" instead of "sampled"

Corrected.

L304: "yet maximize…" add "the number of locations sampled"

Added.

L305: "were" instead of "was"

Corrected.

L308: Correct for consistency "POC(PON):$^{234}$Th"

Corrected.

L317: "…in situ pumps, bottles and…" remove the "and" and change for a comma.

Corrected.

L324-326: Missing reference to "Not et al (2012). Conservative behavior of uranium vs. salinity in Arctic sea ice and brine. Marine Chemistry 130-131; pp. 33–39. doi:10.1016/j.marchem.2011.12.005

Reference added and text modified accordingly (see lines 344).

L337: "...in an oceanographic expedition…", delete "field" is redundant.

Done.

L340: I would recommend changing the names of the eras in order to actually provide some information about each era. Naming era 1 "The Old Man and the Sea" doesn't say much about what happened during this era. I would provide the periods of each era and, if possible, something more such "JGOFS", "GEOTRACES" or

something related to the main advances of each era, e.g., "the beginning of $^{234}$Th as tracer of POC export" or in that line.

We appreciate the reviewers' comment on this regard. Changes have been implemented, including the duration of each specific era and a description for the main theme that characterized it. In the reviewed version of the manuscript, we have kept name of the eras after titles from seminal ocean-related books, but included a quote from each one of them to evince each one of the choices made for the era, which follow an intention are not random.

L344: Bhat had co-authors in that first $^{234}$Th publication.

Corrected.

L345-347: "Bhat et al. (1969)… part of the total $^{234}$Th." What's the purpose of this sentence? It is interesting but it appears out of the blue and it is not discussed any further in a way that it is linked to the following sentences. Maybe rephrase between L345 and L354 to facilitate the flow.

Rephrased (see lines 370-380).

L353: It sounds strange to say that you use "this radionuclide" (referring to $^{234}$Th) to study the scavenging of thorium. All the thorium isotopes are particle-reactive.

Radionuclide replaced by radioisotope.).

L358: Rephrase "realizing the relevance to the downward $^{234}$Th flux" for clarity.

Rephrase.

L367: Correct "increase" instead of "increased"

Corrected.

L371: Correct "prolong" instead of "prologue"

Corrected.

L376: It should be "$MnO_2$" (the 2 should be a subscript)

Corrected.

L380: Linked to my comment from L340, here the authors refer to JGOFS era. Avoid confusions.

All references on the manuscript to the JGOFS era (which were 3 in total in the previous version of the manuscript) has been erased in the reviewed version of the manuscript for the sake of avoiding confusion with the eras that we have defined ion this paper for the $^{234}$Th timeline.

L382-385: Missing the degree (°) symbol for the coordinates of all the time series stations

This issue has been corrected throughout the manuscript.

L388-389: Avoid repeating "introduced"

Changed.

L391: "…a precipitate that serves for removing Th…". Change to "…a precipitate that removes Th…"

Changed.

L393: Maybe refer to it as "For the $MnO_2$ cartridges technique", since $MnO_2$ is the method being used currently, although mainly without the cartridges.

Done.

L394: Same comment as L376

Corrected.

L394: Delete "(MnA and MnB)". There is no graph and, therefore, there is no need to specify the name of the cartridges (which could also be "named" Mn-I and Mn-II or Mn-1 and Mn-2 or many other ways).

Deleted.

L394: Did you mean "and gamma count them"? Not sure it is needed here since gamma counting is mentioned in L398.

Text erased.

L397: Why the differentiation between $MnO_2$ and Mn cartridges? We have only been talking about cartridges with fibers impregnated with $MnO_2$.

The reviewer is right, such a differentiation between $MnO_2$ and Mn cartridges is pointless. Text has been modified to always refer to $MnO_2$ cartridges for consistency (see lines 424-436).

L398: Refer to it as $MnO_2$ cartridges for consistency. Otherwise, modify the text above and refer to Mn cartridges the whole time, after specifying that those are cartridges with fibers impregnated with $MnO_2$.

Text has been modified to always refer to $MnO_2$ cartridges for consistency (see lines 424-436).

L403: The reference to the volumes sampled doesn't seem to have a purposed. It is not discussed or linked in any way to the rest of the paragraph. It provides information but I don't think it is relevant as it is presented right now, plus the sample volume reference is not done for the studies cited between L404-410.

The sentence refereed to volumes sampled has been erased.

L407: Correct "where" instead of "with" or say, "with a predominant particulate size class cut-off of 0.45 µm". If it is just for one study, can you say it is the predominant size class? Were there others in that same study?

0.45 µm was the predominant pore size cut-off for the majority of the studies that carried out particulate analyses. This sentence has been rephrased to clarify this point following reviewer comment on L407 and L408-410 (see line 444).

L408-410: "Note that… McKee et al., 1986)." This sentence seems not to match the size class cut-off of 0.45 µm mentioned just before. Is this operational distinction applying to the previous studies cited between L403 and L405? If that's the case, move it there. If not, remove or integrate better to the paragraph.

See response to comment on L407.

L435: Same comment as L145: 210 should be a superscript

This issue has been corrected throughout the manuscript.

L436: Same comment as L340

Same response as before. We appreciate the reviewers' comment on this regard. Changes have been implemented, including the duration of each specific era and a description for the main theme that characterized it. In the reviewed version of the manuscript, we have kept name of the eras after titles from seminal ocean-related books but included a quote from each one of them to evince each one of the choices made for the era, which follow an intention are not random.

L437: The authors keep referring to the JGOFS era: "…what we are calling the "JGOFS era" (1989-2003)" but this is not an actual era in their manuscript, not by period of time nor by name. Please avoid confusion by naming the eras appropriately to refer to something that has a meaning linked to the history of $^{234}$Th.

All references on the manuscript to the JGOFS era (which were 3 in total in the previous version of the manuscript) has been erased in the reviewed version of the manuscript for the sake of avoiding confusion with the eras that we have defined ion this paper for the $^{234}$Th timeline.

L443-444: It is odd to cite a reference from 2008 in the 2$^{nd}$ era (1992-2001). Moreover, the following sentence starts with "Accordingly, …". Please cite the first paper discussing this topic, not one from 2008.

Done.

L446 or 447-448: Adapt the format of equation 1 to match the format of the different parameters in the description (e.g, add 234 as a superscript, keep consistency with italics and non-italics, etc.)

Done.

L477: Add the units of the $^{234}$Th$_{flux}$ as done for the ratio.

Done.

L452: It should be "Element: $^{234}$Th" instead of "POC:element". I don't think the italics are needed.

Corrected and italics removed.

L453: "its" should be "their"

Corrected.

L453: Delete "dependent" since it already says that they vary with both depth and particle size. Or say that they are "depth and particle size dependent"

Done.

L454-456: "…can be calculated from the disequilibrium between the $^{238}$U-$^{234}$Th resulted from particle sinking…" This has already been said a couple of times before. Simplify: "$^{234}$Th flux can be calculated by evaluating…."

Done.

L456: Why "of the dissolved $^{234}$Th"? It has been done with total $^{234}$Th (see Savoye et al. 2006-Marine Chemistry and references in there).

The reviewer is right. This clarification to dissolved $^{234}$Th is wrong and has been removed.

L458: Add "a": "only a single $^{234}$Th profile"

Done.

L458: Remove the comma after "NSS model"

Done.

L458-460: Simplify it. Too many sentences within one sentence. "…when activity gradients can be assessed in time thanks to repeated sampling, ideally over the course of 2 to 4 weeks."

I think the authors should refer to Resplandy et al. (2012)-DSR-I work (http://dx.doi.org/10.1016/j.dsr.2012.05.015 )

Done.

L473: PIC abbreviation is not required since it isn't used anymore. Also, notice that PIC and POC fluxes are elemental fluxes of the same element, C.

Abbreviation erased and text corrected.

L476: It should be "$MnO_2$" (the 2 should be a subscript)

Corrected.

L481: Missing the degree (°) symbol for the coordinates of the time series station

This issue has been corrected throughout the manuscript.

L487: "with large volumes (2500-3500 L) passing…" instead of "passed"

Corrected.

L498: Why "we initiated"? Delete

Deleted.

L489: Why "and filtered"? Delete

Deleted.

L492-494: There are 7 studies that sampled 2 size fractions but there are only 4 papers cited, 2 for 53 μm screens and 2 for 70 μm screens. Either cite the 3 remaining studies or add "e.g.," when citing the other studies.

Corrected: all the references are indicated now (see lines 535-542).

L494: How many studies used sediment traps? 7 studies did size fractionation, it would be good to mention those that used sediment traps, for consistency.

This information and references have been added (see line 540).

L503: Another reference to the "JGOFS era". Maybe consider this as one of the eras then?

All references on the manuscript to the JGOFS era (which were 3 in total in the previous version of the manuscript) has been erased in the reviewed version of the manuscript for the sake of avoiding confusion with the eras that we have defined ion this paper for the $^{234}$Th timeline.

Note that we do not consider JGOFS as one of the eras in the manuscript because we are discussing the history of $^{234}$Th, not the history of oceanography, and JGOFS was not conceived at all to focus on the study of $^{234}$Th. It simply happened as part of one JGOFS cruise.

L505: "at" instead of "in"

Corrected.

L506: Based the era period described by the authors (1992-2001), the NABE study should not be included here. It's not part of the overlap, it's overlapping with era 1, but not with era 2.

We have chosen as a criterion to distinguish eras seminar publications related to 234$^{Th}$ that changed the course of the technique. In this part of the text, we are presenting the papers published during era 2, which is based on the criteria just mentioned, to which the year in which sampling took place is indifferent. We agree with the reviewer that this point was not clearly explained in the previous version of the manuscript, and it seemed we were mixing events from different eras. In the reviewed version of the manuscript, we have modified the text in all the paragraphs were presenting publications versus year of sampling to make clear this point and avoid any confusion in this regard.

L506-520: There is no consistency in the amount and type of information provided for each process study. For example, why the objective of the EqPac study is described but not for the other studies, for which mainly the dates and locations are provided?

Information provided for each process study has been modified (shortened or extended depending on the case) for the sake of consistency process studies.

L523: "initiatives" should be singular (only OMEX included here).

Done.

L525: "carried out" instead of "made"

Done.

L532-536: "Time-series…….Murray, (1992))". Cut the sentence, it's too long.

Done.

L535-536: 10 studies used sediment trap and also $^{210}$Pb-$^{210}$Po at the same time (same studies for both)? If not, rephrase for clarity

Rephrased.

L536: Same comment as L145: 210 should be a superscript.

This issue has been corrected throughout the manuscript

L538: Same comment as L340

Same response as before. We appreciate the reviewers' comment on this regard. Changes have been implemented, including the duration of each specific era and a description for the main theme that characterized it. In the reviewed version of the manuscript, we have kept name of the eras after titles from seminal ocean-related books but included a quote from each one of them to evince each one of the choices made for the era, which follow an intention are not random.

L543-544 and L551-552: Same sentences. Delete one to avoid repetition or change them to say different things.

Deleted.

L556-557: Missing the degree (°) symbol for the coordinates of the PAP station

Corrected.

L557-558: Why there is so much detail provided for the PAP station and not for the others? Why it is not explained what sensors the other TSS have? Provide similar levels of detail for all.

The reviewer is right, all TSS should be treated equal. We do not consider describing all the TSS in such a hight level of details falls within the scope of the manuscript and there, the detailed information for the PAP site station has been removed in the reviewed version of the manuscript been removed (see line 618).

L560-564: I'm not sure the level of detail provided here is necessary. Simplify.

Done.

L569-571: "An additional improvement… method." This is out of place, this should be moved and discussed in the following era since it was published in 2021.

This text has been moved to era 4 (see lines 716-719).

L573: I guess "contraptions" is supposed to be "concentrations"

Corrected.

L573: Delete "and" after "POC:$^{234}$Th"

Corrected.

L578: What do you mean by "surveys"? Cruises, programs? It can't be studies because those 4 "surveys" are differentiated from the 87 studies. Please clarify

We meant cruises. This part of the text has been deleted in the reviewed version of the manuscript for the sake of clarity.

L580: Are HOT, BATS and DYFAMED considered programs? In table 2 they are referred as TSS. The program that initiated these time series was JGOFS. Maybe refer to them as TSS?

Done.

L584: Missing the degree (°) symbol for the coordinates of the K stations

Corrected.

L586: Correct "JGOFS"

Done.

L587: U.S. JGOFS-Japan refers to a collaborative station between US and Japan?

U.S. shouldn't be there and has been erased.

L587: Missing the degree (°) symbol for the coordinates of the KNOT station

Corrected.

L590: Rephrase: "And a pilot cruise was reported under the GEOTRACES program" or similar

Rephrased.

L590: Correct "…discussed in the next section"

Corrected.

L592: "Finally, a great number of projects and experiments belong to this era (see Table S3 for a chronological summary)." Remove, starting the sentence with "finally" creates great expectations for the reader, but there nothing after that.

Done.

L593: "Sampling"

Done.

L601: "while and" is incorrect. Please choose which one to use.

Done.

L608: Change "used" for "use"

Done.

L610: 210 should be a superscript

Corrected.

L612: Same comment as for L340

Same response as before. We appreciate the reviewers' comment on this regard. Changes have been implemented, including the duration of each specific era and a description for the main theme that characterized it. In the reviewed version of the manuscript, we have kept name of the eras after titles from seminal ocean-related books but included a quote from each one of them to evince each one of the choices made for the era, which follow an intention are not random.

L619: It should be "GEOSECS"

Corrected.

L624: "carried out two years later". Later from which reference year? Just say the year that it took place.

Done.

L626: Correct "cruises"

Corrected.

L629-632: Maybe some of the studies could be cited, including Clevenger et al., (2021) cited in the previous era, or the GEOTRACES intermediate data products from 2014 and 2017 (Mawji et al., (2015) and Schlitzer et al. (2018)) or the trace metal fluxes studies (Black et al., 2018; maybe add some more studies here since you refer to a shift from POC to trace metal fluxes).

New references have been added to this part of the text (see lines 696).

L638: "and known" should be "are known"

Corrected.

L643-645: The 8 cruises that sampled in the NH and the SH are included in the 27 that sampled in the SH or are considered a separate group. If so, maybe mention them apart from the NH cruises.

The text has been rephrased to clarify this issue (see lines 724-727).

L646: Refer to GEOTRACES instead of "a major ocean program".

GEOTRACES is not the only major program under which expeditions took place within era 4 (as shown in Table 2 to which we refer the reader to after our statement). Therefore, this change suggested by the reviewer has not been implemented. Nonetheless, the text has been corrected to avoid confusion.

L676: Add "and" in between "2$^{nd}$ 3$^{rd}$"

Done (see line 730).

L646-651: Why are this projects/experiments/cruises not cited in Table 2? What criteria was used to include them or not in that table?

The compilation of program included in Table 2 represent a selection of selected programs, chosen in terms of number of cruises achieved, duration of the program, relevance of results throughout time, etc. And overall, a global representation of programs from all countries have been followed. HilATS have been deleted from the Table and from the "selected ocean program" in the reviewed version of the manuscript.

L659: Change to "...whether a SS, a NSS or both models were used"

Done.

L660-661: Are those citations referring to what? The use of SS, NSS or both? Why so many citations? Just write "e.g.," and cite max 3.

References now added for each approach including a maximum of 3 (see lines 743-753).

L665 and L666: Same comment as L145: 210 should be a superscript.

Done.

L666: Correct "compared them"

Corrected.

L668: Same comment as L340

Same response as before. We appreciate the reviewers' comment on this regard. Changes have been implemented, including the duration of each specific era and a description for the main theme that characterized it. In the reviewed version of the manuscript, we have kept name of the eras after titles from seminal ocean-related books but included a quote from each one of them to evince each one of the choices made for the era, which follow an intention are not random.

L675: the abbreviation for net primary production is not needed since it is not used anymore

Abbreviation removed.

L677-678: "availability for phytoplankton at a key junction of the global ocean circulation". Rephrase for clarity

Rephrased.

L684: BCP has not been defined yet. It is defined in L697. Abbreviation probably not needed, used a few more times later on but there is a large number of abbreviations in this manuscript already.

We agree with the reviewer that the are many abbreviations in the manuscript, but we consider the target audience of this manuscript are well familiarised with the BCP abbreviation for the Biological Carbon Pump and have decided to keep it. Nonetheless, abbreviation has been defined when it corresponds in the first use of the concept (see line 30), not where it was in the previous version of the manuscript.

L685-686: Why APERO and PICCOLO are no included in the previous group of projects? What is the criteria to differentiate them from COMICS, EXPORTS, CUSTARD, etc.

APERO is in a list apart because is a planned initiative to be started in future years, as indicated in the manuscript. This difference has been stressed out in the reviewed version of the manuscript to avoid confusion. PICCOLO was misplaced in the previous version of the manuscript and has been moved to the list with the rest of projects, as corresponds to an initiative already started.

L689-691: Please clarify what JETZON does. "…international coordinating umbrella to serve as a focal point…" means that they organize or fund cruises, or meetings? Are any of the recent programs cited before (COMICS, EXPORTS, CUSTARD, OTZ, SOLACE, APERO, PICCOLO) included or linked to JETZON in a formal way?

JETZON is a UN Ocean Decade Programme, acting as an international coordinator and focal point for Twilight Zone studies. International projects such as COMICS, EXPORTS, CUSTARD, OTZ, SOLACE, APERO, PICCOLO, etc. are linked to JETZON, that acts a meeting point to share data, results or discussions and organize seminars, among other things. The link is robust although it is not formal. JETZON description is improved in the manuscript (see lines 781-785).

L697: See my comment for L684

We consider the target audience of this manuscript are well familiarised with the BCP abbreviation for the Biological Carbon Pump and have decided to keep it. Nonetheless, abbreviation has been defined when it corresponds in the first use of the concept (see line 30), not where it was in the previous version of the manuscript.

L697: Delete "of"

Deleted.

L700: "time-series site" should be TSS

Corrected.

L707: Use "< 1%" instead of "less than 1%" (if not add a space after "than")

Space added (see line 805).

L718: Correct "for the application"

Corrected.

L726: Change "…since most cruises do not allow repeated measurements…"

Corrected.

L727: Delete "i.e., neglecting changes….". It has already been explained what a SS model implies.

Corrected.

L718-731: This paragraph seems repetitive with previous sections of the manuscript, when presenting SS and NSS models. This section should describe how many studies applied SS and NSS or both and in which cases the application of one model or the other had a significant impact. Maybe citing the work by Resplandy et al. (2012). http://dx.doi.org/10.1016/j.dsr.2012.05.015 can help to provide the overview on this particular gap.

We agree with the review that the approach given to this entire section in the previous version of the manuscript was not the most right one. Accordingly, Section 5 (affecting these paragraphs referred to by the reviewer) has been completely rewritten in the reviewed version of the manuscript to address this and some other comments by both this reviewer and reviewer 3 (see new section 5, lines 791-866).

L732-733: Too broad. Not all the radionuclides present deficits in the water column and not just because of their half-lives, also because of their physicochemical behavior. Please rephrase referring only to $^{234}$Th, since it is the radionuclide of interest for this manuscript.

Done.

L736: Add "e.g," in front of the citation to Ceballos-Romero et al. (2018), there are other studies that have reported similar results.

Added (see lines 817).

L736-744: This paragraph is not referring to a gap in the dataset. This is an explanation about the importance of the bloom period and the reliability of the SS and NSS approaches (issues that the readers accessing this dataset are most probably already aware of). I don't think it is needed, plus it seems a bit repetitive and too extensive considering that an explanation about SS and NSS already stared in L718. Please simplify from L718 to L744 in order to highlight the importance of this considerations and their implications in a much briefly way.

We agree with the review that the approach given to this entire section in the previous version of the manuscript was not the most right one. Accordingly, Section 5 (affecting these paragraphs referred to by the reviewer) has been completely rewritten in the reviewed version of the manuscript to address this and some other comments by both this reviewer and reviewer 3 (see new section 5, lines 791-866).

L745-751 are great, they report the actual data and highlighting the importance of reporting bloom conditions.

Section 6 has been fully re-written following the spirit of this paragraph mentioned by the reviewer.

L755: Correct "occurs"

Corrected.

L758: Delete "sampled" since it has been stated several times that all the oceans were sampled.

Deleted.

L760-766: Again, this is not a discussion about a gap in the dataset, it is explaining again what data is reported, how important it is to combine $^{234}$Th with other tracers or devices to quantify export, etc. This can be deleted since it has already been presented and discussed along the manuscript.

This paragraph has been deleted in the new version of the manuscript.

L761: Correct "provides", since it is the "combined interpretation" that provides us with the tools.

Corrected.

L763-764: The "Ceballos-Romero et al. 2016" citation is not needed or add other studies, this is not the only one that has combined Th-234 with other tools.

Citation removed.

L764-765: "For that reason……. might be missing." These two sentences are important, but they should be moved to the section describing the dataset, section 2. Data.

The text has been moved to section 2 (see lines 146-151).

L777: Change "us" by "as"

Changed.

L778: Please rephrase "Together with particle sinking velocities disequilibrium derived", I do not follow it.

The part of the text affecting this sentence has been entire rewritten (see section 6 of the reviewed version of the manicurist, lines 867-915).

L781: It should be BCP

Corrected.

L783: all the ways to re-evaluate the $^{234}$Th-derived POC fluxes can be done using the dataset of this compilation, but it does not allow to evaluate those fluxes in relation to the particle sinking velocity or the synthesizing modelled particle remineralization, so please rephrase to clarify what can be done with it alone and what could be done if combined with other datasets.

This sentence has been clarified as requested by the reviewer. In fact, the entire section 6 has been rewritten with a new approach that address comment about the lack of actual data in the manuscript from this reviewer.

*Table 1:*

Please check sampling periods. In several instances the dates are placed reversed i.e., first the end of the sampling followed by the starting date. See: Matsumoto, 1975; Minagawa and Tsunogai 1980; Wei and Murray, 1991; Porcelli et al. 2001 (different format used: yyyy/mm/dd); Lepore et al. 2009

All the sampling periods have been checked and corrected when found an issue (done for a total of 5 studies). (see table 2).

This problem is also found in the dataset available in PANGAEA. Please correct it there too. I noticed these ones but please review them all again.

We thank the reviewer for checking this issue in PANGAEA. All the spreadsheets underwent several rounds of quality control, both internal and external, but some formatting issues like the one pointed out by the reviewer were undetected. All the spreadsheets in PANGAEA have been reviewed and corrected. This mistake was detected in a total of 5 studies of the 223 compiled. These studies are the ones indicated by the reviewer plus Speicher et al., 2009 and Lepore et al., 2009.

Also, some of the studies have only one sampling date, therefore, that term could be simplified by reporting just that date instead of reporting it twice. See: Moran and Buesseler, 1992; Buesseler et al. 2000; Cai et al 2002; Coppola et al 2002; Guo et al 2002; Hung and Gong 2007; Buesseler et al 2010 and Pabortsava, 2014

Dates for the studies indicated by the reviewer (and some others with similar start and end dates of sampling) have been simplified as suggested (see table 2).

It would be good to increase the width of the "Reference" column because some of them take more than one row. No data repository is reported in that column, so the title of that column can be simplified.

We appreciate the reviewer's comment on this issue. We consider this table will beneficiate from being depicted in a page with landscape orientation. However, the versions of the manuscript submitted for reviewer depicted the tables in the best way to be read following the format request by the journal (i.e., portrait orientation). We will take into consideration the reviewer's suggestion regarding formatting issues during the proof-reading phase with the journal.

Nonetheless, some changes have been made in the Tale in the reviewed version of the manuscript to ease its reading.

*Table 2:*

As mentioned in my detailed comments, I would like to understand the criteria used to include programs or in this table or not, since some mentioned in the manuscript are not included. On the other hand, OSP- Station PAPA Ocean Weather Station "P" is not mentioned in the manuscript but is included in the table.

The reviewer is right, we missed reporting OSP- Station PAPA Ocean Weather Station "P" in the manuscript. It has been included in the reviewed version as a milestone of era 4 (see lines 420-423).

Please format the table so that the "Acronym" column as well as the "Activity" column are wide enough to have the entire name in one row. Right now, it is too big and it makes it hard to follow.

Same response as for the comment on the formatting of table 1. We consider this table will beneficiate from being depicted in a page with landscape orientation. We will take into consideration the reviewer's suggestion regarding formatting issues during the proof-reading phase with the journal.

Nonetheless, some changes have been made in the Tale in the reviewed version of the manuscript to ease its reading.

*Figure 1:*

Panel c) LTTS has not been used anywhere in the manuscript, the acronym used is TSS. If for some reason you prefer to use long-term, high-frequency time-series stations, do not use the acronym.

Corrected (see Figure 1).

Same for L831, TTS station is the wrong acronym.

Corrected (see Figure 1).

Panels a) and c) have already a lot of symbols and colours reporting data. I don't see the necessity to provide the topography data. A scale of greys for the background might be less shocking and allow to better visualize the data reported in these panels.

We agree with the reviewer that maps contained too much information in the previous version of the manuscript. Accordingly, all maps have been redone in the reviewed version of the manuscript to ease their interpretation.

*Figure 2:*

L842: vi) remove the """ from the category Others in the caption.

Corrected (see Figure 2).

For panel b) the lines for 2001 and 2010 are very out of place. Please add the legend in this graph again if this panel is place right below panel a) for publication.

This issue with the doted lines has been corrected in all the plots. Legend has been added for panel b).

For panels a) and c) the doted lines indicating the tipping points are also not centred in relation to the specific year.

This issue with the doted lines has been corrected in all the plots.

*Figure 3:*

L873: Delete one of the "by the"

Corrected (see Figure 3).

Same as per Figure 1, I think a grey background might allow to better visualize the datapoint reported in the different panels.

All maps have been redone in the reviewed version of the manuscript to ease their interpretation.

 The caption does not match the symbols/colours of the map:

Panel a) ii) $^{238}$U + total $^{234}$Th the caption should be dark blue squares; iii) no $^{238}$U the caption should be magenta dots

Corrected (see Figure 3).

Panels b, c and d) iii) the caption should be no total $^{234}$Th sampled (magenta dots); iv) the caption should be no POC: $^{234}$Th ratios (light blue triangles)

Corrected (see Figure 3).

Notice that the legend for panels b), c) and d) the Th in the light blue triangles shouldn't be a superscript.

Corrected (see Figure 2).

**Reviewer #3**

This study presents a compilation of oceanic $^{234}$Th measurements made at global scale over the past 50 years (1967-2018). The dataset is composed of several parameters including total, dissolved, and particulate $^{234}$Th activity, POC:$^{234}$Th and PON:$^{234}$Th ratios, along with $^{238}$U activity, POC and PON concentrations when available. This set of parameters constitutes the basis for the use of $^{234}$Th as a proxy of carbon and nitrogen export fluxes from the oceanic upper water column. The data were obtained from several sources, the vast majority from peer-reviewed articles (214) and to a lesser extent from PhD manuscripts (4), public data repositories (8), and unpublished datasets (9). The database is composed of 223 excel spreadsheets along with a compilation file hosted in PANGAEA repository. For each data set, relevant metadata (geographic location, sampling date, project, sampling and processing methodology, bloom stage, etc.) have been systematically included. The associated paper introduces the dataset, presents a global overview, and then discusses the timeline of $^{234}$Th measurements according to four periods covering the last 50 years. Finally, the authors discuss gaps in the dataset and present some perspectives on its future uses.

General comments

First, I would like to acknowledge the extensive work that has gone into this very comprehensive and clear compilation of over 50 years of research on the short-lived radionuclide $^{234}$Th. Such a compilation was lacking until now and represents a new important step in the use of $^{234}$Th as a proxy for the export of C and other elements (N, BSi, CaCO$_3$, and trace elements) from the upper water column. The database is well structured and clearly described with detailed metadata of significant importance. It also appears very exhaustive, and I could only identify a few minor omissions or errors (see detailed comments below).

We appreciate the comments of the reviewer. We specially appreciate the very thorough and elaborated revision. We strongly believe its meticulous remarks have greatly improved the text and the whole compilation

My first general concern is with the form, as the manuscript contains a significant number of typographical and editorial errors that would have benefited from careful review before its submission. This concerns in particular the list of bibliographic references which contains a significant number of errors (authors list, authors name, type of reference such PhD manuscript, book chapter, research article). Still on the form, there is a surprising confusion between concentration and activity made throughout the manuscript. The $^{234}$Th and $^{238}$U data you report are activities not concentrations.

We are very grateful to the reviewer for its thorough review, attending both formatting issues and the manuscript content itself. We apologize for all the formatting issues and typos that were overlooked in our previous version. We understand that such compilation should be perfectly edited. We employed time and effort to do that, including more than four people in the task. However, we see now, that our revision was far from perfect. That is why we specially appreciate the effort made by the reviewers to polish text, Figures and compilation itself. We had made an additional effort this time to improve the manuscript and the content. We strongly believe the manuscript is greatly improved after implementing all the reviewer's suggestions.

As for the activities versus concentrations issue, we agree with the reviewer that these terms were used indistinctly in the previous version of the manuscript which generated confusion.

What we report in the compilation for total, dissolved and particulate $^{234}$Th and for $^{238}$U are "activity concentrations" as corresponds to activities per unit volume (measured in units of dpm L$^{-1}$), as defined by ISO 921 (see comment of this on page 4 of the IAEA glossary (chrome-extension://efaidnbmnnnibpcajpcglclefindmkaj/viewer.html?pdfurl=https%3A%2F%2Fwww-ns.iaea.org%2Fdownloads%2Fstandards%2Fglossary%2Fiaea-safety-glossary-draft-2016.pdf&clen=1708013&chunk=true). For the sake of simplicity, we have referred to these data as simply "concentration". as acknowledged in the reviewed version of the manuscript (see lines 85-86).

On the content, I was pleased to read the timeline of the Th studies which summarizes the major stages that contributed to the development of the method. For clarity to the reader, I would recommend to indicate for each period the corresponding years both in the manuscript in the subsection headings and in the figures.

We appreciate the reviewers' comment on this regard. Changes have been implemented, including the duration of each specific era and a description for the main theme that characterized it. In the reviewed version of the manuscript, we have kept name of the eras after titles from seminal ocean-related books but included a quote from each one of them to evince each one of the choices made for the era, which follow an intention are not random.

This information has been added to the text, tables, and figures.

About the different eras, I might have subdivided era 1 and 2 a little differently by considering the JGOFS program as the beginning of era 2. In fact, it seems more logical to me to take into account the sampling periods corresponding to different programs rather than the date of publication of the resulting studies. This is illustrated further in Table 2, where most of the studies belonging to era 2 were performed in the framework of JGOFS.

We agree with the referee that the arrangement of the eras could be logical considering the sampling, not the publications. However, to do a correct and consistent analysis of the data we need to be extremely coherent with the dates associated to the datasets, so in that case, we should use that classification for the whole dataset. And the problem of a classification by sampling is that it cannot be applied to many of the publications analyzed, as they include several samplings, and it is difficult to designate a single sampling date. Furthermore, other publications present annual results of a single station, thus including many years. In those cases, we can only use the date of publication. And for consistency, publication date is applied to all the analysis of the eras, otherwise, consistent statistics could not be obtained.

On the other hand, the eras are chosen by a differential univocal event that can be considered the kick-off of the era. The reviewers were right that the criteria to choose this univocal event was not clear in the previous manuscript. Therefore, we have decided to define four relevant papers for each era as kick-off event. Those papers are the following: First published paper detection $^{234}$Th disequilibrium in the ocean (Bhat et al, 1969), first paper where POC export is quantified from 234Th disequilibrium (Buesseler et al., 1992), publication of the first small volume technique for $^{234}$Th measurement (Buesseler et al., (2001) and Benitez-Nelson et al., 2001) and finally first $^{234}$Th paper published within the GEOTRACES program (Cai et al., 2010).

This way, the eras, and the analysis of their data, are finally distributed as follows:
- 1969-1991: The beginning of $^{234}$Th as tracer of POC export.
- 1992-2000: Introduction of the empirical determination of POC export from $^{234}$Th profiles.
- 2001-2006: Consolidation of the $^{234}$Th technique.
- 2010-present: GEOTRACES program and a new way to study the ocean.

Therefore, the eras are arranged as in the previous version of the manuscript, but we believe are reason for the criteria chosen are well justified in the reviewed version of the manuscript.

References:

Bhat, S. G. G., Krishnaswamy, S., Lal, D., Rama, & Moore, W. S. S. (1969). 234Th/238U ratios in the ocean. Earth and Planetary Science Letters, 5, 483–491. https://doi.org/10.1016/S0012-821X(68)80083-4.

Buesseler, K. O., Bacon, M. P., Kirk Cochran, J., & Livingston, H. D. (1992). Carbon and nitrogen export during the JGOFS North Atlantic Bloom experiment estimated from 234Th: 238U disequilibria. Deep Sea Research Part A. Oceanographic Research Papers, 39(7–8), 1115–1137. https://doi.org/10.1016/0198-0149(92)90060-7.

Buesseler, K. O., Benitez-Nelson, C., Rutgers van der Loeff, M., Andrews, J., Ball, L., Crossin, G., & Charette, M. A. (2001). An intercomparison of small- and large-volume techniques for thorium-234 in seawater. Marine Chemistry, 74(1), 15–28. https://doi.org/10.1016/S0304-4203(00)00092-X.

Benitez-Nelson, C., Buesseler, K. O., van der Loeff, M. R., J. Andrews, L. B., Crossin, G., Charette, M. A., Loeff, M. R. van der, J. Andrews, L. B., Crossin, G., & Charette, M. A. (2001). Testing a new small-volume technique for determining 234Th in seawater. - Journal of Radioanalytical and Nuclear Chemistry, 248(3), 795-799. https://doi.org/- 10.1023/a:1010621618652.

Buesseler, K. O., Bacon, M. P., Kirk Cochran, J., & Livingston, H. D. (1992). Carbon and nitrogen export during the JGOFS North Atlantic Bloom experiment estimated from 234Th: 238U disequilibria. Deep Sea Research Part A. Oceanographic Research Papers, 39(7–8), 1115–1137. https://doi.org/10.1016/0198-0149(92)90060-7.

Regarding the first era, you may think to introduce the GEOSECS program earlier than L365 after the description of the Coale and Bruland papers. I think it could be relevant to mention that it was in the framework of GEOSECS that the concept of scavenging by particles has really emerged, especially for the open ocean. You may cite the seminal work of Turekian (1977) who introduced the concept of the "great particle conspiracy". Furthermore, it might be interesting to mention that during GEOSECS little attention was given to $^{234}$Th in comparison to $^{228}$Th and $^{230}$Th for studying scavenging processes in the open ocean (Broecker and Peng, 1982). It was only later, with the papers of Coale and Bruland, that the role of $^{234}$Th as a tracer of short-term particle dynamics was really highlighted.

We appreciate the review's comment regarding GEOSECS, but we consider there is no need to mention it earlier than done, when starting to discuss $^{234}$Th timeline in era 1 (i.e., section 4.1).

The discussion suggested by the reviewer regarding particle scavenging and the great particles conspiracy has been included in the text, as suggested by the reviewer (see lines 364-369).

Regarding era 4 related to the GEOTRACES program, I would set the starting year to 2008 or even 2007, which corresponds to the International Polar Year (2008) and the start of the GEOTRACES sampling program.

Wee agree with the reviewer here regarding the starting date that should be associated to the GEATRACES program, corresponding to the IPY. However, as explained above, this is not relevant anymore since, in order to follow a strict, straightforward criterion to designate the eras, we have chosen influential $^{234}$Th-related papers as starting point to frame the eras. In this case we have chosen Cai et al. (2010) (https://doi.org/10.1029/2009JC005595) as the first Th-234 results published in the framework of the GEOTRACES program.

Regarding this program, it would be relevant to include in table 2 all cruises that have been sampled for $^{234}$Th so far.

Table 2 include the description of the programs, but the cruises are not included in any of the different programs. Therefore, we find that it will not be consistent to include GEOTRACES cruises and not for the others. But in those cases, if all the cruises were included for all the programs, Table 2 would be immensurable. We hope the reviewer understand our concern about including the cruises here.
All programs under which sampling have taken placed are indicated in the summary table available in PANGAEA (see https://doi.pangaea.de/10.1594/PANGAEA.918125?format=html#download).

For consistency, I would recommend using the name of the section or process study considered as defined by GEOTRACES, you can then indicate which country has been involved in the sampling (US/UK/Netherland/Germany/India/France). This remark is also valid for the Th_database file, the projects performed within the framework of GEOTRACES should be named in a more consistent way such as for instance GEOTRACES (section or process study number, country, and eventually project acronym).

This information is included in each of the individual data files and in the text (see for example lines 290-296).

Still on the GEOTRACES program, I think it is important to mention that not all sections analyzed for $^{234}$Th are available in the 2014 and 2017 Intermediate Data Products (L367). Even if data obtained as part of GEOTRACES are published in peer-reviewed journals, their inclusion in the IDP requires some additional steps (submission and acceptance by the GDAC). Also, you may indicate that the last IDP (2021) was released very recently.

This remark is now included in the text. IDP2021 has been added to the manuscript (see line 563).

I have also some concern regarding the section 5. Significant gaps in the global dataset. It is not clear to me what the gaps you want to discuss are. Reading the first paragraph (L696-701), it seems the gaps you want to consider are related to the current understanding of the Biological Carbon Pump. On this topic, I would recommend to include some more recent reviews (Henson et al., 2019; Boyd et al., 2019; Siegel et al., 2016), which detail some of the processes that require further consideration. Reading the following, I notice you discuss two main points, the first one is related to the spatio-temporal distribution of 234Th data and the second one, too long considered from L711 until the end of the section L768, is related to the modeling approach (SS vs NSS) used for estimating the export fluxes of $^{234}$Th. It is surprising to note that this entire section is mostly discussed using only two references and written by the first author of this review. In my view, this section needs to be reconsidered, first by giving more attention to the existing literature on the SS/NSS approach and the different oceanic contexts to which it has been applied (not only the North Atlantic), and second to the other numerous issues related to the $^{234}$Th method. Among these, it is important to point out the role of physics (lateral and vertical advection and diffusion) (Buesseler et al., 1995; Savoye et al., 2006; Resplandy, 2012; Le Gland et al., 2019; Roca-Marti et al., 2017), the importance of the depth of integration (Buesseler et al, 2020), and finally all the other issues related to the conversion to carbon fluxes using the POC to Th ratio of sinking particles (the choice of the relevant size fraction, the interpolation methods, etc.). By following these guidelines and considering that export fluxes have not been calculated or compiled in this review, you may be able to give recommendations to future users of this database.

We agree with the review that the approach given to this entire section in the previous version of the manuscript was not the most right one. Accordingly, Section 5 has been completely rewritten in the reviewed version of the manuscript to address this and some other comments by both this reviewer and reviewer 3 (see new section 5, lines 791-866).

Finally, and still about the gaps in this dataset, there is one point that could be considered that concerns the quality of available data. I understand that it is difficult to answer this point but if we take into account the successive evolutions of the methods used for the determination of $^{234}$Th, all the data are probably not of the same quality. I think this should be at least mentioned or even taken into account in the form of a quality flag assigned to each dataset.

See specific comment at the beginning of the response to reviewers. To take into account in the discussion of the dataset that there are certain parameters that are missing in the compilation we have modified Section 5. As discussed above, we agree with the reviewer that this section was not well focused, and we have rewritten the whole section accordingly. Section 5 now includes a subsection related to the parameters that are missed in the compilation, including the necessity of a flag to estimate the quality of the data.

For the particular case of the reviewer's comment, we believe that addressing the quality of the data compiled is out of the scope of the compilation, at this stage. It would imply ruling out quality of the data provided by the researchers without a reaching a previous consensus with the authors.
We acknowledge however, that the point made by the reviewer is an important one to be considered, and it would improve the quality of the compilation. For that reason, for next editions of the compilation we will consider including a Flag to the new data included. That flag will be added after agreement with the authors whose data are included the data set.

Detailed comments

L15: the 234Th-238U pair is primarily used for assessing export fluxes, to look at export efficiency you need to compare with the net primary production, which is actually not included in the dataset. Please clarify

The reviewer is right, we do not provide the primary production, parameter required to evaluate the export efficiency and therefore, the sentence pointed out by the reviewer is confusing. In the reviewer version of the manuscript, we have rephrased this part of the text (see abstract).

L20: replace concentrations by activities and at all other occurrences in the manuscript for both 234Th and 238U.

Done throughout the entire manuscript.

L29: the term uptake is a bit ambiguous and not directly related to the 234Th method and to export fluxes. Uptake can be used either for air-sea exchanges or biological assimilation, please clarify.

L35: correct the reference (Cochran and Masqué, 2003)

Corrected (see line 32).

L45: remove "a" in front of 234$^{Th}$

Corrected.

L50: mean life time

Corrected (see line 47).

L52: check the end of sentence "high scavenging"

Checked and corrected.

L55: correct reference for Cochran, 1993 (Kirch is the first name)

Corrected (although note this reference has been erased from this part of the manuscript in the reviewed version).

L57-60: you may cite also the works of Lemaitre et al. (2016) for N, BSi, Fe export in the Kerguelen area (KEOPS2 project) and Lemaitre et al. (2020) for BSi, CaCO3, lithogenic material and trace elements along the GA01 section (GEOVIDE) in the North Atlantic.

This reference has been cited in the text (see line 54), and also included in the Excel files. We thank the reviewer for this reference since we were not aware of this data set.

L63-64: remove the editors list of the special issue

Removed.

L88: add Th after 234

234 removed, that was the mistake.

L104: 238U activity

Corrected.

L105: you mean compiled and not "complied"

Corrected.

L114-115: as mentioned earlier, data obtained in the frame of GEOTRACES are not necessarily in the IDP

This sentence has been modified to take this into account (see lines 103-105).

L124: not necessary to define supplementary material with an acronym (figure or table number is sufficient).

Acronym removed.

L129: could you provide more details on how the bloom stages were distinguished?

Done (see lines 119-129).

L145: please clarify what you mean with CHN data and how this differ from POC and PON concentrations reported in the dataset (as mentioned L104). Furthermore, CHN is not the only analytical method that can be used to determine POC and PON, EA-IRMS is also widely used for this purpose.

We have used the CHN label to indicate studies that provide POC and/or PON concentrations (in $\mu$mol L$^{-1}$). This has been clarified din the reviewed version of the manuscript. Other analytical methods that exit to measure CHN data in addition to an elemental analyzer have also been mentioned, as suggested by the reviewer (see lines 89-92).

L151: Check sentence "Where data…"

Checked and corrected.

L168: remove concentration for 234$^{Th}$

Removed.

L169: correct units $\mu$mol dpm$^{-1}$ for POC and PON to 234Th ratios

Corrected.

L172: correct >53 $\mu$m and not <53$\mu$m

Corrected.

L173: was responsible instead of is

Corrected.

L178: check units $\mu$mol L$^{-1}$

Checked and corrected.

L181: correct one sigma

Corrected (see line 190).

L190: remove the _ after authors

Removed.

L193: check sentence and correct "used for"?

Removed.

L198: the previous compilation by Le Moigne et al. (2013) include 234Th fluxes, POC:Th ratios, POC fluxes and NPP estimates.

We have included this information in the reviewed version of the manuscript (see lines 205-208).

L200: correct aspects

Removed.

L201-202: check the end of the sentence

This text has been rephrased.

L205: remove SM

Removed.

L208: correct larger

Corrected.

L214: put the acronym in brackets

Done.

L222: correct "on at least.."

Corrected.

L235: repeated occupation?

Corrected.

L242: Correct "the way ocean is currently…"

Corrected.

L275: you may include the KEOPS project (sampling in 2005) with the study of Savoye et al. (2008) and also the great Calcite Belt expedition with the study of Rosengard et al. (2015) carried out both in the Atlantic and Indian sectors of the SO.

We thank the reviewer for the suggestion; however, we have only identified a total of 210 and 399 [234]Th data points for the studies of Savoye et al., (2008) and Rosengard et al., (2015) in comparison to the e.g., 1610 and 1040 data points reported by Owens et al., (2015) and Buesseler et al., (2020) sampled in 2010 and 2018 respectively and referred to in this part of the text. Is for this reason that we do not consider these 2 studies suggested by the reviewer make a significant peak in the annual distribution of [234]Th data points relative to the years selected, this is i) 1992 ii) 1997 iii) 2004, iv) 2010 and v) 2018), and we have decided not to include these 2 studies in the reviewed version of the manuscript.

References:

Owens, S. A., Pike, S., & Buesseler, K. O. (2015). Thorium-234 as a tracer of particle dynamics and upper ocean export in the Atlantic Ocean. *Deep Sea Research Part II: Topical Studies in Oceanography*, *116*(0), 42–59. https://doi.org/http://dx.doi.org/10.1016/j.dsr2.2014.11.010

Buesseler, K. O., Benitez-Nelson, C. R., Roca-Martí, M., Wyatt, A. M., Resplandy, L., Clevenger, S. J., Drysdale, J. A., Estapa, M. L., Pike, S., & Umhau, B. P. (2020). High-resolution spatial and temporal measurements of particulate organic carbon flux using thorium-234 in the northeast Pacific Ocean during the EXport Processes in the Ocean from RemoTe Sensing field campaign. *Elementa: Science of the Anthropocene*, *8*(1). https://doi.org/10.1525/elementa.030

L276: remove the # and correct "in terms"

Removed and corrected.

L283: correct "field"

Corrected.

L284-285: what do you mean with "chemical scavenging"

Corrected to simply "scavenging".

L290: correct "through years"

Corrected.

L292: check the beginning of the sentence

Checked and corrected.

L305: correct "were"

Corrected.

L329-333: the end of the paragraph is not related to 238U measurements but to additional metadata information (model assumption and bloom stages)

The end of this paragraph has been removed as included repeated information.

L357-358: check sentence

Checked and corrected.

L364-366: I would consider the GEOSECS program earlier (see my general comment)

We appreciate the reviewer's comment on this regard, but as previously mentioned, we consider there is no need to mention the GEOSECS program (or any other program) earlier than in Section 4, where we discuss the timeline of the $^{234}$Th technique, to which these and other programs are relevant.

L387: Clarify the beginning of the sentence

Clarified.

L397: correct "do not"

Corrected.

L404: remove concentration with activity

Not removed, see reply to general comment regarding the use of activity versus concentration in the manuscript.

L409: Nucleopore

This sentence has been rephrased and "Nucleopore" does not appear anymore.

L420-423: I would also mention that physical terms were not considered for solving the 234Th activity balance.

The mention to the physical term has been included (see lines 500-503).

L423: remove of or rephrase

The entire paragraph has been rephrased (see line 501-505).

L433: Clarify what kind of models were used (one or two boxes) and the corresponding fluxes (scavenging fluxes J and export fluxes P). Also check the end of the sentence.

The kind of models used has been specified in the reviewed version of the manuscript (lines 497-509).

L437: JGOFS started in 1987 according to Table 2

After consulting more sources, we have marked the beginning of JGOFS program in 1984 and corrected this date throughout the entire manuscript (see table 2).

L441-442: correct relative to its parent nuclide or 238U

Corrected.

L443: clarify what you mean with "response time" and also to what corresponds the second equation

It was an error, it should say mean residence time, the second equation has been deleted from the text, as it was a very specific one, and it did not fit in in the text (line 485).

L455: correct "with time"

Corrected.

L456: here you keep focusing on the SS vs NSS approach. You need to better account for the physical terms (according to Buesseler et al., 1995 in specific ocean settings such as upwelling regions). Furthermore, vertical diffusion can be quantified from a single 234Th profile if the diffusivity coefficient is known or assumed.

Section 5 (affecting these paragraphs referred to by the reviewer) has been completely rewritten in the reviewed version of the manuscript to address this and some other comments by both this reviewer and reviewer 3.

L458: check sentence, you may change to "when temporal fluctuations in 234Th activity can be assessed". In addition, and as mentioned by Savoye et al. (2006), NSS approach requires the same water mass to be sampled. This is another difficulty of the NSS approach that can be difficult to meet in dynamical settings.

The whole paragraph has been rewritten including all the suggestions of the reviewer and the discussion on the physical terms (see lines 501-507)

L466: correct "0.001<colloids…"

Corrected (see line 512).

L471: There is a confusion, the 3-D model was not built to estimate the gradients of 234Th activities but to estimate the physical components to the flux (V terms). Without these components, the sinking flux would have been largely underestimated in the equatorial upwelling region.

This is corrected in the text (see lines 517-519).

L482: I don't think the SEATS time-series was operated in the frame of the French JGOFS program, most probably by Taïwan.

The reviewer is right, SEATS time-series station is operated by Taiwan, established din the framework of JGOFS. This fact has been corrected in the reviewed version of the manuscript (see lines 522-523).

L489: Clarify the end of the sentence "filtered"

Clarified.

L494: sediment traps collect sinking particles not bulk.

Corrected.

L505: correct "at the BATS…"

Corrected.

L506: replace "boxes" with areas

Corrected.

L508: check coordinates for the NABE experiment

We've identified no mistake with these coordinates (see line 557).

L518: correct "field work"

Corrected.

L520: Australian

Corrected.

L523: correct "other major initiatives…"

Corrected.

L534: remove "during sampling"

Corrected.

L537: clarify what are the CHN data

Clarified.

L569: the reference Clevenger et al. (2021) is apparently missing in the reference list.

Corrected.

L573: clarify "234Th contraptions"

Corrected.

L626: Correct "Portland (USA, Oregon)" and "Numerous cruises"

Corrected.

L638: correct "are known…"

Corrected.

L664: remove "of the studies"

Corrected.

L666-667: check the sentence

Checked and corrected.

L680: correct "emphasizes"

Corrected (see line 771).

In the section, you may include the MOBYDICK project (Marine Ecosystem Biodiversity and Dynamics of Carbon around Kerguelen: an integrated view, 2018-2022) during which the 234Th method has been implemented.

We thank the reviewer for this useful information which has included in the reviewed version of the manuscript (see lines 776-777).

For the APERO project, it will start in 2022 to 2026 and the cruise is planned for 2023 in the western North Atlantic.

We thank the reviewer for this useful information which has included in the reviewed version of the manuscript (see line 777-780).

L706: correct "gaps"

Corrected.

L722: correct "with time"

Corrected.

L777-778: correct "such as" and clarify the whole sentence

Corrected and sentence clarified.

Table 1. Check start and end date (sometimes inverted). The design needs to be improved as it is very difficult to identify which parameters has been measured for each studies.

Checked and corrected (see table 2).

We appreciate the reviewer's comment on the design of the table issue, and we agree with it. We consider this table will beneficiate from being depicted in a page with landscape orientation. However, the versions of the manuscript submitted for reviewer depicted the tables in the best way to be read following the format request by the journal (i.e., portrait orientation). We will take into consideration the reviewer's suggestion regarding formatting issues during the proof-reading phase with the journal.

Nonetheless, some changes have been made in the Tale in the reviewed version of the manuscript to ease its reading.

I noted a few mistakes for the studies that I know (I didn't check all entries in this table)

The study of Stewart et al. (2007) reports POC:Th ratio from in-situ pumps and sediment traps but not PON:Th ratios (check the crosses)

We thank the reviewer for this comment. In the previous version of the manuscript the information detailed in the caption did not match the information as included in the table. We have checked and corrected the entire table, include the studies highlighted by the reviewer.

Thomalla studies in the Atlantic ocean was published in 2008

In addition to the study in 2008, Thomalla also published additional data in the North Atlantic in her paper of 2006 and her PhD thesis in 2007. This author kindly sent us the PhD dissertation to include data from it in the compilation.

References:

Thomalla, S., Turnewitsch, R., Lucas, M., & Poulton, A. (2006). Particulate organic carbon export from the North and South Atlantic gyres: The 234Th/238U disequilibrium approach. *Deep Sea Research Part II: Topical Studies in Oceanography*, *53*(14–16), 1629–1648. https://doi.org/10.1016/j.dsr2.2006.05.018

Sandy Thomalla. (2007). *Particulate organic carbon and mineral export from the North and South Atlantic gyres: the 234Th/238U disequilibrium approach*. University of Cape Town.

For Jacquet et al. (2011), PON:Th ratio were not provided but POC:Th ratio from sediments traps yes

Corrected and full table checked.

For the Rutgers van der Loeff et al. (2011) study, underway sampling were performed and reported

We thank the reviewer for providing this detail. This underway information has been corrected in the excel file (Reference_USE 0142) and in table 2.

Planchon et al. (2015) study does not report PN:Th ratio but C:Th ratio from sediment traps. Also dissolved Th was plotted in Figure 2. The PN:Th ratios for this cruise (KEOPS2) can be found in Lemaitre et al. (2016).

We thank the reviewer for this comment. Table 2 has been entirely reviewed and corrected when necessary for the information provided by each study.
Additionally, PN:234Th ratios from KEOPS2 reported by Lemaitre et al. (2016) have been included in the excel spreadsheet corresponding to this field expedition (i.e., Planchon et al., 2015, Reference_USE 0129).

For the Lemaitre et al. (2018) study, please correct the reference as follow:

Lemaitre, N., Planchon, F., Planquette, H., Dehairs, F., Fonseca-Batista, D., Roukaerts, A., Deman, F., Tang, Y., Mariez, C., and Sarthou, G.: High variability of particulate organic carbon export along the North Atlantic GEOTRACES section GA01 as deduced from 234Th fluxes, Biogeosciences, 15, 6417–6437, https://doi.org/10.5194/bg-15-6417-2018, 2018.

Corrected and all references checked.

The study of Maïti et al. (2016) reports total Th profiles

Corrected and full table checked.

Table 2.

As for Table 1, the design needs some improvements. Column headings needs to be clarified and further details on the other programs than JGOFS could be included. This could be especially the case for GEOTRACES as well as EXPORTS. Furthermore, I do not see the reason why a given program is considered a major program. For instance, the HiLATS program is indicated only for 2001

The compilation of program included in Table 2 represent a selection of selected programs, chosen in terms of number of cruises achieved, duration of the program, relevance of results throughout time, etc. And overall, a global representation of programs from all countries have been followed. HilATS have been deleted from the Table and from the "selected ocean program" in the reviewed version of the manuscript.

Check the start date for DYFAMED

Checked and corrected to 1987, when the first $^{234}$Th sampling was reported by Schmidt et la., (1990).

Note that the compilation published by Coppola et al., (2021) was also consulted to this end, but 2 different dates are stated there (1988 and 1991) and no conclusion could be drawn from it.

References:

Schmidt, S., Reyss, J. L., Nguyen, H. V., & Buat-Ménard, P. (1990). 234Th cycling in the upper water column of the northwestern Mediterranean Sea. *Global and Planetary Change*, *3*(1–2), 25–33. https://doi.org/10.1016/0921-8181(90)90053-F

Coppola, L., Diamond Riquier, E., & Carval, T. (2021). Dyfamed observatory data. In *SEANOE*. https://doi.org/https://doi.org/10.17882/43749

Figure 1: use the same acronym as in the text for long-term time series (TTS)

Corrected (see figure 1).

Figure 3: check the legend "by the" written twice

Corrected (see figure 3).

Regarding the dataset, it would be worth checking if the data of Stuckel et al. (2015) and Ducklow et al. (2018) have been included.

We thank the reviewer for these references provided. We confirm that they are not included in the current version of the compilation but will be included in the next version along with other two studies (see below for references) that we have identified as missing during this review process.

Ma, H., Zeng, Z., He, J., Han, Z., Lin, W., Chen, L., Cheng, J., & Zeng, S. (2014). 234Th-derived particulate organic carbon export in the Prydz Bay, Antarctica. *Journal of Radioanalytical and Nuclear Chemistry*, *299*(1), 621–630. https://doi.org/10.1007/s10967-013-2842-y

Turnewitsch, R., Reyss, J.-L., Nycander, J., Waniek, J. J., & Lampitt, R. S. (2008). Internal tides and sediment dynamics in the deep sea—Evidence from radioactive 234Th/238U disequilibria. *Deep Sea Research Part I: Oceanographic Research Papers*, *55*(12), 1727–1747. https://doi.org/10.1016/j.dsr.2008.07.008

References:

Boyd, P.W., Claustre, H., Levy, M., Siegel, D.A., Weber, T., 2019. Multi-faceted particle pumps drive carbon sequestration in the ocean. Nature 568, 327–335. https://doi.org/10.1038/s41586-019-1098-2.

Buesseler, K.O., Andrews, J.A., Hartman, M.C., Belastock, R., Chai, F., 1995. Regional estimates of the export flux of particulate organic carbon derived from thorium-234 during the JGOFS EqPac program. Deep Sea Research Part II: Topical Studies in Oceanography 42, 777–804. https://doi.org/10.1016/0967-0645(95)00043-P.

Buesseler, K.O., Boyd, P.W., Black, E.E., Siegel, D.A., 2020. Metrics that matter for assessing the ocean biological carbon pump. Proc Natl Acad Sci USA 117, 9679. https://doi.org/10.1073/pnas.1918114117.

Ducklow, H.W., Stukel, M.R., Eveleth, R., Doney, S.C., Jickells, T., Schofield, O., Baker, A.R., Brindle, J., Chance, R., Cassar, N., 2018. Spring–summer net community production, new production, particle export and related water column biogeochemical processes in the marginal sea ice zone of the Western Antarctic

Peninsula 2012–2014. Philosophical Transactions of the Royal Society A: Mathematical, Physical and Engineering Sciences 376, 20170177. https://doi.org/10.1098/rsta.2017.0177.

Henson, S., Le Moigne, F., Giering, S., 2019. Drivers of Carbon Export Efficiency in the Global Ocean. Global Biogeochemical Cycles 33, 891–903. https://doi.org/10.1029/2018gb006158.

Le Gland, G., Aumont, O., Mémery, L., 2019. An Estimate of Thorium 234 Partition Coefficients Through Global Inverse Modeling. Journal of Geophysical Research: Oceans 124, 3575–3606. https://doi.org/10.1029/2018JC014668.

Lemaitre, N., Planquette, H., Dehairs, F., Planchon, F., Sarthou, G., Gallinari, M., Roig, S., Jeandel, C., Castrillejo, M., 2020. Particulate Trace Element Export in the North Atlantic (GEOTRACES GA01 Transect, GEOVIDE Cruise). ACS Earth Space Chem. 4, 2185–2204. https://doi.org/10.1021/acsearthspacechem.0c00045.

Lemaitre, N., Planquette, H., Dehairs, F., van der Merwe, P., Bowie, A.R., Trull, T.W., Laurenceau-Cornec, E.C., Davies, D., Bollinger, C., Le Goff, M., Grossteffan, E., Planchon, F., 2016. Impact of the natural Fe-fertilization on the magnitude, stoichiometry and efficiency of particulate biogenic silica, nitrogen and iron export fluxes. Deep Sea Research Part I: Oceanographic Research Papers 117, 11–27. https://doi.org/10.1016/j.dsr.2016.09.002.

Resplandy, L., Martin, A.P., Le Moigne, F., Martin, P., Aquilina, A., Mémery, L., Lévy, M., Sanders, R., 2012. How does dynamical spatial variability impact 234Th-derived estimates of organic export? Deep Sea Research Part I: Oceanographic Research Papers 68, 24–45. https://doi.org/10.1016/j.dsr.2012.05.015.

Roca-Martí, M., Puigcorbé, V., Iversen, M.H., van der Loeff, M.R., Klaas, C., Cheah, W., Bracher, A., Masqué, P., 2017. High particulate organic carbon export during the decline of a vast diatom bloom in the Atlantic sector of the Southern Ocean. Deep Sea Research Part II: Topical Studies in Oceanography 138, 102–115. https://doi.org/10.1016/j.dsr2.2015.12.007.

Savoye, N., Benitez-Nelson, C., Burd, A.B., Cochran, J.K., Charette, M., Buesseler, K.O., Jackson, G.A., Roy-Barman, M., Schmidt, S., Elskens, M., 2006. 234Th sorption and export models in the water column: A review. Marine Chemistry 100, 234–249.

Siegel, D.A., Buesseler, K.O., Behrenfeld, M.J., Benitez-Nelson, C.R., Boss, E., Brzezinski, M.A., Burd, A., Carlson, C.A., D'Asaro, E.A., Doney, S.C., 2016. Prediction of the export and fate of global ocean net primary production: the EXPORTS science plan. Frontiers in Marine Science 3, 22.

Stukel, M.R., Asher, E., Couto, N., Schofield, O., Strebel, S., Tortell, P., Ducklow, H.W., 2015. The imbalance of new and export production in the western Antarctic Peninsula, a potentially "leaky" ecosystem. Global Biogeochemical Cycles 29, 1400–1420. https://doi.org/10.1002/2015GB005211.

---

## Author Comment (AC7)

**Authors' Response to Review Comments**

**(Manuscript Number: 'essd-2021-259')**

We are very thankful to all the reviewers for their thorough and constructive reviews. We have addressed all the issues.

**Reviewer #3**

Here are very minor editorial remarks that would be good to correct

L254: you may use the term magnitude in addition to efficiency

Added.

L261: remove the dot before and

Removed.

L268: To the Pacific Ocean

Corrected.

L295: have

Corrected.

L368: not sure, but ranked instead of rank?

Checked and corrected.

L439: remove "the" before studies

Removed.

L647: remove one dot

Removed.

L651: choose between sampled and collected

Sampled chosen.

L657: remove while

Removed.

Table 2. Correct SOIREE experiment

Corrected.

Figure 4. Check the era number in Fig4 c and d

Corrected.

**Reviewer #2**

I am very pleased to see the changes done to this manuscript. I believe it has improved significantly. I particularly enjoyed the new Sections 5 and 6, which are very clear, to the point, and provide very interesting suggestions on how to improve future data sets and how to use this great compilation. Congratulations to the authors for the titanic effort in putting this data set together and for the beautiful description of the history of 234Th.

I still have some minor comments, which I hope the authors will consider addressing. Please find below a detailed review of the manuscript.

I will start with the two specific issues that I pointed out in my previous review:

1) The use, in some instances, of total particulate carbon instead of POC. The authors replied: "Only data reported as POC in the sources consulted have been reported as POC data. And equivalent for PIC data." I cannot find a column in the data set that refers to PIC, only to CHN analyses and then all the different possibilities of POC/Th ratios, so I am not sure what the authors mean by "and equivalent for PIC". Please see, for example, Owens et al. (2015), where total carbon is provided instead of POC.

The reviewer is right, not all studies provided POC but rather total particulate carbon, such as the example of Owens et al., (2015). For those cases, we have indicated it in the "comments" section of the "METHODS" in the metadata spreadsheet. This has been noted in the reviewed version of the manuscript (see lines 87-89).

We have reviewed all the datasets compiled and we have only found this to be the case for the study mentioned by the reviewer, Owens et al., 2915, and for an additional one: Puigcorbé, V. *et al.* (2015) 'Small phytoplankton drive high summertime carbon and nutrient export in the Gulf of California and Eastern Tropical North Pacific', *Global Biogeochemical Cycles*, 29(8), pp. 1309–1332. doi: 10.1002/2015GB005134.

Regarding PIC, the reviewer is right, no information of this parameter is included in the compilation. In the original version of the manuscript, this parameter was included in the "CHN" category, but after the first review, this criterion changed based on several comments by the reviewers and "CHN" was chosen to only indicate POC and PON data. The existence of PIC:234Th is indicated as extra information the "comments" section of the "METHODS" in the metadata spreadsheet when reported by the data authors, but it does not have a specific column for it. It was a mistake replying to the reviewer "...And equivalent for PIC data". We apologize for the confusion.

2) Regarding the addition of PAR or fluorescence data, I'm very happy to see that the authors have dedicated several lines to explain the importance of these parameters and to highlight the fact that this will be included in future versions of the data set.

We are very glad the changes made in review 1 please the reviewer. We agree with the reviewer that those changes improved the manuscript.

L15: Looking at the PANGAEA list of references, I found the 9 data sets but seems a repetition having those data sets together with the papers that actually published the data from those data sets. For example:

We appreciate the reviewer's thorough work checking the list of references in PANGAEA and the datasets published. A great effort has been devoted to 1) identifying datasets not published in papers and only available in repositories, and 2) acknowledging papers that published the same cruises but different stations and/or data points. Nonetheless, we acknowledge this effort does not prevent repetition in datasets happening by mistake. See detailed replies for each of the examples.

- California Current Ecosystem LTER; Stukel, Michael R (2019): This contains data from 2006 to 2017 that has been published in papers authored by Stukel.

Please not the repository has been cited as indicated by the author (i.e., *Total Thorium-234 (Th-234) taken from discrete water column samples collected during CCE Process Cruises (2006 - 2017)*), but this data set corresponds to a cruise in 2017 (P1706 on board R/V Roger Revelle), as indicated in the summary table provided in PANGAEA, that we have not found published in any article. If the reviewer is aware of an article containing this dataset, we appreciate communicating it to us so we can incorporate it to the compilation.

- Roca-Martí et al (2015) POLARSTERN cruise ANT-XXVIII/3 data set available in PANGAEA, the data has been published in papers authored by Roca-Martí et al (2015) and Puigcorbé et al. (2017).

The reviewer is right, cruise ANT-XXVIII/3 has been reported by in Puigcorbe et al., 2017, Deep Sea Research Part II (10.1016/J.DSR2.2016.05.016) & Roca-Marti et al., 2017 (10.1016/j.dsr2.2015.12.007). However, each author reported different stations between them, and also different to those reported in the dataset referred to by the reviewer. This is acknowledged in a comment in the 3 data sets that contains data from cruise ANT-XXVIII/3 (i.e., 0133, 0136, and 0193).

- Rutgers van der Loeff data sets from Antarctic cruises available in PANGAEA have also been published in papers authored by himself or others.

Same case as before, the data reported from repositories were not found in publications. In the case of Rutgers van der Loeff, the 3 datasets reported as found in repositories correspond to cruises PS58 in 2000, PS69 between 2005 and 2006, with this one split in 2 datasets, each one corresponding to each year. Data that can be found in a repository but also published is linked to the publication and not catalogued as "data from repositories", such is the case for the dataset 0142, which can be found in the repository https://www.bodc.ac.uk/data/documents/nodb/275768/ but also published in doi:10.1016/j.dsr2.2011.02.004.

Similar case for some PhD theses. I noticed that in Table 1 different sampling periods are chosen for those that seem to be "repeating" (i.e., Owens, 2013 vs Owens et al. (2015)). Does that mean that the repositories or PhD theses contain additional data that has not, indeed, been published?

Yes, that is what it means. Data reported from repositories has not been found in any publication, as clearly stated in the manuscript.

I didn't go in detail to check that in the actual data set in PANGAEA, but I would just like a confirmation, by the authors, that duplication has been considered and avoided.

We give our confirmation and hope that detailed explanations provided for the examples mentioned above by the reviewer are enough.

Also in the list of references presented in PANGAEA, I noticed that there Bhat et al. (1969) is cited as Bhat et al. (1968).

The reviewer is right. We identified the source of error being a mistake with the information provided in the
websiteofScienceDirect

(https://reader.elsevier.com/reader/sd/pii/S0012821X68800834?token=CC3E25D891574BDB2CF57EE17B3C49D C3B7EF1FC953A5466363E1B39760BDE5486F60984002D1FAA658D0A9638A3B42F&originRegion=us-east1&originCreation=20220328112051). We have made aware both the ScienceDirect and PANGAEA and requested the changes.

I also saw that the personal communications and some reports (e.g., Kawakami, 2012) as well as some PhD theses (Owens, Pabortsava and Thomalla theses are included but Luo's thesis is not) are not included in the list of references. I believe they should all be shown there too since their data is included in the data set.

We agree with the review that all references should be included and this changes have been requested to PANGAEA. This includes datasets published in repositories by Turnewitsch 2001 and Kawakami, 2012, PhD dissertation by Luo in 2013, and the 9 datasets reported by personal communication.

L33: "half-lives" is plural, which "accounts" singular

Corrected.

L34: Transforming into

Corrected.

L36: radionuclides' distributions

**Corrected.**

L85-88: Adding the 5 different categories described results in 56631 data points instead of 56633, as stated. Please double check the numbers

**Checked and corrected.**

L96: Unless I missed one, I can only see 4 PhD dissertations in Table 1: Thomalla, 2007; Luo, 2013; Owens, 2013; Pabortsava, 2014, although Owens, 2013 is separated in 3 rows linked to 3 areas and/or sampling periods.

The reviewer is right the "T" used to indicate data extracted from a PhD dissertations was missing in Table 1. It corresponds to Lemaitre, N.: Multi-proxy approach (234Th, Baxs) of export and remineralisation fluxes of carbon and biogenic elements associated ith the oceanic biological pump. PhD dissertation, Joint PhD with Université de Bretagne Occidentale, Brest, France, Brussels, France. [online] Available from: https://we.vub.ac.be/en/phd-nolwenn-lemaître, 2017. The "T" has now been included.

L150: remove "in": "will be included"

Removed.

L154: "in future version"; change to "versions" or add an "a": "in a future version"

"a" added.

L253: Remove the dot after the Yool citation and make 15 a superscript (15N)

**Removed.**

L292-293: GA02 seems to be reported in a confusing way here. GA02 published by Owens et al. (2015) takes place in the South Atlantic, it goes from Punta Arenas (Chile) up to the equator, not from Denmark to South Atlantic as stated. Not sure why "Denmark" is specified for the Puigcorbé et al. (2017a) publication, when the authors already said it went from 64°N to the equator (it started south east of Greenland). The purpose of the information provided in this part of the text was to indicate the country leading the cruise, which in the case of GA02 was Netherlands, not Denmark. Mistakes in reporting these cruises have been corrected.

L362: "the concept of scavenging by particles has really emerged". The sentence should be in past tense. Remove "has".

**Removed.**

L368: Not sure what "are rank" means in this sentence.

**Corrected.**

L376-379: "Several authors combined total and particulate 234Th concentrations (e.g., Minagawa and Tsunogai 1980; Santschi et al., 1979). While the sampling of the 234Th dissolved phase was initiated by McKee et al. (1984) ...". I don't think this sentence is accurate. Santschi et al. (1979) publication already differentiates between total, particulate (>0.45 @m in this case) and dissolved (<0.45 @m) (see Fig. 2). Also, it is a bit confusing to say "combined total and particulate 234Th concentrations", sounds like if they were adding both things together, while, what I think the authors mean, is that those authors published data for total and particulate, while not for dissolved. Is that correct?

The reviewer is right about publication by Santschi et al. (1979). This part of the text has been corrected.

L383: Use "that" instead of "this" to refer to Coale and Brunland (1985) study.

Corrected.

L407: The ")" after 2003 should be deleted

Deleted.

L408: Reconsider based on my comment above about Santschi et al. (1979) publication

Modified based on reviewer's comment about Santschi et al. (1979) publication.

L415: Correct "field work" instead of "worked"

Corrected.

L427: "seawater" instead of "sea-water"

**Corrected.**

L428: "elements" instead of "radionuclides" since it is not a radionuclide selective technique.

**Corrected.**

L445-446: Santschi et al. (1979) work is actually cited here as one of the studies measuring the dissolved fraction, which does not agree with the previous statement (see comment L376-379)

This issue has been solved with the changes made and explained in comments for L376-379.

L456: it should be TSS

**Corrected.**

L474: Do you mean the "V" term in general (physical advection and diffusion) or just advection?

Yes, we meant that. This sentence has been rephrased for clarification.

L480: Correct "marked by a new approach was introduced"

**Corrected.**

L494: "element:234Th" should have a capital E

**Corrected.**

L499: It should be "advective" instead of "adjective"

**Corrected.**

L501-503: "In the open ocean, the most relevant physical process is vertical upwelling..." Well, that depends if it in that area there is upwelling or not. There are also areas with downwelling. Please rephrase to introduce the problematic of neglecting physical transport in certain areas of the open ocean.

Done.

L503-505: It seems very specific to talk about vertical and horizontal advections. Can this sentence be made a bit more general?

**Sentenced modified to be more general.**

L505: The jump to NSS seems a bit abrupt since we were talking about the physical part, not the temporal. The "alternatively", refers to SS, which was mentioned 6 lines above, before talking about advection and upwelling.

**Sentence rephased to make less abrupt.**

**L518: V term (singular)**

**Corrected.**

L518-519: I am not sure it is necessary to explicitly say "without these components, the sinking flux would have been....". There is no explanation of the impact of other milestones, e.g., L515 the authors don't say: "Use of 210Pb-210Po which will allow to evaluate the POC export flux over longer timescales". Plus, the importance of considering the V term in certain ocean settings is already described above in L500-505.

**Sentence mentioned by the reviewer removed.**

L523-527: Similar comment regarding the impact of a certain milestone. Maybe summarize it by saying something like: "... by Rutgers van der Loeff and Moore (1999) to allow particulate and dissolved concentrations to be analyzed from a single aliquot and to be measured by beta counting on board". This explains what the milestone did without explaining the particular impact that it had (i.e., residence time calculations in two phases; increased temporal and spatial resolution). I believe the impact of certain milestones should be introduced/explained/presented in the paragraphs prior to the milestones bullet points. If the authors decide to keep the impact of these milestones, please review other milestones to add their impact.

**Done.**

L536-539: "Only 6 studies sampled 2 size fractions..." but there are 8 publications (studies) cited, plus 2 more that used 70m instead of 53m, so 10 publications (studies) in total. Please rephrase and check the numbers.

**Numbers have been checked and sentence rephrased for clarification.**

L539-540: "Additionally, to a lesser extent....by a total of 6 studies". Why by a lesser extent if the number of studies that provides 2 size fractionations is also stated to be 6? If referring to all the studies that used pumps vs traps, then you should provide the number of studies using pumps to allow a proper comparison between both sampling methods.

We agree with the reviewer on his observation and "to a lesser extent" has been removed.

L569: "Pacific Ocean and had two...". Change "and" by "which"

**Corrected.**

L571: DYFAMED has been sampled repeatedly, not just in 1987, within the years of this era. I understand that the sampling year does not determine the era, the publication does, but see comments below regarding Schmidt et al. publications, for consistency with prior statements.

We thank the reviewer for this observation and agree with him. New references has been included to acknowledge the repeated sampling of DYFAMED beyond era 2 (see response to next comment).

L571: 8) Should Schmidt et al. (2002)-Strong seasonality in particle dynamics of north-western Mediterranean surface waters as revealed by 234Th/238U as well as Schmidt et al. (2005) 234Th measured particle export from surface waters in north-western Mediterranean: comparison of spring and autumn periods, be cited here too, mentioning "extended in the next era" as done before in L560, 561-562, 565-566, 568-569?

We thank the reviewer for this observation and agree with him. The reference has been included.

L580: Should Schmidt et al. (2002)- Particle residence times in surface waters over the north-western Iberian Margin: comparison of pre-upwelling and winter periods, be cited here too, mentioning "extended in the next era" as done before in L560, 561-562, 565-566, 568-569?

We thank the reviewer for this observation and agree with him. The reference has been included.

L584: "increased" instead of "increase"

**Corrected.**

L605: "avoided" instead of "avoid"

**Corrected.**

L606: "along with other biogeochemical parameters", this is not a direct consequence of the smaller sample volume used, meaning that the other parameters did not see their sampling resolution modified as a consequence of the small volume technique used for 234Th. I am not sure I am following what the authors mean with this sentence, but I believe the following paragraph talks about what the small volume technique allowed regarding the coupling with other parameters. I would delete the "along with other biogeochemical parameters" here since it is well explained and flows better in the following paragraph.

**We agree with the review the sentence pointed out was out of context and it has therefore been deleted.**

L614-615: This sentence reads a bit weird "by Pike et al. (2005) and Cai et al., (2006), marked the major milestones of this era, whose timeline summaries as follows". Should it be "Pike et al. (2005) and Cai et al. (2006) works, which are major milestones of this era, whose timeline..."?

**Corrected.**

L619: Close the ( ) of the coordinates.

**Corrected**

L628: Here, but also seen in other cases, "in-situ", but "in situ" is also used. The journal might have a specific criterion, but the authors should pick one of the two forms.

**We have picked in-situ and corrected that throughout the entire manuscript.**

L638 and 644: The Schimdt et al. (2002a,b) publications are in this area but the sampling took place during the previous era (1997-98 and 1994, respectively). This paragraph (L636-647) starts by referring to sampling "Sampling in the context of selected....". I mentioned these studies before (comments to L571 and L580). I know eras use publication dates and not sampling dates, so maybe just rephrase slightly the beginning of this paragraph.

**Rephrased.**

L638: "And some other new...". How about starting the sentence differently? e.g., "Other new studies, such as.... were also initiated".

Sentence rephrased as suggested by the reviewer.

**L640: There are 2 dots after (2004). Same for L647 after (2007)**

**Extra dots removed.**

L649: Same comment regarding the reference to sampling instead of publication. Maybe say something like "The publications included in this era report data covering the entire ocean and with increased number of data points in all regions, except the Indian Ocean..."

**Sentence replaced by the once suggested by the reviewer.**

Maybe in L349-353, where the authors explain that the 4 eras are determined by publications, highlight the fact that that means that cruises that took place within the time frame of a previous era are included in the era of their publication. Also, maybe you could also mention the fact that, based on what it is shown in Table 1, if a later publication (from a later era) used data from a cruise that was already published in a previous era, that publication is included in the earlier era, despite of the year of the actual publication, correct? Examples: Stewart et al. 2011 is included in era 3 despite being published during the time frame of era 4; Murray et al. (2005) belongs to era 2 although based on the year of publication should be from era 3, same for Friedrich and Rutgers van der Loeff (2002b). By the way, in Table 1 this reference has a "b" after 2002, but there is only one publication by Friedrich and Rutgers van der Loeff in 2002, so the "b" should be removed from the citation in Table 1.

We consider this suggestion by the reviewer very useful and clarifying and it has been included in the reviewed version of the manuscript (see lines 354-359). The reviewer is absolutely right about the examples made regarding table 1.

**The "b" in the reference Friedrich and Rutgers van der Loeff 2002 has been removed.**

I think that the fact that publication dates are leading the eras is a very valid way to divide the history of 234Th. Based on that, when the authors say, for example "a total of 114 cruises...where compiled in era 4" that includes cruises that might have taken place before 2010, correct? I think this should be strongly highlighted (a bit stronger than it is already said) at the beginning of the discussion because I find myself thinking that 114 cruises in era 4 means 114 cruises between 2010 and present, but some go as far back as 1997. I believe the confusion makes sense because there are sentences such as L633: "...golden age of 234Th so far in terms of cruises carried out by year". Those cruises were not carried out all in this era, but reported, as stated afterwards. Similar with L654: "the large number of field expeditions carried out during this era", again, these cruises did not take place during this era, their data was reported during this era, which is different. I think the authors usually use "report" or similar verbs, but I would ask them to please be careful and double check that there are no sentences, like the two I just mentioned, that can lead to confusion. I have my doubts about considering the number of cruises included in each era as an indication of the "productivity" of the era, since there is a known delay between sampling and publication, maybe the number of publications would be more indicative of such "golden era", but I guess that is just my personal opinion.

As stated on a previous comment mentioning this issue, we consider this suggestion by the reviewer very useful and clarifying and new text has been included at the beginning of the discussion to emphasize the delay between sampling date and publication date, which determines to which era a data set belongs (see lines 354-359).

Additionally, all text has been reviewed to replaced "sampling" by "published" and "reported" when describing datasets within an era. And number of publications has been used as a indicative of productivity of the eras (see lines 646-647 and line 739).

L662: Delete one set of ( ) in the Kawakami et al. 2004 and Smoak et al. 2004 citation.

Deleted. L668: Remove ( before Lepore Removed. L670: "leagues" instead of "legs" Corrected. L678: "synthesis" should be plural, "syntheses" Corrected L690: I think the order should be (Oregon, USA) Corrected. L691: Change the comma for a dot after "sampling"

**Corrected.**

L692-694: Check the parentheses, there is a bit of everything in each of the IDP citations

**Corrected.**

L695: Mention that the shift that you are referring to is linked to 234Th, since GEOTRACES is much bigger than 234Th. The POC reference is only for Th while the reference to standards, intercalibrations, protocols, etc. applies to many other trace elements and isotopes. Rephrase a bit to clarify it.

Done.

L698-699: Check the parentheses

Corrected.

L706: "as part of the carried out in the context" - Correct it

**Corrected.**

L716: Missing a "has": "...technique has been recently carried out..."

**Corrected.**

L732: "published 4 years later", remove the "a"

**Corrected**

L743-746: The numbers don't match: 64 studies (out of 82) modeled 234Th, from them 60 used the SS approach + 1 NSS + 26 both (SS&NSS) = 87 studies total

We thank the reviewer for spotting this mistake, in the reviewed version of the manuscript it has been corrected.

L749: Add a space after (34%) and "of" after measurements or delete "measurements" and just say "reported CHN data"

**Corrected.**

L751-753: I am not sure the reference to studies that derive POC fluxes from Po and Th and compare them to sediment traps is necessary since that is not a data parameter reported in the data set (only if there is Po data or sediment trap data, but not if the authors derived POC fluxes from that and compared among the different approaches). It's ok if the authors consider it is valuable information for the reader but, is in this era the first time that this was done? If yes, say it, if not, then write a similar sentence on the previous eras were this sort of multiple tracer/direct measurements of POC fluxes was done. Note that Stewart et al. (2010) is not included in Table 1 because the 234Th data for that study are "presented elsewhere", which highlights what I was pointing out: having studies comparing POC fluxes obtained through Th-Po-Traps is not a variable of the data set. Therefore, highlight that this is something new seen in this era, but don't list it as if it was the last of the parameters gathered for the data compilation.

We consider the reviewer made a good point here and the reference to the studies mentioned has been deleted in the reviewed version of the manuscript.

L758: PICCOLO requires more information describing the goals, similar to the other projects.

**Done.**

L776-777: Delete the info about MOBYDICK, already shown in L767-769

Deleted.

L778-799: Fix singular and plurals; e.g "There is projects", "that include", "their methodologies". Are you referring to one (APERO) or more (APERO and JETZON)? I would say just one, APERO, since JETZON is a network, and it includes APERO in it.

Corrected to singular.

L793: "version" should be plural

Corrected.

L833: remove "in"

Removed.

L843-844: Check parentheses for the Rosengard and Owens citations

**Checked and corrected.**

L846: "expect to resolve the mechanisms that control the transfer of carbon to depth" – It seems a bit vague and also quite bold to say that having the PAR or fluorescence profiles will tell us about the mechanisms driving the transfer of carbon to depth. What do you really mean?

Done.

L847-848: "However.....CTD (ref)". This has already been said recently (L841). Missing the reference.

Sentence in line L841 has been removed to avoid repetition and missing reference has been added.

L849: Correct "Is that for reason that"

Corrected.

L850-851: There is an "and" and a verb missing: "...fluorescence and PAR data are crucial..."

**Corrected.**

L852-853: This sentence is exactly the same that the one at the end of the previous paragraph (L837-838). I would delete it in the previous paragraph (L837-838) and leave this one, but avoid repeating it, especially when they are so close together.

Sentence repeated removed when appears for the first time.

L854-859: Again, two exact sentences, in this case one after the other: "...overall, only a 36%....234Th fluxes."

Sentence repeated removed when appears for the second time.

L863: Change "as" for "of" – "...as much of this information..."

**Changed.**

L864-866: I am not sure I understand exactly what that means. Do you mean that, regardless of the authors proving the bloom stage or not you will assess it somehow? Do you mean that you will provide what criterion was used when a bloom stage was reported? Will this be done for all the future publications/data sets that will be included? Could you please expand a bit or clarify what you mean?

Yes, we mean that in a future version of the compilation we would like to apply a standard criterion to the whole compilation to define the bloom stage, regardless of the authors proving the bloom stage. We plan to work on a criteria that takes into consideration the authors criteria, but most importantly, we plan to classify those dataset with no bloom stage assigned. The criterion used will be provided in the metadata information and will be added in addition to the classification provided by the data authors, when available. This will be applied to the datasets they are already present in the compilation and to those to be included in future versions of it. We have clarified this point in the reviewed version of the manuscript since it was not clear before.

L875: "ratios in the upper 100 m" which would include all the samples from surface to 100 m. But then in L879-880 "...probability of 234Th reaching equilibrium (or not) with its parent at 100 m", which sounds like only data at 100 m is being compared. I think it makes sense to do the second option, because we usually observe a deficit above 100 m, which would bias the histogram. Later on in L883 "...reach equilibrium around 100 m depth". What is the reference depth used? If you only plotted data collected at 100 m, please make it clear, if you used 100 m +/- 10 m (or similar) it is ok, just say that, but avoid saying "within the upper 100 m" because that includes all the data points collected from 0 to 100 m.

The reviewer made a really good point here. This part of the text has been rephased to clarify that only data at 100 m +/- 10 m is being compared.

L912: Correct the reference to Buesseler 2020

Corrected.

L913: It should be "BCP strength"

**Corrected.**

L921-922: Not sure what the authors mean by "major features...with an average of 2.03+/-0.18 dpm/L". This seems the average Th-234 concentration in surface waters.... Is this the "major featureS" (notice that features is plural)? I think the sentence needs to be clarified.

Sentence rephrased for clarification. Features removed.

Table 1: Remove the "-" from Matsumoto, 1975 sampling period.

Removed.

I just realized that PON:234Th ratios are only considered for filtration methods. Is that because there were no studies providing such data for sediment traps?

Only 10 dataset out of the 223 compiled reported PON:234Th data from sediment traps (in comparison to the 64 studies that reported POC:234Th with traps). Regarding datapoints, there are 196 PON:234Th data from sediment traps (in comparison to the 1244 datapoints of POC:23Th reported with sediment traps). So, we did not consider PON:234Th ratios measure in traps relevant enough to be included in the Table, which already contains lots of information.

Check the title for era 3, it says 2001-20010, there is an extra 0 in 2010.

**Corrected to 2009.**

Figure 2: The labeling of the panels based on the information provided in the caption is incorrect since panel c) should be panel b).

**Corrected.**

Figure 4: It took me a while to understand the M#. There is not reference to that in the list of milestones below each era, and I think that would make it easier. What I mean is...Would it be possible to have the "arrow" with the year and next to it the M#? Right now, for example for the 1989 I see grey xxxxxx in front of what would be M11 and M12. By the way, shouldn't 1989 have 3 milestones? 1) BATS, 2) Th as tracer of export production and 3) OSP-PAPA? In the timeline 1984 has 3 milestones (M4, M5 and M6) but in the list shown below era 1 for 1984 we can only see one, "concept of great particle conspiracy" [2]. The opposite happens in 1977, where there is only one milestone in the timeline (M3) but in the list below era 1 it looks like there should be 2 milestones: 1) the beginning of JGOFS program and 2) the Mn-cartridges [3]. Please review the figure and consider my suggestion of including the M# in the listing below each era.

We agree with the reviewer that including the milestone number (M#) after the arrow with the year make the reading of the figure easier and we have included it in the reviewed version of the figure.

We also would like to apologies to the reviewer for the many mistakes in the previous version of Figure 3 in era 1 which complicated the understanding if it. We have carefully reviewed the figure and we believe all mistakes are solved and both the timeline and the list with the items describing the milestones match the text in Section 4.

For example, for the case of era 1 where we spotted several mistakes, there are now 10 relevant years within the era, each one of them has a milestone, expect for years 1984 and 1988 which has 2 milestones. This results in a total of 12 milestones in this era (from M1 to M12, as plotted in the timeline and indicate in the itemized list of this era in the figure).

In era 2 we identified a mistake in the last 2 years with milestones in the timeline, which we indicated as 1997 and 1998 but are actually 1998 and 1999. See it corrected now.

In the rest of eras, we have not identified mistakes.

Again, tank to the reviewer for helping us clarifying this figure, which is crucial for the manuscript.

Figure 4: This comment is related to that long one I made before about the confusion between eras linked to publications and not to cruises. Saying "...sampling stations during era 1" can lead to think that those stations were sampled between 1969 and 1991. Maybe add something like "sampling stations considered (or reported) within era 1".

**Corrected as suggested by the reviewer.**

The type of data "iii) total 234Th not sampled" in panels b), c) and d), means that just one fraction of 234Th (i.e., either dissolved or particulate, or both maybe both, but not added by the authors) was sampled?

The reviewer is right. The "no 234Th" category referred to by the reviewer means that from the 234Th data gathered in the compilation (i.e., dissolved, particulate and total 234Th, and POC:234Th ratios and PON:234Th ratios from both filtration methods and sediment traps) total 234Th has not been reported in that dataset but at

least one of the other parameters has been reported. Same reasoning for the "POC:234Th ratios" category. This has been clarified in the caption of the figure.

Is it correct to assume that those stations in the north east of Greenland in panel b) are plotted twice because they belong to two of the categories (i.e., no total 234Th and no POC/Th ratios)? Similar to those in the Caribbean? This means that one study can be in two categories, correct? If that's the case, maybe it could be useful to have a note in the caption mentioning it. If that is correct, however, I'm not sure being part of category ii) as stated, means that it can be used to estimate POC fluxes because a study could be part of category ii) "no 238U" but also be part of category iv) "no POC/Th ratio", correct? If that's the case, being part of category ii) does not imply being useful to obtain POC fluxes.

The reviewer is again absolutely right in his reasoning and therefore, only datasets in category i) can be used to estimate POC fluxes. This has been corrected in the caption of Figure 4.

Figure 5: Similar comment. Please rephrase to ensure clarity: "...annual field expeditions reported within eras...."

Corrected.

Figure 7: Similar to comment from L875 et al.: Maybe it is better to say "at 100 m" instead of "within the upper 100 m"

**Corrected.**

Cite Davidson et al., (2021) for panel b)?

Citation included.